# A Percolation Model of Emergence: Analyzing Transformers Trained on a Formal Language

**Ekdeep Singh Lubana**[*1,5], **Kyogo Kawaguchi**[*2,3,4], **Robert P. Dick**[5], **Hidenori Tanaka**[1,6]

[1]CBS-NTT Program in Physics of Intelligence, Harvard University
[2]Nonequilibrium Physics of Living Matter RIKEN Hakubi Research Team,
  RIKEN Center for Biosystems Dynamics Research
[3]RIKEN Cluster for Pioneering Research
[4]Institute for Physics of Intelligence, Department of Physics, The University of Tokyo
[5]EECS Department, University of Michigan, Ann Arbor
[6]Physics & Informatics Laboratories, NTT Research, Inc., Sunnyvale, CA

## Abstract

Increase in data, size, or compute can lead to sudden learning of specific capabilities by a neural network—a phenomenon often called "emergence". Beyond scientific understanding, establishing the causal factors underlying such emergent capabilities is crucial to enable risk regulation frameworks for AI. In this work, we seek inspiration from study of emergent properties in other fields and propose a phenomenological definition for the concept in the context of neural networks. Our definition implicates the acquisition of general regularities underlying the data-generating process as a cause of sudden performance growth for specific, narrower tasks. We empirically investigate this definition by proposing an experimental system grounded in a context-sensitive formal language, and find that Transformers trained to perform tasks on top of strings from this language indeed exhibit emergent capabilities. Specifically, we show that once the language's underlying grammar and context-sensitivity inducing regularities are learned by the model, performance on narrower tasks suddenly begins to improve. We then analogize our network's learning dynamics with the process of *percolation on a bipartite graph*, establishing a formal phase transition model that predicts the shift in the point of emergence observed in our experiments when intervening on the data regularities. Overall, our experimental and theoretical frameworks yield a step towards better defining, characterizing, and predicting emergence in neural networks.

## 1 Introduction

Modern neural networks, e.g., large language models (LLMs) (Gemini Team, 2023; OpenAI, 2023; Anthropic, 2023; Touvron et al., 2023), exhibit a broad spectrum of capabilities, allowing them to serve as the "foundation" for downstream, application-specific systems (Bommasani et al., 2022; Ahn et al., 2022; Driess et al., 2023; Schick et al., 2024). As these models scale, either via addition of more data, parameters, or compute, an intriguing behavior is at times observed: until a certain critical scale is reached, there are capabilities that the model does not exhibit; however, beyond this point, such capabilities suddenly "emerge" (Wei et al., 2022; Srivastava et al., 2022; Brown et al., 2020; Yu et al., 2022; Steinhardt, 2023; Pan et al., 2022; Anil et al., 2023; Kirsch et al., 2022; He et al., 2024; Elhage et al., 2021; Tigges et al., 2024). More specifically, the performance of the model on a task or benchmark meant to evaluate said capabilities witnesses substantial growth in performance, even though the *overall* training loss undergoes minimal, if any, improvements (Arora & Goyal, 2023; Du et al., 2024). Empirical evidence in fact suggests that, at times, several capabilities can emerge simultaneously (Wei et al., 2022; Wei, 2022).

Beyond developing a better scientific understanding of neural networks, understanding emergent capabilities is crucial to enable risk-centric regulation frameworks for AI, which assume a system's capabilities can be preemptively conjectured (NIST, 2023; EU Council, 2024; OSTP, 2023; Kaminski, 2023). To this end, recent work has made attempts at identifying factors that decide whether a

---

*Equal contribution. Code at `https://github.com/EkdeepSLubana/ConceptPercolation`.

capability will emerge. For example, Okawa et al. (2023) and Arora & Goyal (2023) implicate the underlying compositional structure of a capability as the cause for its sudden learning. Hoffmann et al. (2023) argue capabilities that involve interactions between specialized components within a model are likely to yield sudden performance improvements once the correct interaction mechanism is learned; e.g., the interaction between the *previous token* and *copy* attention heads to enable in-context learning (Elhage et al., 2021; Singh et al., 2024; Reddy, 2023). Meanwhile, Schaeffer et al. (2023) argue emergent abilities are an artifact of poorly defined, discontinuous evaluation metrics, claiming that models undergo continuous, persistent improvements during training. Recent work has however demonstrated that even continuous metrics can witness sudden improvements, with such changes co-occurring with the model's learning of a new capability (Chen et al., 2024; Du et al., 2024). This undermines the claim that emergent capabilities are merely an artifact of evaluation protocols.

Taken together, the orthogonal explanations and disparate results above have resulted in emergence becoming an unclear phenomenon in machine learning. At its core, however, we claim that the concept has never been defined in prior work (see App. B for a detailed related work). This has arguably led to distinct mechanisms causing sudden changes in model performance to all be labeled as "emergence". What is the *phenomenology* that this term is meant to capture in the context of neural networks? Is it merely a sudden increase in performance with scale, or *broader than that*? Given a reasonable definition, can we show, even if in a simplified system, that emergent capabilities are commonplace? Can we use this simplicity to better *understand what drives their sudden learning*?

**This work.** To address the questions above, we propose a phenomenological definition for emergence and try to understand what drives it in a toy task of learning formal languages (Chomsky, 1956; Cagnetta & Wyart, 2024; Liu et al., 2023a; Wen et al., 2023; Liu et al., 2022a; Friedman et al., 2023; Jain et al., 2023; Merrill et al., 2023). Specifically, we argue three characteristics should be observed to claim a capability is emergent (see Def. 1): beyond (i) sudden performance improvement for a specific task, we claim emergence is more likely to represent a meaningful concept if (ii) performance on several tasks improves simultaneously and (iii) there are precise regularities underlying the data-generating process learned by the model at the point of emergence. The intuition, borrowed from the study of emergence in other fields (see Fig. 1), is that if multiple tasks witness improvement in performance, there is likely some shared structure to them and the model learns this structure at the point of emergence. For example, when in-context learning emerges in LLMs, the model learns that past context helps disambiguate the next token better (Elhage et al., 2021)—a regularity present in natural language. This leads to in-parallel improvement in several downstream tasks' performance (Wei et al., 2022; Wei, 2022; Lu et al., 2023); thus, in-context learning can be deemed an emergent capability under the scope of our definition. In this sense, understanding emergence can be formalized as a study of identifying which data-regularities the model learns at the point of sudden learning of *a set of* capabilities, and understanding why those regularities are relevant to said capabilities. Adopting this perspective, we make the following contributions.

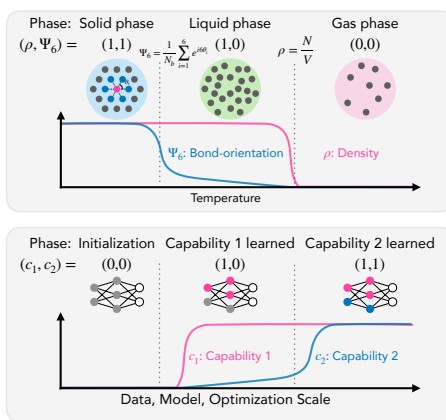

Figure 1: **Emergence as phases of learning.** Emergence is a well-characterized phenomenon in natural sciences (Anderson, 1972; Newman et al., 2001; Newman, 2003) and deeply entangled with the notion of phase changes, i.e., when change in some control variable (e.g., temperature) yields systematic changes in a system's underlying regularities (e.g., formation of hexagonal configurations in a crystal) and simultaneously affects several of its properties. We argue for a similar characterization of emergence in machine learning: identifying systematic changes in a model's behavior that influence its downstream abilities and lead to sudden performance improvements. For example, learning a language's syntax will affect all downstream capabilities where coherent, grammatically correct generations are necessary.

- **Formal Languages as an Experimental System for Studying Emergence.** We define a probabilistic context-sensitive grammar (PCSG) with type constraints that allow an entity or a subject in a sentence (e.g., man) to be seen in the context of only a predefined set of properties (e.g., walk).

We train models to perform minimalistic reasoning tasks over samples of this language and find their data scaling curves simultaneously show sudden learning across several metrics.

- **Learning of general data regularities underlies simultaneous jumps in specific metrics.** We find points of sudden change for metrics evaluating individual tasks correlate with the model learning two relevant regularities that underlie our formal language: grammatical rules and type constraints. Despite the simplicity of our setup, we claim learning of such data-regularities is what leads to sudden growth in the performance of narrower tasks in large-scale models.

- **A percolation model predicts the scaling of when capabilities emerge.** We propose a formal model grounded in the theory of graph percolation (Cohen et al., 2002) that captures our experimental observations, and show that if we can describe the regularities the model is learning at the point of emergence, a predictive theory for sudden learning can (at times) be constructed—analogous to theories of phase transitions in physics; see Fig. 1.

## 2   A PHENOMENOLOGICAL DEFINITION OF EMERGENCE

To analyze emergence, we first establish what we mean by the term for the purpose of this work. Specifically, we define emergence in a phenomenological manner, i.e., by assembling the characteristic properties associated with scaling curves claimed to depict emergent learning. We emphasize our definition is merely *a* definition for emergence, and does not necessarily represent all possible perspectives (Luccioni & Rogers, 2023).

**Definition 1.** *(Emergence of a capability.) We say a capability $\mathcal{C}$ is emergent with scaling along a relevant axis (e.g., amount of data, compute, parameters) if:*

- *discontinuous improvement occurs in the performance of a task where $\mathcal{C}$ is required;*
- *multiple tasks simultaneously show discontinuous performance improvement; and*
- *the model learns regularities underlying the data generating process such that discontinuous progress in $\mathcal{C}$'s learning directly correlates with the learning of said regularities.*

The definition above assigns a broader meaning to emergence than mere sudden performance improvement on a narrow task: it argues there should be precise regularities underlying the data that are learned by the model, yielding downstream effects on *several* capabilities and hence sudden improvements in performance of several tasks. Note that we intentionally leave the notion of 'regularity' informal in the definition. The salient property of a regularity is that if a model learns it, downstream tasks should become easier to perform. For example, a fine-grained notion of a data regularity can be learning of context-sensitivity, which can aid in-context learning (Reddy, 2023; Edelman et al., 2024; Olsson et al., 2022); a more coarse-grained regularity can include the model learning the syntactical rules of a language that help it with generation of coherent language and hence with any task where coherence is important (Chen et al., 2024). *In this sense, what is emergent is the learning of a regularity, and what is observed is a change in the model's capabilities.* Hypothesizing what this regularity is by identifying shared characteristics of a set of tasks that simultaneously show sudden learning, one can likely develop an evaluation meant to precisely gauge learning of the corresponding regularity and hence infer at what point an independent training run will show sudden improvements.

We note the intuition for Def. 1 comes from prior work in the fields of complex systems and physics (Anderson, 1972; Newman et al., 2001; Newman, 2003), from where the term has sought its inspiration in recent machine learning literature (Steinhardt, 2023; Wei et al., 2022). Therein, emergence describes the scenario where rapid changes occur in a system's properties as some control parameter is varied. A range where the system's properties change relatively smoothly is called a phase, and a change of phase with a change in the control variable is called a phase transition. A crucial step in studying emergence in physics is identifying an *order parameter*—a measure that captures the formation of some specific regularity in the system such that the development of this regularity is what alters the system's properties and drives a phase transition. For example, in Fig. 1a, a system of particles transitions through phases (solid, liquid, gas) as the temperature is changed; the formation of a crystalline structure with the decrease in temperature can be identified by analyzing the bond-orientation order parameter, while the liquid-to-gas transition can be described by a jump in particle density. We argue that we must similarly define order parameters for studying emergence in neural networks as well, i.e., we must develop evaluation measures that are focused towards detecting the learning of *specific data regularities* that are *generally of use* to several downstream capabilities.

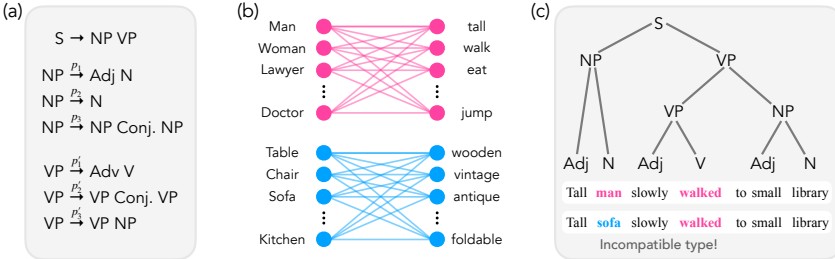

Figure 2: **Grammar and type constraints to define our formal language.** (a) We use a PCFG to define our language's grammar (shown rules are examples; see App. F.3 for precise details). The grammar's terminals are parts-of-speech from English and yield symbolic sentences that can be populated by tokens from the language's vocabulary. (b) Akin to natural language, wherein properties of an entity constrain sentences seen in a dataset corresponding to that entity, we define constraints (called type constraints) on our language that restrict which tokens can be seen together in a sentence. These constraints map entities to descriptive or relative properties, hence restricting which descriptive adjectives and verbs are valid for an entity. (c) Once a symbolic sentence is sampled from the grammar, we populate it with tokens from the language while respecting the type constraints. Training on string from this language in fact shows that the model deems sentences that do not respect type constraints to be extremely unlikely (see App. F.2).

## 3 Formal Languages as an Experimental System for Emergence

Having defined our perspective on emergence, we now define a toy experimental system that allows us to precisely study the concept in a controlled setting. We note that our focus will be on emergence under data scaling in an online learning scenario (i.e., a sample is unlikely to be seen multiple times). To this end, we follow recent work on understanding language modeling and use formal languages to define our experimental setup (Allen-Zhu & Li, 2023; Jain et al., 2023; Murty et al., 2023; Valvoda et al., 2022; Liu et al., 2023a; 2022a). As discussed in detail next, the formal language we use in this work is (minimally) context-sensitive, with underlying syntactical rules defined using a probabilistic context-*free* grammar (PCFG) and context-sensitivity enabled through posthoc *type constraints*. The grammar and type constraints serve as two regularities that underlie our language, and, as we show in Sec. 5, their learning bottlenecks learning of other, narrower capabilities.

**Definition 2.** *(Grammar.) A PCFG, denoted* G, *is a randomized process that generates strings using two sets of symbols:* **terminals** (T) *and* **non-terminals** (NT). *A string* $\sigma$ *is a sequence of terminals, ending with a special* **end-of-string** *symbol* (EOS), *e.g.,* $\sigma = t_1 t_2 \ldots$ EOS. *The grammar includes* **production rules** *(denoted* R*), which define how non-terminals can be expanded. Specifically, a rule has the form* $A \rightarrow \alpha$, *where* $A \in$ NT *(the left-hand side) and* $\alpha$ *(the right-hand side) is a sequence of symbols from* T $\cup$ NT. *The process begins with a special* **start symbol** (S $\in$ NT). *To generate a string, the PCFG repeatedly applies production rules: a non-terminal is replaced by the sequence on the right-hand side of a randomly chosen rule with that non-terminal on the left-hand side. This process continues until all non-terminals are replaced by terminals, producing a string* $\sigma$. *The set of all strings the grammar can generate is denoted as* $\Sigma$. *A string* $\sigma$ *is called* **grammatical** *if* $\sigma \in \Sigma$.

See Fig. 2 (a) for a visualization and App. C.1 for a more detailed treatment of PCFGs in general. The terminal symbols used in our work include standard parts-of-speech from English, specifically: subjects, objects, verbs, adjectives, adverbs, conjunctions, determiners, and prepositions. Multiple short phrases can be combined together via conjunctions and verbs to form longer sentences (e.g., a verb can connect a subject phrase and an object phrase). Overall, the grammar G yields *symbolic* strings that are solely comprised of parts of speech (see Fig. 2 (a)); e.g., our grammar might yield a symbolic string like `adjective subject adverb verb preposition adjective object`. We next map these strings to our language.

Let $\mathcal{V}$ denote the vocabulary of our language $\mathcal{L}$. Each token $v \in \mathcal{V}$ has a part-of-speech $t := \texttt{part}(v)$ associated with it. Thus, one can define a *context-free* language by simply sampling a symbolic string $\sigma$ from G, and then replacing symbols therein with tokens from the vocabulary that match the sampled parts (e.g., a `subject` symbol may be replaced by the token `man`). For example, the example string above can be resolved as `Tall man slowly walked to short building`. However, natural language is rich with constraints defined by the physical properties of an entity, which thereby restrict which tokens are seen in the context of which other tokens, hence yielding *context-sensitivity*.

For example, one does not expect to see a sentence `Tall telephone slowly walked to short building`, since the entity `telephone` is neither expected to be `tall` nor the ability to `walk`. We develop an abstraction for such constraints by representing them as a bipartite graph (see Fig. 2 (b,c)). In the following, we often use the term 'entities' to jointly refer to subjects and objects. We also use the term 'descriptors' to refer to adjectives that modify entities, reserving the term 'adjectives' for ones that do not.

**Definition 3.** *(Type Constraints Graph.) Let a **property** $k$ be a binary variable; the set of all properties is denoted $K$. Properties can be either **descriptive** (used to define descriptors; e.g.,* `tall`*) or **relative** (used to define verbs; e.g.,* `walk`*). A **concept class** $C$ is defined via the set $K_C \subset K$ that denotes which properties are valid for that class. When the properties in $K_C$ take values, we get an entity $e$ from the class, denoted as $e \in C$. The set of all possible entities is denoted $E$. The **type constraints graph** $\mathcal{G} := (E, K, I)$ is a bipartite graph over entities and properties in the language whose edges $I$ denote whether an entity $e \in E$ possesses property $k \in K$.*

As an example, consider the class of `Humans`, which includes entities connected to properties like `tall`, `right-handed`, etc.; an entity from `Humans` will be assigned a subset of these properties. When sampling sentences from our formal language, the type constraints will restrict which tokens can be seen together, i.e., which descriptors and verbs go with an entity, hence yielding context-sensitivity and making $\mathcal{L}$ a probabilistic context-sensitive language. Given two randomly sampled entities from the same class, they can be expected to share a subset of properties, giving a signal to the model trained on $\mathcal{L}$ that these entities are related (i.e., they belong to the same class).

## 4 LEARNING TASKS AND EXPERIMENTAL SETUP

Having described our language $\mathcal{L}$, now we briefly discuss our experimental setup (see App. E for details). We train a GPT architecture model (Andrej Karpathy, 2023) with the standard autoregressive language modeling objective. Data is sampled "online", i.e., we sample a fresh batch of strings every iteration from $\mathcal{L}$. Unless mentioned otherwise, $\mathcal{L}$ is constituted of $|E| = 900$ entities and $|K| = 18000$ properties, equally and disjointly distributed over $|C| = 10$ classes, and with edges connecting entities to $p = 0.15$ fraction valid properties of a class in a uniformly random manner; results ablating these settings are in App. F. Before being fed into the model for training or evaluation, strings sampled from the language are restructured into a format that enables the specification of particular tasks (see Fig. 3). Specifically, we train the model to learn the following tasks with 80/10/10% splits.

Figure 3: **Task definitions.** Our model is trained and evaluated on three types of tasks. (i) Free generation: the model generates sentences with correct grammar. (ii) Unscrambling: the model is provided with a set of words and must reorder them to form valid sentences. (iii) Conditional generation: model is given a set of entities or properties and must generate valid sentences using them. Note that examples in the figure are merely indicative. See App. C.1 for details.

- **Free generation:** Produce a valid string, i.e., one that respects the grammar and type constraints.
- **Unscrambling:** A string is sampled from $\mathcal{L}$ and randomly permuted; the model is expected to unscramble it. This task is known to show sudden learning in LLMs (Wei et al., 2022).
- **Conditional Generation:** A set of tokens corresponding to entities or properties are shown to the model, which is expected to generate a string combining these tokens in a valid manner.

**Evaluation Protocols.** Given an input $x$, which may correspond to any of the three tasks above, denote the model output as $f(x)$. Let $\mathbb{1}(.)$ be an indicator variable that evaluates to 1 if its input is logically true. We often decompose evaluations according to strings of two types: (i) *descriptive*, i.e., ones that describe that an entity possesses a descriptive property, and (ii) *relative*, i.e., ones that demonstrate a subject, object, and verb can be combined to create a valid sentence. We track several metrics throughout training to avoid confounding from discontinuous scores (Schaeffer et al., 2023).

- **Grammaticality/Type Check.** Grammaticality involves checking whether model output follows the underlying grammar $G$, i.e., $\mathbb{1}(f(x) \in \Sigma)$; several other evaluations stress-testing how accurately the model learns the grammar, e.g., by assessing likelihoods of invalid sentences (see App. F.3–F.5). Type checks involve first extracting subjects, objects, and properties from the

sentence and then evaluating whether this set of tokens is allowed in the context of each other. We decompose type checks as *descriptive* (do entities and descriptors match), *relative* (do subject, object, and verb match), and *all* (product of all constraints, including adjectives and adverbs).

- **Exact Match / Per Token Accuracy.** Used for evaluating unscrambling. Assume the ground-truth unscrambled sentence $y$ has $l$ tokens. We compare whether the model output exactly matches the ground-truth $\left(\Pi_{i=1}^{l} \mathbb{1}(y_i = f(x)_i)\right)$ or the per-token match ratio $\left(1/l \sum_{i=1}^{l} \mathbb{1}(y_i = f(x)_i)\right)$.

- **Conditions Satisfied.** Used for evaluating conditional generation. If the model is expected to produce a sentence with $m$ conditioning tokens $\{v_{c_1}, \ldots, v_{c_m}\}$, we analyze how many of those tokens are present in $f(x)$, i.e., we evaluate $1/m \sum_{i=1}^{m} \mathbb{1}(v_{c_i} \in f(x))$.

- **Average Probability of Valid Tokens.** Used for evaluating descriptive type constraints in free generation and unscrambling. Specifically, we sample a sentence from $\mathcal{L}$ that remarks on an entity possessing a property (i.e., a descriptive sentence), and then evaluate the probability of the property being the next token when this sentence is inputted to the model. For example, let $x =$ The fire was large. We evaluate $\Pr\left(\mathbb{1}(f(x)_1 = \texttt{large})|x_{-1}\right)$, where $x_{-1}$ denotes the sentence up to the last token and $f(x)_1$ denotes the first token predicted by the model.

**Other Evaluations.** We perform several other evaluations, e.g., analyzing likelihoods of sentences that do not follow grammar or type constraints to check how well the model follows our language; comparing the distribution of lengths and parse tree depth for model's generations with the language's; analyzing grammaticality and type check accuracy for unscrambling and conditional generation; rank of property predictions; and the evolution of attention maps across time. See App. F for these results.

## 5 RESULTS: EMERGENT CAPABILITIES IN FORMAL LANGUAGE LEARNING

We now evaluate (i) whether our setup demonstrates emergence (see Def. 1), and (ii) whether we can extract insights into the mechanisms of what leads to emergence (all results are averaged over 3 seeds). While experiments below are for a specific setup, in App. F we show that our claims consistently generalize to an extremely broad array of setups and metrics. In the following, we often use the terms "phase" and "phase change"; see discussion around Def. 1 for context on these terms.

### 5.1 PHASES OF LANGUAGE AND CAPABILITIES ACQUISITION

We plot the model's performance as a function of training iterations. Since we are in an online learning, constant stepsize setting, this analysis corresponds to studying the effects of data scaling. Results are reported in Fig. 4 and show there are three phases to the learning dynamics.

**Phase 1: Grammar acquisition.** We find the model first learns to produce grammatically correct sentences, as measured by the grammaticality measure defined in Sec. 4. This process is relatively rapid, as we see the model starts generating grammatically accurate sentences in a short period of approximately 100 iterations; attention heads also rapidly evolve and reflect the parse structure of a sentence (see App. F.5) In this regime, however, the narrower tasks of unscrambling and conditional generation exhibit poor performance. However, precisely when grammaticality improves, we find that per-token accuracy starts to improve. This indicates that the model **learning a broad regularity underlying the data (i.e., grammar) impacts learning of narrower capabilities.**

**Phase 2: Acquisition of relative type constraints.** At around 1000 iterations, we find there is a sudden increase in the model's performance on relative types from essentially zero to perfect accuracy; precisely at this point, we find the loss for all tasks, especially free generation, shows a sudden drop. Interestingly, we find this sudden improvement occurs precisely at the point where the model reaches its maximum performance on grammaticality for the first time. That is, **as soon as the first regularity underlying the data is learned, the model rapidly learns the next relevant regularity of relative type constraints**. Improvement occurs in descriptive constraints as well (and hence the overall Type Check performance), but hovers around slightly above 0.1. This is expected, since with $|C| = 10$ classes, if a model produces grammatically correct sentences, it will achieve a random performance of $1/|C| = 0.1$ on descriptive type checks. This also implies that the model is primarily relying on its syntactical knowledge and does not respect descriptive type constraints much.

During this phase, we see that shortly after the phase change, there is a sudden increase in performance for both unscrambling and conditional generation, across all metrics. These tasks' losses also show another loss drop occurs at this point; though the drop seems smoother in the total loss, likely due

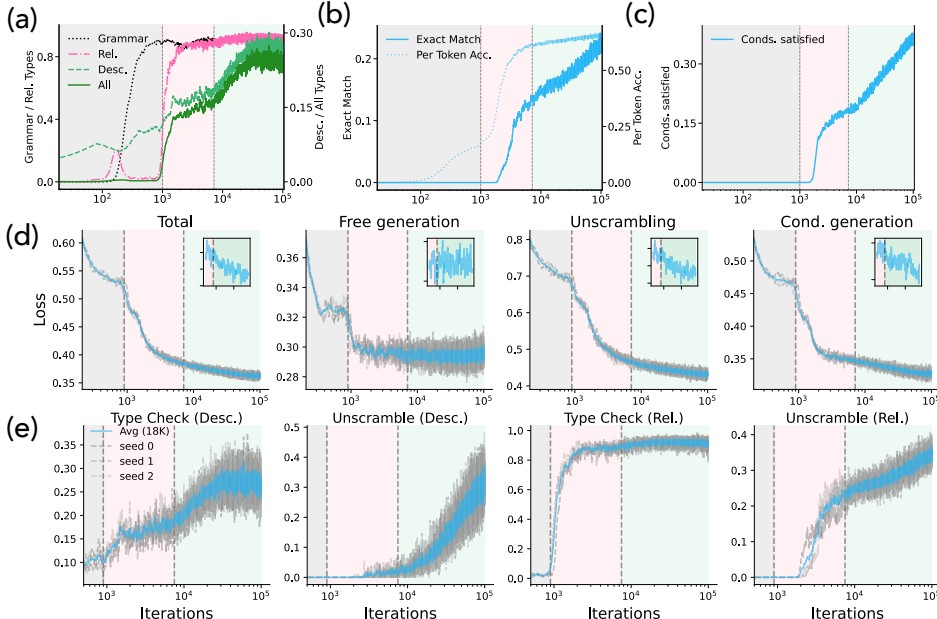

Figure 4: **Learning of regularities in the data drives emergent capabilities.** For a detailed discussion, see main text. **(a) Grammaticality and Type Check evaluations** as a function of iterations (or, equivalently, data). We see phases in the learning dynamics corresponding to *emergent acquisition of regularities underlying our language*: grammar, relative type constraints, and descriptive type constraints, shaded gray, pink, and green respectively. **(b, c) Performance on Unscrambling and Conditional Generation.** After a slight delay from phase boundaries, we see sudden improvements in the performance of individual tasks. **(d) Learning curves.** Loss also shows sudden changes at phase boundaries corresponding to the learning of regularities present in the data. **(e) Performance on descriptive/relative sentences.** Decomposing by sentence type, we find a sublinear growth in descriptive type checks drives performance boost on descriptive sentences for the unscrambling task.

to averaging effects (Michaud et al., 2023). As shown in Fig. 4e, we find that this performance improvement is driven by sentences that require primarily correctness of grammar and relative type constraints, i.e., knowledge of which descriptors are associated with an entity is not necessary to perform well on these sentences. This also explains the loss drop seen in Fig. 4: once grammar and relative type constraints are learned, the model learns to use them to solve inputs that do not require knowledge of descriptive properties, **leading to a sudden improvement in both loss and accuracy**.

**Phase 3: Learning of descriptive type constraints.** During Phase 2, we find that the model's performance on descriptive type checks witnesses minimal improvement. However, as training proceeds, the model enters a third phase at whose boundary we see a sudden change of slope from a saturation region to approximately proportional growth in the performance of descriptive type checks with log-amount of data/iterations (i.e., sublinear growth). With a slight delay, we see a similar effect kicks in for the unscrambling and condition generation tasks as well, which start to show approximately linear improvement with log-amount of data/iterations. Zooming in at this point (see inset plots in Fig. 4), we see there is in fact a small, but nevertheless noticeable, loss drop in the unscrambling and condition generation tasks. We emphasize that since the model has seen merely an order of $10^4$ iterations up to this point, if we assume the model can perfectly learn in only a few observations that some entity and a property can be seen together in a sentence, then our experimental setting can on average see only up to $15\%$ performance (which matches the observed performance) and $20\%$ at best (see App. E.0.1; the argument is that only a subset of pairs is shown during training, restricting maximum performance). However, as the model enters and progresses through the third phase, it shows a much larger rate of improvement and reaches $\sim 30\text{--}35\%$ performance, indicating it is generalizing beyond the pairs of entities and properties it has seen together during training. If the model were simply relying on memorized knowledge, the observed performance would be infeasible.

**We thus claim that the model is implicitly inferring, based on the regularity of type constraints that underlies our data, which properties and entities constitute a valid context.** This suggests a memorization effect is at play during Phase 2, and the end of this phase corresponds to a transition from a memorizing to a generalizing solution.

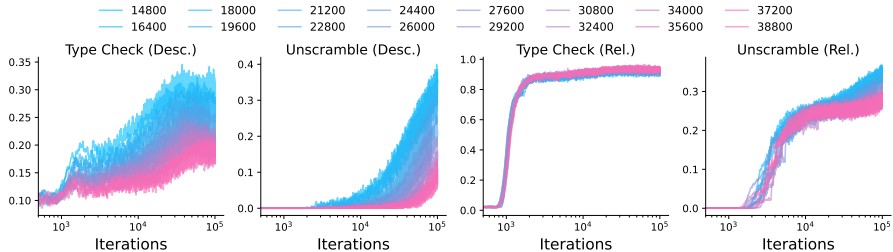

Figure 5: **Effect of Scaling number of descriptive properties.** Scaling descriptive properties in our language, we find relative type checks and unscrambling performance for relative sentences are essentially unaffected by the number of properties. Meanwhile, both descriptive type checks and unscrambling performance for descriptive sentences show a change in performance and delay in transition points. Interestingly, we find the geometry of performance curves is extremely consistent; for descriptive type checks, this geometry indicates a memorization to generalization picture.

## 5.2 Effect of Number of Descriptive Properties on Language Acquisition

Given the above picture of a model's learning dynamics, we next ask how the phase boundaries change with an increase in the number of properties $|K|$ (see Fig. 5). We intentionally scale only the number of descriptive properties and hypothesize the learning of both grammar and relative type constraints to not be affected by this change. We relegate grammar learning to appendix (see App. F.2), which, as expected, is not affected by $|K|$ since it is an entirely independent data-regularity from type constraints. However, we see that even relative type constraints' learning is not affected by the increase in descriptive properties, indicating the model deems them (justifiably) to be independent regularities. Focusing on descriptive type constraints then, we see performance curves for descriptive type checks and unscrambling performance on descriptive sentences are indeed affected, achieving higher values for fewer properties (i.e., the easier task). We further make two more interesting observations. (i) *The point of transition from memorization to generalization is delayed as we increase the number of properties.* This is most prominently seen in the delay in the transition point where the ability to unscramble descriptive sentences emerges. (ii) *We find the geometry of these performance curves are extremely similar to the geometry we observed for our base setting studied in Sec. 5.1*, indicating despite the increase in difficulty of the task, the same learning dynamics are at play. We devote the next section to formulate a hypothesis justifying these observations.

## 6 Emergence as Percolation

We next propose a framework for modeling the emergence of capabilities that require a model to compose unseen entities and descriptive properties, e.g., learning descriptive type constraints, which, beyond allowing a model to produce accurate free generations, will aid with narrower tasks like conditional generation and unscrambling. We argue the relevant data-regularity to analyze for this purpose is the *concept class*: if a model understands what entities and properties belong to a concept class, regardless of whether they have been seen together in a string, it will deem their co-occurrence valid. We thus develop an abstraction for concept classes as bipartite graphs, casting their learning as a problem of percolation on such graphs (see also App. D.1).

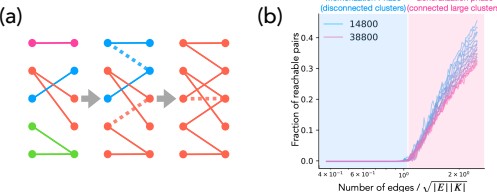

Figure 6: **Casting the ability to compose unseen entities and properties as percolation on a bipartite graph.** (a) When only a fraction of the concept classes are included in the dataset and the edge density is low, nodes (e.g., entities and properties) form many disconnected clusters, indicated by different colors (left). As more concept classes are added (dashed edges) to the bipartite graph, the small clusters begin to merge (middle). With a sufficient number of edges, a macroscopic number of nodes become connected, forming a single cluster (right). (b) Our formalism establishes this transition as a second-order phase transition, where the size of the largest cluster increases non-linearly as the fraction of connected node pairs is scaled. Shown curves are from simulations on bipartite graphs with the number of nodes equal to the number of entities and properties in our formal language experiments (see Sec. 4 or App. E).

### 6.1 Matrix representation of data

Recall that a concept class is defined as a set of entities that are expected to have shared properties (see Def. 3). The question is whether upon subsampling pairs of entities and properties from a concept class, can the model learn that, in fact, all

pairs of entities and properties are valid and compose the concept class. For instance, in the case of a concept class such as `human`, the set of entities can include humans with different genders (e.g., `man`) as well as human-associated entities such as a `lawyer` (see Fig. 2). The corresponding properties for the `human` concept class will be, for example, `walk, jump, tall`. A `man`, being `human`, is expected to have all these properties, although strings specifying these properties for a `lawyer` may be rare or even absent in the training data. We are interested in the case where the data, such as strings, includes examples of these pairs of entities and properties. We can represent this by a matrix whose rows and columns represent the entities and the properties, and the matrix values indicate the quantity or density of data available for each composition, e.g., an *entity-descriptor* pairing.

**Definition 4. *(Concept Density Matrix.)*** *Let $D$ be an $|\texttt{E}| \times |\texttt{K}|$ matrix with real-valued entries between 0 and 1, inclusive. Each entry $D_{ek}$ represents the density for the entity and property pair $(e, k)$ (e.g., the amount of data that represents the specific composition), where $e \in \{1, ..., |\texttt{E}|\}$ and $k \in \{1, ..., |\texttt{K}|\}$ are the indices of the entities and properties, respectively.*

For example, consider the case where there are three values of entities and properties ($|\texttt{E}| = |\texttt{K}| = 3$), with entities (rows) being {`Man`, `Lawyer`, `Telephone`}, and properties (columns) being {`Walk`, `Stoic`, `Ring`}. The corresponding $D$ can be: A common composition such as `Man walking` will lead to a value of 1 at the intersection of `Man` and `Walk`, i.e., $D_{00} = 1$, where $D_{ij}$ denotes element at row-$i$ and column-$j$. Conversely, a highly unlikely composition like `Lawyer ringing` will be absent in the dataset, and will be represented by a zero at the respective matrix position, i.e., $D_{12} = 0$. We can also assume for example that `Man ringing` or a `Telephone walking` are rare, which yields $D_{13} = D_{31} = 0$. We next introduce the **concept propagation matrix** to model the inference of novel entity-feature combinations from the incomplete data represented in $D$.

**Definition 5. *Concept Propagation Matrix.*** *An $n$-th order concept propagation matrix ($n \geq 0$) is defined as $T^{(n)} = (DD^T)^n D = C^n D$, where $C := DD^T$.*

The concept propagation matrix can be intuitively understood using a bipartite graph, as shown in Figure 6a. A bipartite graph in this case is a sub-graph of the **type constraints graph** (see Def. 3 and Fig. 2), where one set of nodes represents entities while the other represents properties, and edges indicate the presence of entity-feature pairings in the training data. The strength of connectivity of the graph directly corresponds to the values in the concept composition propagation matrix, $T^{(n)}$. Specifically, if two concepts are connected by a path of minimal length $2k + 1$ (i.e., the shortest path between them alternates between the two sets $k$ times), the corresponding entry in $T^{(n)}$ becomes non-zero only for $n \geq k$. That is, the number of propagation steps $n$ required for the object and feature pair to be associated is determined by the minimal number of hops needed to connect the two nodes in the graph. Conversely, if two nodes belong to disconnected regions of the graph, their composition remains fundamentally unlearnable, regardless of the order of propagation. This is reflected by the corresponding entry in $T^{(n)}$ remaining zero for all $n$. In the bipartite graph, this amounts to having two distinct clusters that are connected within themselves but not across each other. For example, in the case where the concepts represented by the first and third rows belong to disconnected regions of the graph, and consequently, their composition (e.g., `Lawyer ringing`) cannot be achieved even after an infinite number of hops between nodes. We call such a situation *learning of a concept class*: the system understands that `Man` and `Laywer` are both humans, whereas `Telephone` is not. Our experiments show that the model deems sentences composing entities and properties from incorrect classes to be much less likely than the correct ones (see Fig. F.3), i.e., the model indeed learns the structure of concept classes.

## 6.2 Percolation Transition on Descriptive Constraints

Using the bipartite graph framework, the generalization, or the learning of the concept class, can be defined as the situation where a large cluster of entity-property connected pairs arises despite the sparse concept density matrix. A critical aspect to examine is the proportion of the inference matrix values where $T_{ek}^{(\infty)}$ is non-zero, out of the total possible pairs $|\texttt{E}| \times |\texttt{K}|$. This particular scenario aligns with the bond percolation problem on a bipartite graph. In bond percolation, we investigate how the largest connected cluster's size varies with the probability $p$ of each edge (bond) being present. In a typical setting, there exists a critical threshold value, $p = p_c$, called the percolation threshold. Below this threshold ($p < p_c$), the graph typically exhibits a disconnected phase characterized by the absence of extensively connected clusters, with most nodes either isolated or part of smaller clusters. Above this threshold ($p > p_c$), the graph transitions to a connected phase, significantly increasing

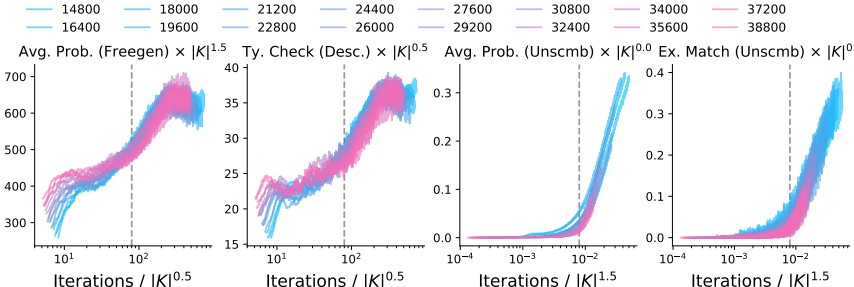

Figure 7: **Scaling of point of emergence matches with our theory.** We replot the result from Fig. 5 by rescaling the x-axis with a power of the number of descriptive properties; see App. G.1 for further discussion on the intuition underlying this visualization. For Average probability of generating a valid descriptive property given some object in context under free generation and descriptive type check accuracy, we see a 0.5 exponent scaling yields a collapse of the transition point—this matches the toy model of percolation posited in Sec. 6.2. We also see a very clear scaling of the point where the ability to unscramble descriptive sentences starts to emerge, but with an exponent of 1.5.

the likelihood of a vast connected component spanning a large portion of the graph. This shift from a predominantly disconnected state to one with a macroscopic cluster is a defining characteristic of the percolation process, and this transition sharpens as the number of components in the system increases. (see Fig. 6b for a schematic). In a setting where connecting edges are selected randomly on the graph with probability $p$, the percolation threshold is obtained as $p_c \simeq \sqrt{1/|\mathrm{E}||\mathrm{K}|}$ for large $|\mathrm{E}|$ and $|\mathrm{K}|$ (see App. D.2 for a derivation). This means that when around $\sqrt{|\mathrm{E}||\mathrm{K}|}$ edges are connected, there is a qualitative change in the growth of the cluster size. For $p > p_c$, the number of nodes included in the connected cluster becomes macroscopic, i.e., the probability a randomly selected pair of entity and property are connected becomes finite. We posit that the percolation threshold corresponds to the point at which our Transformer generalizes from the sparse learning of pairs to a complete representation of concept classes. Since increasing the iterations through online learning should amount to increasing $p$ (i.e., seeing more combinations in data), the iteration at which the transition occurs in the model performance should be proportional to $\sqrt{|\mathrm{E}||\mathrm{K}|}$. When the number of seen pairs surpasses this threshold, the model can infer novel compositions, even for entity-property pairs not explicitly present in the training data.

We next check whether the theoretically posited scaling manifests in our experiments. Specifically, we plot again the various performances of the model as a function of iterations divided by the square root of the number of properties (i.e., number of entities is kept constant). See Fig. 7 for results. *We find that indeed there is a growth trend in the descriptive type check metric and the average probability scores of free generation of descriptive sentences that occur at iterations proportional to* $\sqrt{|\mathrm{K}|}$. This is in contrast to the first large growth that is observed in these evaluations, which seems to be occurring at iteration numbers that do not depend on $|\mathrm{K}|$ (see Fig. 5 and Fig. 53), likely because this step corresponds to the point where the model learns about syntax, which requires a constant amount of data irrespective of the number of properties (see Fig. 18). We also analyzed the scaling of the transition point for the narrower task of unscrambling upon change in the number of properties. Here, we found a different scaling of $|\mathrm{K}|^{3/2}$. While we leave explaining this scaling for future work, since the task involves the composition of entities and properties, we expect a transition and scaling effect to occur for it as well; however, learning the precise circuit to perform unscrambling over the learned grammar and type constraints will yield a delay (as was seen in experiments) that likely has some interaction with the complexity of the language.

## 7 CONCLUSION

In this work, we propose a phenomenological definition for emergence that implicates learning of general regularities underlying the data process as the source of rapid performance improvements on narrower tasks. We then use a formal language that involves two precisely defined regularities—grammar and type constraints—and a set of narrowly defined tasks on its strings. We define "order parameters" for these regularities and find (i) precise phases in the learning dynamics of a Transformer trained on the language and (ii) the model suddenly acquiring capabilities corresponding to the narrower scope tasks close to said phases' boundaries. To explain these results, we propose a model that analogizes learning of type constraints to the problem of graph percolation, finding a strong qualitative match with this hypothesis.

ACKNOWLEDGMENTS

The authors thank Intelligent Systems group at Harvard, especially Maya Okawa, Core Francisco Park, and Leni Shor for feedback on the project and contributions to predecessors of this work. ESL thanks Gautam Reddy, Naomi Saphra, Pulkit Gopalani, Wei Hu, and Yonatav Belinkov for fruitful conversations. ESL's time at University of Michigan was supported by NSF under award CNS-2211509 and at Harvard by the CBS-NTT Physics of Intelligence program. KK acknowledges support from JSPS KAKENHI Grant Numbers JP19H05795, JP19H05275, JP21H01007, and JP23H00095.

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

## A    FURTHER DISCUSSION AND FUTURE WORK

In this work, we take inspiration from other fields (e.g., physics and complex systems) and propose a phenomenological definition for emergence of capabilities in neural networks. Specifically, the definition argues that at the point of emergence, the model learns general regularities underlying the data-generating process which are instrumental to the learning of specific, narrower capabilities; acquisition of such regularities then leads to sudden performance improvement on several tasks (often with some delay). While relatively informal, this definition brings the notion of emergence in the context of neural networks closer to its meaning in physics, wherein the formation of systematic regularities is known to drive *phase changes* that involve sudden changes in the system's properties. Characterizing these phase changes requires hypothesizing what the regularity is, and defining an "order parameter" that can help gauge its change. Drawing on this definition and perspective, we then propose an experimental setup that involves learning of a formal language with two precisely defined regularities in the data—grammar and type constraints (what properties are valid in the context of what entities)—and a set of narrowly defined tasks. Defining order parameters for these regularities (grammaticality and type checks), we find there indeed are phases in the model's learning dynamics, and the model suddenly acquires capabilities corresponding to the narrower scope tasks (unscrambling and conditional generation) close to these phase boundaries. Interestingly, the learning curves show a rather distinct geometry that remains consistent as we alter the number of properties in our language.

To explain these results, we propose a model that analogizes learning of type constraints in the formal language learning task to the problem of graph percolation. Drawing on the theory of percolation on bipartite graphs, which shows phase transitions in the formation of connected components on a graph, we argue this problem is similar to learning of concepts classes or type constraints in our setting, and hence should show scaling of the point of emergence where the model starts to follow descriptive type constraints that is of the order of $\sqrt{|E||K|}$. Our results show a strong qualitative match with this hypothesis. We also find an extremely clean scaling for other tasks' transition point, e.g., for unscrambling's results on descriptive sentences; explaining these results is left for future work.

While our goal in this work was primarily demonstrative, i.e., to develop a bridge with other fields studying emergence, we believe several exciting avenues now open up. For example, given that the whole point of the theory of phase transitions is that we can predict the point of emergence, can we draw on this rich literature to propose models for explaining and predicting emergent capabilities in neural networks? Can we go beyond the toy task of formal language learning studied in this work and analyze a more naturalistic setting? For example, can we identify data-regularities underlying emergent capabilities in open-source models, e.g., Pythia checkpoints (Biderman et al., 2023), and demonstrate that our proposed perspective enables prediction of when capabilities emerge in LLMs? To begin, we can perhaps focus on capabilities that require similar knowledge acquisition and its compositional generalization on a downstream task.

## B    RELATED WORK

**Explaining emergence.** Focusing on the sudden learning characteristic of emergent capabilities, a few recent works have tried to explain the factors driving this phenomenon. For example, compositionality has been implicated for having a "multiplicative" effect on a model's performance, where the argument is that a model cannot perform well on a compositional task until the abilities needed to perform individual tasks involved in that composition are acquired (Okawa et al., 2023; Arora & Goyal, 2023; Yu et al., 2023; Srivastava et al., 2022; Wei et al., 2022; Hoffmann et al., 2022; Gokhale, 2023); when they are acquired, performance suddenly grows. A few papers have also shown that learning of specific capabilities (i.e., ones not compositional in nature) can be sudden (Chen et al., 2024; Nam et al., 2024; Kirsch et al., 2022; He et al., 2024; Michaud et al., 2023; Cui et al., 2024).

**Grokking vs. Emergence.** We focus on the effect of data scaling on a model's capabilities; often called 'learning curve' or 'data scaling' analysis (Viering & Loog, 2022; Blumer et al., 1989; Bousquet et al., 2021; Seung et al., 1992; Watkin et al., 1993; Amari, 1993; Haussler et al., 1994). On surface, this might look similar to the seemingly related phenomenon of grokking (Power et al., 2022; Liu et al., 2023b; Žunkovič & Ilievski, 2022; Murty et al., 2023; Barak et al., 2022; Edelman et al., 2023; Nanda et al., 2022), wherein a model's performance on a task rapidly improves long after it has fit the

training data. However, we emphasize that we focus on an *online learning* setting in our experiments, i.e., a given sample is unlikely to be seen multiple times during training. Emergence is generally studied in such online learning scenarios. Since there is no distinction between train versus test data in such a setting, we argue mechanistic explanations of grokking identified in past work that involve a perfect training data memorization phase (Nanda et al., 2023; Liu et al., 2022b) are unlikely to help explain our results of emergence under data scaling in an online learning scenario.

### B.1    IS EMERGENCE A MIRAGE, EXPLAINED VIA USE OF POOR METRICS?

Schaeffer et al. (2023) argue emergent scaling curves are a consequence of poorly defined, discontinuous evaluation metrics, and the seemingly sudden learning goes away once partial, continuous credit is given to the model. *While a credible claim, we provide three arguments below to demonstrate it is certainly not the complete story.*

- **Emergence with continuous metrics.** There is significant evidence for emergent abilities in neural networks that does not rely on use of discrete metrics at all. For example, in a recent work, Gopalani et al. (2024) show a sudden performance increase in a regression setting with an *entirely continuous metric*—specifically, mean square error. *Phase transitions* and corresponding emergent abilities have been formally discussed in other relevant works, e.g., by Lu et al. (2024) in regards to in-context learning and by Cui et al. (2024) in regards to positional code learning in a histogram computation task. Meanwhile, Chen et al. (2024) show a *sudden loss drop* in BERT training, which, again, is not a continuous metric. Similarly, Okawa et al. (2023) have shown task compositionality can drive emergent abilities, with their follow-up work using *continuous metrics* to demonstrate learning to compose yields emergent learning curves (Park et al., 2024). Overall, given this substantial evidence, we believe the argument by Schaeffer et al. (2023) is narrower in scope than what is claimed by the authors—it undermines and ignores legitimate cases of emergence in neural network training.
    - **Relation to our work.** In our work, we show performance improvements co-occur with loss drops (Fig. 4), i.e., a continuous metric. Thus, we can comfortably say our results are not confounded by Schaeffer et al.'s argument that poorly defined metrics drive emergence. In fact, as we mention in Sec. 4, we also report several other (both continuous and discrete) metrics that show sudden improvements, further providing evidence that our results are not confounded by use of poorly defined metrics.
- **Assumption on per-token power law scaling.** To develop their claim, Schaeffer et al. (2023) propose a toy theoretical model that involves the *assumption that an individual token's loss follows power-law scaling*. **This assumption is too strong**: the loss, when averaged across a large number of tokens, indeed follows a power law (as popularly shown in literature on scaling laws); however, **individual token dynamics are in general drastically different**, and in general extremely unlikely to follow a power law scaling. In fact, this point was most recently demonstrated by Schaeffer et al. (2024) themselves! In this follow-up work, the authors show that learning dynamics of individual tokens corresponding to task output locations can show discontinuous progress. For further evidence in this vein, see the papers by Michaud et al. (2023) and Du et al. (2024). We thus believe that beyond the scope of their argument being narrower than original presented in the paper, the underlying rationale behind Schaeffer et al.'s argument also involves a very strong assumption.
    - **Relation to our work.** In our work, we can concretely show Schaeffer et al.'s assumption does not hold: e.g., in Figure 4, we show loss curves for individual tasks, finding that learning dynamics of tokens corresponding to just these individual tasks (a subset of the overall tokens in a sentence) does not follow a power law! Thus, we can again comfortably conclude that Schaffer et al.'s assumption, which was too strong to begin with, does not hold in our setting and hence their claims cannot explain our results.
- **A "metric" must at least gauge progress towards learning of task.** To demonstrate their claim empirically, Schaeffer et al. (2023) propose several alternative metrics that turn a discontinuous learning curve into a continuous one. However, we emphasize a metric is only useful if it captures progress towards learning of the task. For example, if the structure of a task is ignored, it is certainly easy to define arbitrary continuous metrics for a task; however, such metrics are unlikely to help measure progress toward learning a task. As a precise example, consider the addition of

two numbers, say 10 and 11, and the metric called *token edit distance* (Schaeffer et al., 2023) that assesses the average distance between digits in the model's output, denoted xy, from the ground truth of 21; that is, $(|\mathtt{x}-2|+|\mathtt{y}-1|)/2$. For both $\mathtt{xy} = 22$ and $\mathtt{xy} = 11$, this metric equals 1; however, clearly 22 is a better approximation for the ground truth of 21. That is, once we account for the structure of the task, i.e., the fact that error in the most significant digit should be penalized more than error in the least significant one, we see limitations in token edit distance as a metric for assessing a model's ability to add numbers. We argue claims relating emergence to sensitivity of metrics can be confounded by use of metrics that do not respect the structure of the task, but give the impression of continuous learning.

– **Relation to our work.** As a stress-test of our claims, we intentionally evaluate metrics that do not respect the structure of the task; e.g., in our *unscrambling* task, we evaluate the metric of *per-token accuracy*. This metric can be deemed equivalent to the *token edit distance* metric proposed by Schaeffer et al. (2023), since it does not respect sentence order. Intriguingly, we find that even this metric demonstrates sudden improvements in our experiments. The first sudden improvement, corresponding to syntax acquisition, is easily visible; the latter sudden improvement, corresponding to learning of type constraints, is only visible when zoomed. This possibly indicates that even metrics that do not respect task structure show sudden improvements, but the improvement is relatively marginal and hence not easily visible.

## C  DATA-GENERATING PROCESS: DEFINING OUR FORMAL LANGUAGE

Our data-generating process involves defining a formal language, sampling sentences from this language, and then defining tasks to be performed upon these sentences (specifically, free generation, unscrambling, or conditional generation). In this section, we discuss the precise details of how the language is implemented.

### C.1  DEFINING A GRAMMAR USING PCFGS

To define a grammar for our language, we use the framework of Probabilistic Context-Free Grammars (PCFGs). To keep the paper self-contained, we provide a short primer on PCFGs below and then discuss our precise version of it in detail. For a more thorough discussion on PCFGs, we refer the reader to one of the several well-written tutorials (Collins, 2013) and books (Sipser, 1996).

#### C.1.1  SHORT PRIMER ON PCFGS

Broadly, a PCFG is defined via a 5-tuple $G = (\mathtt{NT}, \mathtt{T}, \mathtt{R}, \mathtt{S}, \mathtt{P})$, where:

- $\mathtt{NT}$ is a finite set of non-terminal symbols.
- $\mathtt{T}$ is a finite set of terminal symbols, disjoint from $\mathtt{NT}$.
- $\mathtt{R}$ is a finite set of production rules, each of the form $A \rightarrow \alpha\beta$, where $A \in \mathtt{NT}$ and $\alpha, \beta \in (\mathtt{NT} \cup \mathtt{T})$.
- $\mathtt{S} \in \mathtt{NT}$ is the start symbol.
- $\mathtt{P}$ is a function $\mathtt{P} : \mathtt{R} \rightarrow [0, 1]$, such that for each $A \in \mathtt{NT}$, $\sum_{\alpha:A\rightarrow\alpha\in\mathtt{R}} \mathtt{P}(A \rightarrow \alpha\beta) = 1$.

To *generate* a sentence from a PCFG, the following process is used. Pseudocode for this generation process is provided in Algo. 1.

1. Start with a string consisting of the start symbol $S$.

2. While the string contains non-terminal symbols, randomly select a non-terminal $A$ from the string. Choose a production rule $A \rightarrow \alpha\beta$ from $\mathtt{R}$ according to the probability distribution $\mathtt{P}(A \rightarrow \alpha)$.

3. Replace the chosen non-terminal $A$ in the string with $\alpha$, the right-hand side of the production rule.

4. Repeat the production rule selection and expansion steps until the string contains only terminal symbols (i.e., no non-terminals remain).

5. The resulting string, consisting entirely of terminal symbols, is a sentence sampled from the grammar.

---

**Algorithm 1: Pseudocode for generating a sentence from a PCFG:** Process to sample a sentence from a given PCFG $G$.

```
def generate_sentence(G):
    # Initialize the string with the start symbol S
    string = [S]
    # While the string contains non-terminal symbols
    while any(is_nonterminal(symbol) for symbol in string):
        # Select a non-terminal A from the string
        A = select_nonterminal(string)
        # Choose a production rule A → α according to P
        rule = sample_rule(A, G.P)
        # Replace A in the string with α
        string = apply_rule(string, A, rule)
    # Return the generated sentence composed of terminal symbols
    return string
```

---

### C.1.2 INSTANTIATING THE GRAMMAR UNDERLYING OUR LANGUAGE

While generally one directly samples sentences from a grammar, in this work, we define a grammar that operates over *symbols*, i.e., whose terminals are variables that are not yet populated by any specific values from the language's vocabulary. We emphasize this is an unconventional manner for defining a PCFG, as one would generally use a standard vocabulary of the language to directly define terminal symbols. However, to enforce type constraints, we find this unconventional format aids in making the implementation easier. Specifically, one can simply sample an entirely *symbolic sentence*, and then enforce type constraints at the step when these symbols have to be populated.

Overall, our grammar, denoted G, is defined using the following.

- **Terminal symbols:** $T = \{\texttt{Subj}, \texttt{Obj}, \texttt{Verb}, \texttt{Conj}, \texttt{lVerb}, \texttt{Desc}, \texttt{eAdj}, \texttt{dAdj}, \texttt{Adv}, \texttt{Prep}\}$.
  - Here, Subj is a symbol for a subject, Obj for an object, Verb for verbs, Conj for conjunctions, lVerb for a linking verb, Desc for descriptors, eAdj for adjectives used for entities, dAdj for adjectives used for descriptors, Adv for adverbs, and Prep for prepositions.
- **Non-terminal symbols:** $NT = \{\texttt{S}, \texttt{sNP}, \texttt{sT}, \texttt{oNP}, \texttt{oT}, \texttt{VP}, \texttt{vT}, \texttt{descT}\}$.
  - Here, S denotes the start symbol, sNP can be interpreted as a noun phrase with a subject in it, sT as the immediate ancestor of the subject symbol, oNP as a noun phrase with an object in it, oT as the immediate ancestor before the object symbol, VP as a verb phrase, vT as the immediate ancestor of the verb symbol, and descT as the immediate ancestor of a descriptor symbol.
- **Production rules** $R$:

$$
\begin{aligned}
\texttt{S} &\to \texttt{sNP VP} \; [1.0] \\
\texttt{sNP} &\to \texttt{sT} \; [0.8] \mid \texttt{sNP Conj sNP} \; [0.2] \\
\texttt{VP} &\to \texttt{lVerb descT} \; [0.4] \mid \texttt{Verb Prep oNP} \; [0.4] \mid \texttt{VP Conj VP} \; [0.2] \\
\texttt{oNP} &\to \texttt{oT} \; [0.7] \mid \texttt{oT Conj oNP} \; [0.3] \\
\texttt{sT} &\to \texttt{eAdj Subj} \; [0.8] \mid \texttt{Subj} \; [0.2] \\
\texttt{oT} &\to \texttt{eAdj Obj} \; [0.8] \mid \texttt{Obj} \; [0.2] \\
\texttt{descT} &\to \texttt{dAdj Desc} \; [0.8] \mid \texttt{Desc} \; [0.2]
\end{aligned}
$$

Note that since non-terminals can appear on both left and right hand side of a rule, there is recursion possible in our grammar and hence sentences can get very long. We restrict sentence lengths to 75, yielding a language where sentence lengths vary from 4–75 tokens. Probability over rules was partially adapted from prior work by (Hupkes et al., 2020).

Given the above, we can now sample symbolic sentences such as `Subj lVerb Desc`. We will populate these symbols with tokens from our vocabulary $\mathcal{V}$. As noted above, while in general this population step would be performed as the final step of the grammar, to enforce type constraints and enable context-sensitivity, we separate it from the grammar.

**Implementation.** To implement the grammar, we use the NLTK package (Bird et al., 2009), which provides an easy interface to define PCFGs. Moreover, the package provides pre-implemented parsers that help perform grammaticality checks, i.e., if our model produces a sentence $f(x)$ for some input $x$, we can simply use the parser to check whether $f(x)$ is grammatically correct.

## C.2    TYPE CONSTRAINTS

As described in the main paper, we instantiate a minimal notion of context sensitivity by constraining when an entity is seen in the context of a property or verb. There are two subtle ways in which such constraints will affect the generated sentences.

- **Constraining properties.** When a symbolic sentence with a descriptor is sampled, the descriptor symbol will be populated with a property that is valid for the relevant entity in the sentence.
- **Constraining subjects and objects.** Subjects and objects broadly distinguish entities (or, to be precise, nouns) in a sentence. For properties that help define verbs (e.g., `Walk`), we instantiate a notion of directionality that determines whether the entity can take the action suggested by the verb corresponding to the property or whether the action can be taken upon it. Accordingly, when a verb is selected, only a subset of subjects and objects that can take and have the action of verb be taken upon them are left valid to form a sentence.

Overall, then, say we have a symbolic sentence. We populate the symbols in the sentence as follows.

- **Check if there is a verb to populate. If so:**
    1. Randomly sample a verb from the vocabulary and fill it in.
    2. Sample required number of entities that can occur on the right side of the verb, i.e., can populate objects
    3. Sample required number of entities that can occur on the left side of the verb, i.e., can populate subjects
- **Check if there are descriptor to populate. If so:**
    1. If the entities are not populated yet, populate them.
    2. Use the parse tree for the symbolic sentence to identify property of which entity should populate the descriptor.
    3. Randomly sample a descriptor from the valid properties of said entity.
- **Check if there are adjectives to populate. If so:**
    1. Identify whether the adjective corresponds to an entity or a property
    2. Populate the adjective with a valid adjective from the group of adjectives reserved for entities versus properties
- **Check if there are adverbs, link verbs, prepositions, or conjunctions to populate. If so:**
    1. Sample an adverb, link verb, preposition, or conjunction from the vocabulary. We intentionally do not make these parts context-sensitive, since the remaining parts are sufficient to induce context-sensitivity and enable our experiments.

Pseudocode describing the process above is detailed in Algo 2.

## C.3    DEFINING THE OVERALL CONTEXT-SENSITIVE LANGUAGE

Our language $\mathcal{L}$ is defined by first instantiating the underlying grammar as described in App. C.1 and then the type constraints in App. C.2. We note that since the grammar is a randomized process and token roles are randomly filled by using the type constraints graph, the odds of seeing the same

**Algorithm 2: Pseudocode for populating a symbolic sentence:** Fills in symbols while respecting the type constraints.

```
def populate_sentence():
    # Check if there is a verb to populate
    if has_verb():
        # Sample a verb from the vocabulary and fill it in
        verb = sample_verb(vocabulary)
        populate_verb(verb)
        # Sample entities for the right side (objects)
        objects = sample_entities(right)
        # Sample entities for the left side (subjects)
        subjects = sample_entities(left)
    # Check if there are descriptors to populate
    if has_descriptors():
        # Populate entities if not already populated
        if not entities_populated():
            populate_entities()
        # Use parse tree to identify which entity's property to populate
        entity = get_entity_for_descriptor(parse_tree)
        # Sample a descriptor from the valid properties
        descriptor = sample_descriptor(valid_properties, entity)
        populate_descriptor(descriptor)
    # Check if there are adjectives to populate
    if has_adjectives():
        # Identify if the adjective corresponds to an entity or a
         property
        role = identify_role(token)
        # Sample a valid adjective based on the role
        adjective = sample_adjective(role)
        populate_adjective(adjective)
    # Check if there are adverbs, link verbs, prepositions, or
     conjunctions to populate
    if has_adverbs_etc():
        # Sample an adverb, link verb, preposition, or conjunction from
         the vocabulary
        word = sample_word(vocabulary)
        populate_word(word)
```

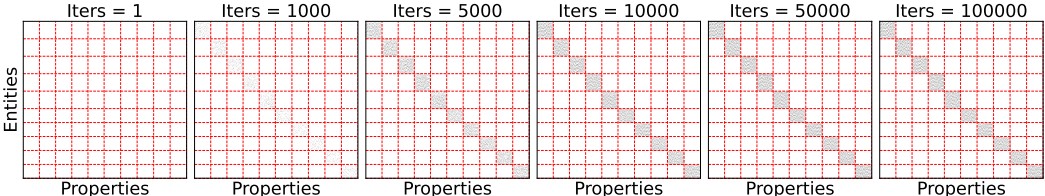

Figure 8: **Adjacency matrix with time.** As the model undergoes training, entities are seen in the context of more properties. Recording whether a pair of object and property have been seen together, we can get the adjacency matrix for a bipartite graph corresponding to entities and properties that constitute the data-generating process, as shown in the plots. The red-dotted lines indicate class boundaries. This matrix can be deemed as the adjacency matrix representation of the empirical type constraints graph, and, as training goes on, it will get closer to the ground truth graph. Note that several entities and properties will never be seen together, however, other entities from the class may be paired with a property, allowing the model a signal to infer that the entities likely belong to the same class.

sample multiple times are exceedingly low. Primary hyperparameters for defining $\mathcal{L}$ include number of entities and number of properties, denoted $|\texttt{E}|$ and $|\texttt{K}|$, respectively. Unless mentioned explicitly, we fix these hyperparameters to 900 and 18000 respectively. In several experiments we do vary these variables though. Thus, we also note that we are slightly abusing notations here and using $\mathcal{L}$ to refer to a single language. In actuality, however, what we have is a family of languages with the same grammar, but varying number of entities and properties. The vocabulary consists of entities (subjects and objects), descriptors, verbs, adjectives, adverbs, prepositions, and conjunctions. All languages we analyze have the same number of verbs ($= 200$), linking verbs ($= 2$), adjectives ($= 20$), adverbs ($= 20$), prepositions ($= 3$), and conjunctions ($= 2$).

We also note that the type constraints graph merely describes which properties are valid for a class. For a specific entity, only a fraction of these entities might be visible during training. Specifically, we constrain the sampling process such that only 15% of valid properties of a class are actually associated with an entity. However, as training occurs, the model gets to see several entities in the context of several properties. Even though certain pairs will never be seen together due to the restriction discussed above, two randomly sampled entities will still have a non-zero proportion of properties in whose context they have both been seen, hence giving the model some signal that the entities have shared characteristics (see Fig. 8). This is likely what leads to the percolation-like process we observe in the main paper to come into play, and hence yields us a $0.5$ power law scaling for the transition point where model's performance on generating sentences with descriptors or performing reasoning tasks on sentences with descriptors starts to improve.

A few example sentences from the language are reported in Figs 9, 10. Note that there are a large number of symbolic sentences possible; we merely report two of these to provide intuition. We also reemphasize that naturalistic sentences used as examples in the main paper were to merely analogize the structures our language is trying to capture. It is not difficult to see that the sentences in the examples provided here have a similar structure and constraints as those naturalistic examples.

**Task:** free generation
**Input:** null
**Output:** eAdj19 subj5 has pAdj7 descriptor1496

**Task:** unscrambling
**Input:** has eAdj19 descriptor1496 pAdj7 subj5
**Output:** eAdj19 subj5 has pAdj7 descriptor1496

**Task:** conditional generation
**Input:** subj5 descriptor1496
**Output:** eAdj19 subj5 has pAdj7 descriptor1496

Figure 9: **Exemplars where an entity's properties are described.** Each box represents a different task; specifically, free generation, unscrambling, and conditional generation.

**Task:** free generation
**Input:** null
**Output:** eAdj16 subj102 adv9 verb64 in eAdj1 obj41

**Task:** unscrambling
**Input:** eAdj16 subj102 adv9 verb64 in eAdj1 obj41
**Output:** eAdj16 subj102 adv9 verb64 in eAdj1 obj41

**Task:** conditional generation
**Input:** verb64 obj41
**Output:** eAdj16 subj102 adv9 verb64 in eAdj1 obj41

Figure 10: **Exemplars where a subject and object are bound via a verb.** Each box represents a different task; specifically, free generation, unscrambling, and conditional generation.

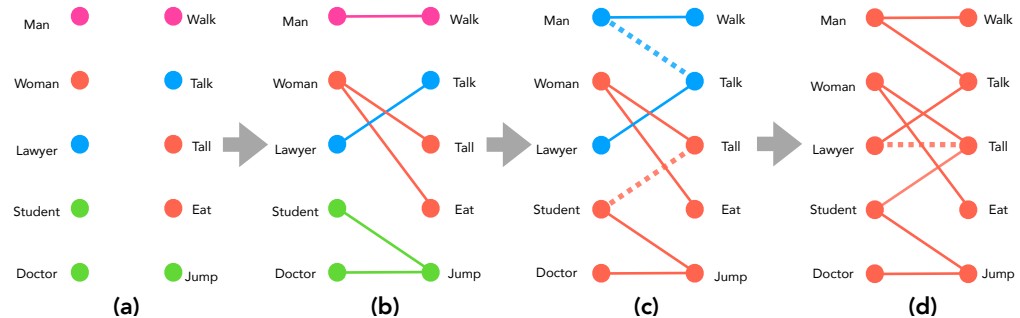

Figure 11: **Viewing learning of descriptive type constraints as the problem of percolation on a bipartite graph.** Imagine entities and properties of a class represented as nodes of a bipartite graph. As training proceeds (marked using large gray arrows in the figure), edges are added to this graph to denote the model has seen sentences containing nodes connected by the edge, i.e., an entity and a property. (a) At initialization, there will be no edges in the graph. (b) However, with time, edges begin to get added (bold, colored edges). (c) With time, the model will see the same properties being used for the different entities (dotted, colored edges). For ex., it may see that both a `Woman` and a `Student` can `Eat`. This leads to a path of edges that connect entities and properties which have not been seen by the model in a shared context (e.g., the model may be able to infer in panel (c) that a `Woman` can `Jump`). (d) If enough properties are shared between different entities, the model can infer that in fact all such entities belong to the same class and hence can be seen in the context of any property that is valid for the class. This will yield the learning of the broader concept class structure, as indicated by evolution of color of nodes and edges: from left to right, color is increasingly red, indicating there is a path of edges that connects all entities and properties to each other. We analogize this overall process to execution of the process of percolation on a bipartite graph: as edges are added, we can define the existence of a valid path between two nodes to suggest that there is enough information available to believe these nodes are related to each other; with enough edges, there is a sudden transition that leads to the emergence of a large connected component in the graph, indicating enough information is available to learn the structure of the class. This can enable generalization to unseen combinations of entities and properties.

## D    LEARNING OF DESCRIPTIVE TYPE CONSTRAINTS AS PERCOLATION ON A BIPARTITE GRAPH

We analogize learning of type constraints as a function of time to the addition of edges to a bipartite graph whose nodes denote entities and properties; an edge denotes a given combination of entity and property has been seen by the model during training. Consequently, as time goes on, more edges are added to this graph, yielding a percolation phase transition once a critical edge density necessary for achieving the transition is met. This transition leads to the emergence of a large connected component. We claim this percolation process captures the dynamics of our model learning type constraints. Below, we first discuss the analogy between percolation and learning of type constraints in some detail (App. D.1), and then derive the scaling of the critical edge density needed for the percolation phase transition (App. D.2). Latter forms the basis of our experiments in Sec. 6.2.

### D.1    MORE DISCUSSION OF THE ANALOGY

Below, we discuss steps involved in the reasoning that help analogize learning of type constraints with the problem of percolation phase transition on a bipartite graph. See Fig. 11 for a visualization.

**Step 1: Defining the type constraints graph.** Imagine a bipartite graph whose nodes on one side are entities of a class (e.g., `Man`, `Woman`, `Lawyer`, `Doctor`, etc. forming the class of `Humans`); meanwhile, the other set of nodes is the properties these entities can have (e.g., `Walk`, `Talk`, `Eat`, `Jump`, etc. for class of `Humans`). We call this graph the type constraints graph (see also Figure 2b). When sampling a sentence from our language to train our models, we use the type constraints graph to constrain which entities and properties can be seen together in a sentence. While all entity-property

combinations of a class form valid sentences, given the combinatorially many combinations, at a given point of time, the model is unlikely to have seen (and hence learned) all possible combinations. We can however test whether the model "knows" that an unseen combination of entities and properties can form a valid sentence: if it does, the model has learned to combine an unseen entity and property belonging to the same class.

**Step 2: Analogizing what the model knows to percolation on a bipartite graph.** At initialization, the model does not know that an entity of a class can possess properties of that class. For example, the model does not know that the entity `Man` can have the property of being `Tall`. As the model undergoes training, it will see sentences connecting entities to properties valid for them; e.g., seeing the sentence "That man is very tall" can lead to the model learning that the entity `Man` can be `Tall`. We define the concept density matrix (Definition 4) to capture this dynamic at any given time. The matrix entry corresponding to a pair of entities and properties is assigned the value of 1 at a given time $t$ if the model has seen this pair of entity and property in a sentence. This matrix can be deemed as the adjacency matrix of a bipartite graph whose nodes, again, are entities and properties of a class, and whose edges denote whether the model has the knowledge that an entity–property combination can produce a valid sentence. As the model undergoes training, more edges are added to this graph (see Figure 6, left panel). We argue this is reminiscent of the problem of bond percolation on a bipartite graph, wherein edges are added to the graph one-by-one, and, at some critical edge density, there is a phase transition that yields a large connected component covering a large proportion of the graph. That is, there is the sudden emergence of a path between any two randomly selected nodes of the graph. Beyond the point of transition, the rate of growth of this connected component is approximately log-linear in edges added.

**Step 3: Percolation as a model for predicting unseen combinations of entity–property pairs can be composed to form a valid sentence.** If the percolation analogy is correct, we expect the transition point in our empirical results where the model starts to learn descriptive constraints in a log-linear manner with respect to time to scale as a power-law of exponent 0.5 in the number of node combinations of the graph (i.e., $\propto \sqrt{|E||K|}$). As results in Figure 7 show, this is indeed the case!

### D.2 PERCOLATION THRESHOLD IN THE BIPARTITE GRAPH SETUP

For general bipartite graphs that are uncorrelated, meaning that they are completely described by the degree distributions $P_1(k)$ and $P_2(k)$ for the entities and properties, respectively, the percolation threshold is

$$p_c = \sqrt{\frac{\langle k \rangle_1 \langle k \rangle_2}{\langle k(k-1) \rangle_1 \langle k(k-1) \rangle_2}}. \tag{1}$$

Here, $\langle \cdot \rangle_i$ denotes the expected value with respect to $P_i(k)$, and we require $|E|\langle k \rangle_1 = |K|\langle k \rangle_2$ for consistency. The case of randomly selecting connecting edges as demonstrated in the main text will correspond to starting from a complete bipartite graph, in which case $P_1(k) = \mathbb{1}_{k,|K|}$ and $P_2(k) = \mathbb{1}_{k,|E|}$, leading to $p_c = \sqrt{1/(|E|-1)(|K|-1)} \simeq \sqrt{1/|E||K|}$.

To derive Eq. (1) we use the generating function as explained in (Newman et al., 2001). Firstly, we introduce the generating function for the degree distribution of two concepts, $i = 1, 2$:

$$G_i^0(x) \quad := \quad \sum_{k=0}^{\infty} P_i(k)x^k \tag{2}$$

The generating function can be used to calculate moments of the probability distribution, such as the mean and variance, by taking derivatives:

$$\left.\frac{dG_i^0(x)}{dx}\right|_{x=1} = \left.\sum_{k=0}^{\infty} kP_i(k)x^{k-1}\right|_{x=1} = \sum_{k=0}^{\infty} kP_i(k) = \langle k \rangle_i \tag{3}$$

$$\left.\frac{d^2G_i^0(x)}{dx^2}\right|_{x=1} = \left.\sum_{k=0}^{\infty} k(k-1)P_i(k)x^{k-2}\right|_{x=1} = \sum_{k=0}^{\infty} k(k-1)P_i(k) = \langle k(k-1) \rangle_i. \tag{4}$$

Here we denoted the average over the degree distribution of concept $i$ as $\langle \cdot \rangle_i$.

Another useful property of generating functions is that the generating function of the sum of the degrees can be described by the power of generating functions. For example, the distribution of the sum of degrees from two randomly selected nodes from sets $i$ and $j$, denoted by $\tilde{P}_{ij}(k)$, will satisfy

$$\sum_{k=0}^{\infty} \tilde{P}_{ij}(k)x^k = G_i^0(x)G_j^0(x). \tag{5}$$

With these properties in mind, we further introduce the generating function for the distribution of outgoing edges from a node that we arrive at by following a randomly chosen edge:

$$G_i^1(x) \quad := \quad \frac{\sum_{k=0}^{\infty} kP_i(k)x^{k-1}}{\sum_{k=0}^{\infty} kP_i(k)} = \frac{G_i^{0'}(x)}{\langle k \rangle_i}. \tag{6}$$

which can be obtained by noticing that the probability of the degree of a node arrived at from a randomly chosen edge is proportional to $kP_i(k)$. The decreased power of $x$ by one in the numerator is to exclude the originally chosen edge.

We further introduce the generating function for the distribution of the number of concepts in $i$ that can be reached from a node in the concept $j(\neq i)$ that is connected to a randomly chosen edge as $\tilde{G}_i^1(x)$, and the same when randomly choosing a node in concept $j$, $\tilde{G}_i^0(x)$. These functions satisfy

$$\tilde{G}_i^0(x) \quad = \quad G_i^0(G_j^1(x)) \tag{7}$$

$$\tilde{G}_i^1(x) \quad = \quad G_i^1(G_j^1(x)). \tag{8}$$

We also introduce the generating function for the distribution of the sizes of components in concept $i$ that are reached by choosing an edge, $H_i^1(x)$, and the same when choosing a node in concept $i$, $H_i^0(x)$. These satisfy

$$H_i^0(x) \quad = \quad x\tilde{G}_i^0(H_j^1(x)) \tag{9}$$

$$H_i^1(x) \quad = \quad x\tilde{G}_i^1(H_j^1(x)). \tag{10}$$

Here, the key assumption is that there is no closed loop of edges in the network, which holds if the fraction of connection is low and there is no cluster (i.e., sub-critical regime).

The average cluster size of concept $i$, i.e., the number of nodes in $i$ that are connected with each other, is then $\langle S_i \rangle = H_i^{0'}(1)$, which is

$$\langle S_i \rangle = 1 + \frac{\tilde{G}_i^{0'}(1)}{1 - \tilde{G}_i^{1'}(1)}, \tag{11}$$

using the derivatives of Eqs. (9),10). The percolation threshold is when the denominator in the second term of Eq. (11) becomes zero, so

$$\tilde{G}_i^{1'}(1) = G_i^{1'}(1)G_j^{1'}(1) = \frac{G_1^{0''}(1)G_2^{0''}(1)}{\langle k \rangle_1 \langle k \rangle_2} = \frac{\langle k(k-1) \rangle_1 \langle k(k-1) \rangle_2}{\langle k \rangle_1 \langle k \rangle_2} = 1 \tag{12}$$

Now, when the connection of each a probability of connection $p$ associated with each bond on top of the original graph, the generating function of the degrees will become

$$G_i^0(x;p) \quad = \quad \sum_{k=0}^{\infty} \sum_{n=k}^{\infty} P_i(n) \binom{n}{k} p^k (1-p)^{n-k} x^k \tag{13}$$

$$= \quad \sum_{n=0}^{\infty} P_i(n)(px + 1 - p)^n = G_i^0(1 + (x-1)p). \tag{14}$$

From the first line to the second line, we used $\sum_{k=0}^{\infty} \sum_{n=k}^{\infty} = \sum_{n=0}^{\infty} \sum_{k=0}^{n}$. We can then rewrite Eq. (12) as

$$\frac{\langle k(k-1) \rangle_1^p \langle k(k-1) \rangle_2^p}{\langle k \rangle_1^p \langle k \rangle_2^p} = p^2 \frac{\langle k(k-1) \rangle_1 \langle k(k-1) \rangle_2}{\langle k \rangle_1 \langle k \rangle_2} = 1, \tag{15}$$

from which we obtain Eq. (1). Here we used $\langle k(k-1) \rangle_i^p = G_i^{0''}(1;p) = p^2 G_i^{0''}(1) = p^2 \langle k(k-1) \rangle_i$ and $\langle k \rangle_i^p = G_i^{0'}(1;p) = pG_i^{0'}(1) = p\langle k \rangle_i$.

### D.2.1 EXPONENT IN THE CLUSTER SIZE

The critical exponent associated with the number of nodes in the cluster for $p > p_c$, $S \sim (p - p_c)^\beta$, is determined to be $\beta = 1$ when there is no specific structure in the graph. To see this, let us consider that $u = H_i^0(1)$ is the probability that a node in $i$ is included in a finite size cluster (i.e., not the large connected cluster). Recall that $H_i^0(1)$ was the generating function of the number of nodes in concept $i$ included in the cluster in the subcritical regime ($p < p_c$); we are here assuming that the statistics will not change even in the supercritical regime ($p > p_c$) when neglecting the large cluster. Then, from Eqs. (9,10), we have

$$H_i^0(1) = u = \tilde{G}_i^0(H_j^1(1)) = \tilde{G}_i^0(\tilde{G}_j^1(u)) \tag{16}$$

$$= G_i^0(G_j^1(G_j^0(G_i^1(u)))) =: f(u) \tag{17}$$

which is a self-consistent equation.

By writing $u = 1 - \epsilon$, we have $f(1 - \epsilon, p) = 1 - \epsilon f'(1, p) + \epsilon^2 f''(1, p)/2...$, where the derivative is taken for $u$. Noticing that $f'(1, p_c) = 1$, we obtain the relation

$$\epsilon = (p - p_c) \left.\frac{\partial^2}{\partial u \partial p} f(u, p)\right|_{u=1, p=p_c} \left[\frac{1}{2} \left.\frac{\partial^3}{\partial u^2 \partial p} f(u, p)\right|_{u=1, p=p_c}\right] + o(p - p_c) + o(\epsilon) \tag{18}$$

$$\sim (p - p_c), \tag{19}$$

indicating $\beta = 1$.

As an interesting generalization, a classic result (Cohen et al., 2002) shows that even for the situation where $p_c > 0$, the power $\beta$ can deviate from one. This corresponds to when the differential coefficients in Eq. (18) diverge, corresponding to cases where the second or third moment being ill-defined. For the case of $P_i(k) \sim k^{-\gamma}$ with $3 < \gamma < 4$, we can show that $\beta = 1/(\gamma - 3)$.

### D.2.2 TRANSITION BEHAVIOR FOR FINITE INFERENCE STEPS

The mapping of the inference scheme to the percolation problem becomes precise only in the context of infinite inference steps. For a finite number of steps, denoted as $n$, the pertinent question is the number of node pairs across the sets connected within $2n + 1$ edges. Using the average degrees $\langle k \rangle_1$ and $\langle k \rangle_2$ respectively, a node in the first set can reach approximately $\langle k \rangle_1^{n+1} \langle k \rangle_2^n$ nodes after $2n + 1$ steps. Hence, the approximate fraction of connected edges within $2n + 1$ steps is $|E| \langle k \rangle_1^{n+1} \langle k \rangle_2^n$.

# E  EXPERIMENTAL DETAILS

**Model architecture.** We train a two-block Transformer based on the nanoGPT architecture (Andrej Karpathy, 2023) using the standard autoregressive language modeling objective, i.e., next token prediction. Each block contains two attention heads, an MLP, GELU activation, and processes / produces 128 dimension representations. Both token and position embeddings are learned during training.

**Optimization setting.** Models are trained using the Adam optimizer with $10^{-3}$ stepsize, batch-size of 128, and $10^{-4}$ weight decay for $10^5$ iterations (or until the run collapses due to cluster challenges; e.g., power outages). Gradient clipping at norm of 1 is applied. No learning rate schedule is used. Unless stated otherwise, results are averaged over three seeds.

**Data configuration.** Sentences are sampled "online", i.e., we sample a fresh batch of data every iteration by following the rules of the language. The language has $M$ entities and $N$ properties uniformly distributed over $C$ classes, with edges connecting properties sparsely and randomly distributed over valid properties for a given object (specifically, only 15% connections are made). We slightly abuse notations by using $\mathcal{L}$ to refer to our language, since in actuality we have a family of languages with the same grammar, but varying number of entities and properties. We note that since the grammar is a randomized process and token roles are randomly filled by using the type constraints graph, the odds of seeing the same sample multiple times are exceedingly low.

**Tokenization.** We use a one-hot, manually defined tokenization scheme wherein each token is associated with a unique token ID.

### E.0.1  PERFORMANCE OF A MEMORIZING SOLUTION ON DESCRIPTIVE SENTENCES

As the model undergoes training, its accuracy at getting descriptive constraints right can, at max, be the following: $\texttt{Acc} = 0.1 + f * \max\left(1, \frac{0.25Bt|C|}{|\mathsf{E}||\mathsf{K}|Rf}\right)$, where $f$ is fraction of pairs from the type constraints graph the model can see during training, $0.25$ is approximately the proportion of randomly sampled sentences that are descriptive in nature, $B$ is batch-size, $t$ is number of iterations, and $R$ is number of repetitions needed to internalize that an entity and property constitute a valid context. Since we see the third phase in a regime where $t \sim 10^4$, assuming at least 4 repetitions are necessary for internalizing a pair, we have $\texttt{Acc} \sim 0.15$.

## F    FURTHER RESULTS: ROBUSTNESS ACROSS SETTINGS AND EVALUATIONS

In this section, we report several more metrics relevant to assess how well the model has internalized the language and how well it is able to perform tasks on top of strings from the language. We report results across several configurations as well. Rarely, but certainly sometimes, runs crashed due to cluster issues. These configurations are not reported, or reported until the point of crash if sufficient time had passed in training.

- Base setting: This is the setting used throughout the paper, i.e., with 10 classes and 900 entities.
- Varying number of properties: ranging from 14800—38800, in increments of 1600.
- Different class setting: we change number of classes to 2 and repeat all evaluations in this setting.
- Different entities setting: we change number of entities to 1800 and repeat all evaluations in this setting.

We specifically report the following results. Both in the main paper and in the results below, evaluation metrics are averaged over 1000 randomly sampled strings.

- Loss / learning curves under different settings: App. F.1.
- Grammaticality and type checks under different settings: App. F.2.
- Negative log likelihoods of sentences from the langauge and their perturbed versions (e.g., where type constraints are not correct): App. F.3.
- How well does the model follow our language, where we analyze NLLs of sentences generated by the model, distribution of length, and parse tree depth: App. F.4.
- Evolution of Attention maps: App. F.5.
- Further results on unscrambling: App. F.6.
- Further results on Conditional Generation: App. F.7. Conditional generation evaluations turn out to be extremely time-expensive, with a single run taking approximately 4 days to finish when conditional generation is evaluated (compared to 12 hours without). This is likely a result of model generating extremely long sentences to compose conditioning tokens that can involve multiple subjects, objects, and properties. We thus primarily focus on free generation and unscrambling in the results reported in this section. We do provide results for conditional generation in one more setting with 27600 properties to demonstrate that our findings from the main paper (i.e., in the 18000 properties setting) generalize.

## F.1 LEARNING CURVES

We plot learning curves for different settings in this section. Results are reported for varying number of properties, averaged over 3 seeds, and results for the base setting used in the main paper where number of properties is fixed to be $18000$. For the latter setting, we show the average run alongside individual runs.

### F.1.1 BASE SETTING WITH VARYING NUMBER OF PROPERTIES

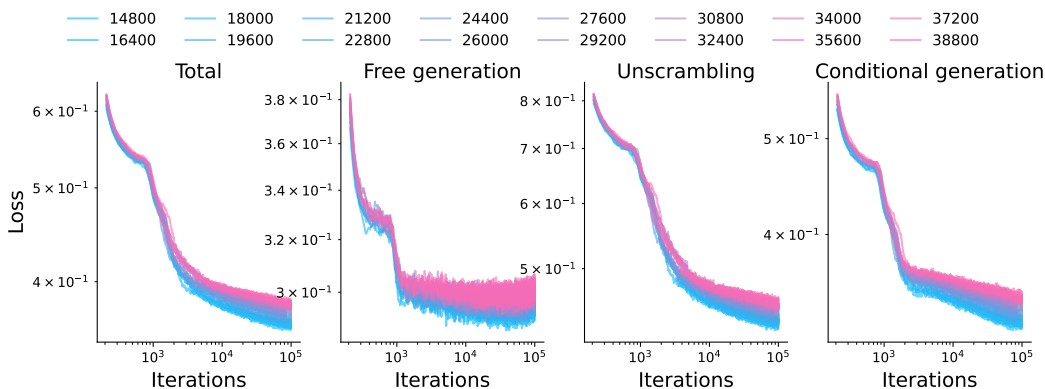

Figure 12: **Learning curves with varying number of properties (base setting).** As number of properties are varied, we see the overall loss is substantially more continuous, but individual tasks can see sudden learning. For example, in free generation, we see a sudden loss drop. This point is precisely when the model learns to produce grammatically correct sentences. Very slightly after this point, both tasks of unscrambling and conditional generation see improvement as well. There is a second change of slope for these tasks between $10^3$ to $7 \times 10^3$ iterations, depending on the number of properties. These points match match the moment where the model starts to improve in its Type Check performance.

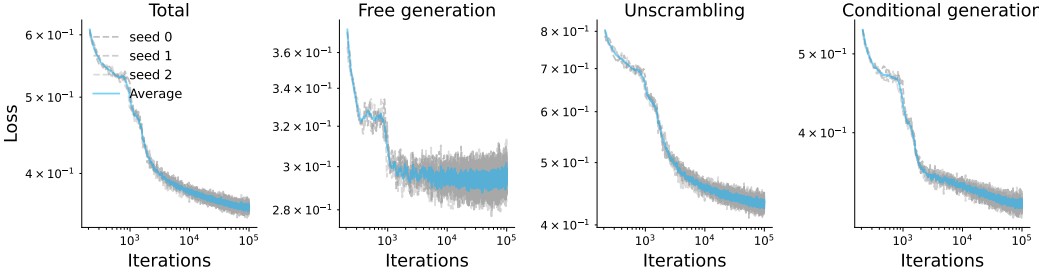

Figure 13: **Learning curves for setting with $18000$ properties (base setting).** We report this plot to zoom into a specific configuration. As can be seen, there is some minimal variance across runs, but mostly points of transition are in a similar range.

### F.1.2 CHANGING TO 1800 ENTITIES AND VARYING NUMBER OF PROPERTIES

We change the number of entities to $1800$ (compared to base setting of $900$) and report learning curves under varying number of properties. See Figs. 14, 15.

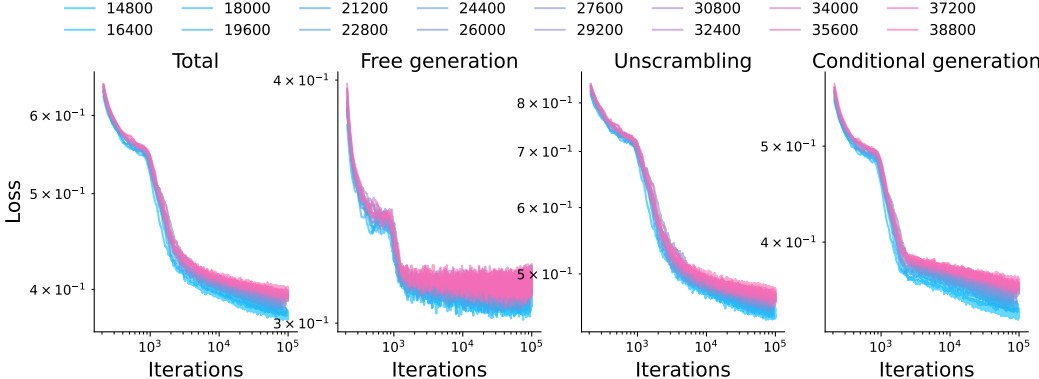

Figure 14: **Learning curves with varying number of properties (1800 entities).** As number of properties are varied, we see the overall loss is substantially more continuous, but individual tasks can see sudden learning. For example, in free generation, we see a sudden loss drop. This point is precisely when the model learns to produce grammatically correct sentences. Very slightly after this point, both tasks of unscrambling and conditional generation see improvement as well. There is a second change of slope for these tasks between $10^3$ to $7 \times 10^3$ iterations, depending on the number of properties. These points match match the moment where the model starts to improve in its Type Check performance.

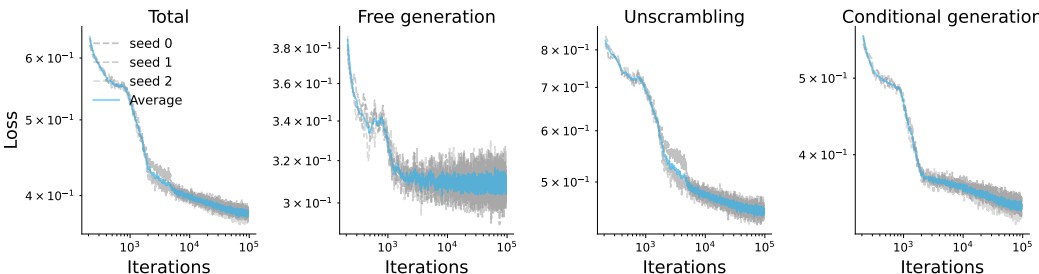

Figure 15: **Learning curves for setting with** 18000 **properties (1800 entities).** We report this plot to zoom into a specific configuration. As can be seen, there is some minimal variance across runs, but mostly points of transition are in a similar range.

### F.1.3 CHANGING TO 2 CLASSES AND VARYING NUMBER OF PROPERTIES

We change the number of classes to divide entities and properties over to 2 (compared to base setting of 10) and report learning curves under varying number of properties. See Figs. 16, 17.

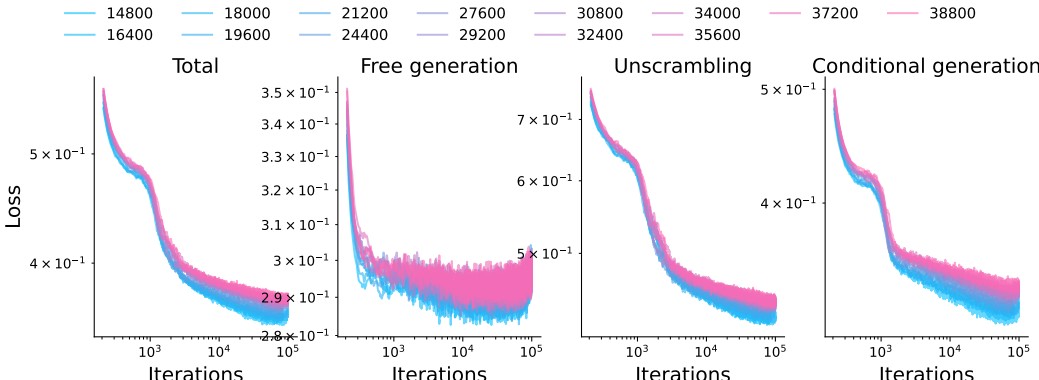

Figure 16: **Learning curves with varying number of properties (2 classes).** As number of properties are varied, we see the overall loss is substantially more continuous, but individual tasks can see sudden learning. For example, in free generation, we see a sudden loss drop. This point is precisely when the model learns to produce grammatically correct sentences. Very slightly after this point, both tasks of unscrambling and conditional generation see improvement as well. There is a second change of slope for these tasks between $10^3$ to $7 \times 10^3$ iterations, depending on the number of properties. These points match match the moment where the model starts to improve in its Type Check performance.

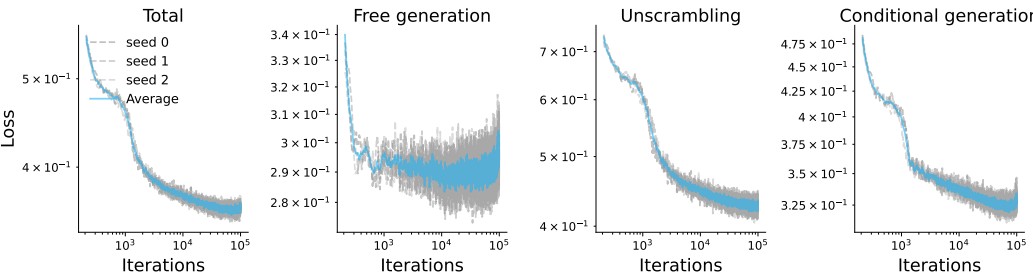

Figure 17: **Learning curves for setting with** $18000$ **properties (2 classes).** We report this plot to zoom into a specific configuration. As can be seen, there is some minimal variance across runs, but mostly points of transition are in a similar range.

## F.2 GRAMMATICALITY AND TYPE CHECKS

As the model learns rules of our language, we can track how grammatical its sentences are and whether they satisfy type constraints, as done in the main paper. We report similar results for different settings in this section. Specifically, we report results for varying number of properties, averaged over 3 seeds, and for the base setting used in the main paper where number of classes is fixed to be 18000. For the latter setting, we show the average run alongside individual runs.

For **grammaticality**, we merely use the NLTK parser to check whether the generated sentences by the model under free generation are grammatically valid, i.e., they follow the rules of the grammar.

For **Type Checks**, we extract the subjects, objects, and any properties in the sentence to checks whether they are valid under the type constraints graph (see Def. 3).

### F.2.1 BASE SETTING WITH VARYING NUMBER OF PROPERTIES

We report results under varying number of properties with the base experimental setting. See Figs. 18, 19.

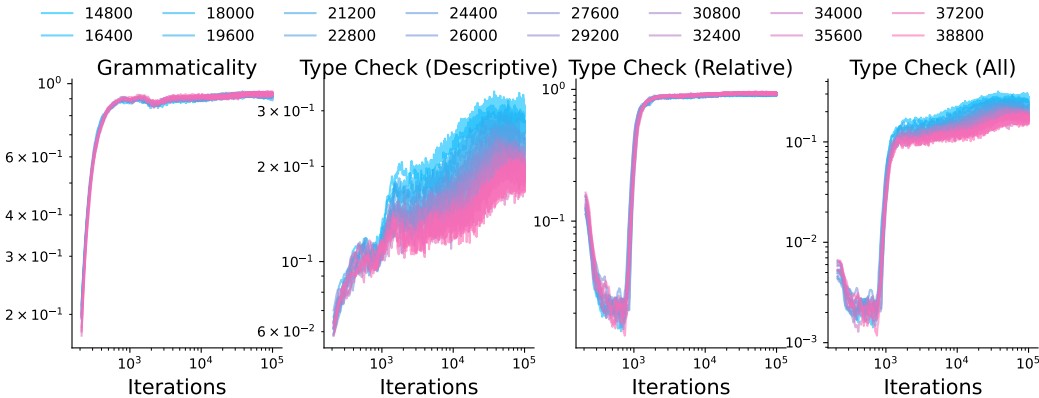

Figure 18: **Grammaticality and Type Checks with varying number of properties (base setting).** As number of properties are varied, we see the grammar is learned around broadly the same time, i.e., grammar learning is invariant to number of properties (as also seen in learning curves). For type check, we see as soon as grammaticality reaches its maximum, relative constraints quickly improve, leading to a boost in accuracy of all constraints evaluation (rightmost panel). Descriptive constraints see a transition at this point as well, but then after a period of saturation (akin to a saddle point), start to improve at an approximately linear rate (on log-log scale) until saturating again. These results match the base setting shown in paper.

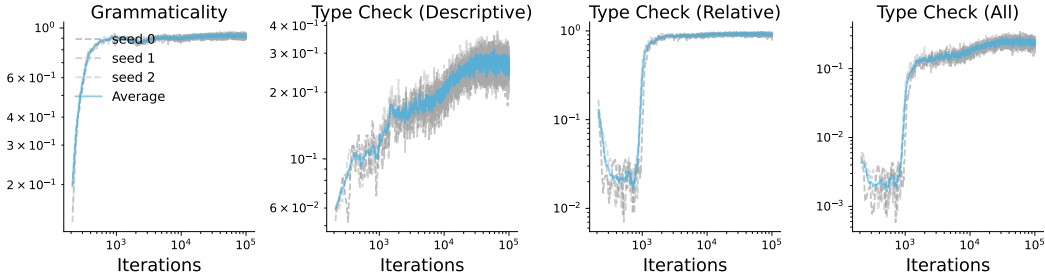

Figure 19: **Grammaticality and Type Checks with 18000 properties (base setting).** We report this plot to zoom into a specific configuration, in particular the one used in main paper. As can be seen, there is some minimal variance across runs, but mostly points of transition are in a similar range.

### F.2.2 CHANGING TO 1800 ENTITIES AND VARYING NUMBER OF PROPERTIES

We change the number of entities to 1800 (compared to base setting of 900) and report results under varying number of properties. See Figs. 20, 21.

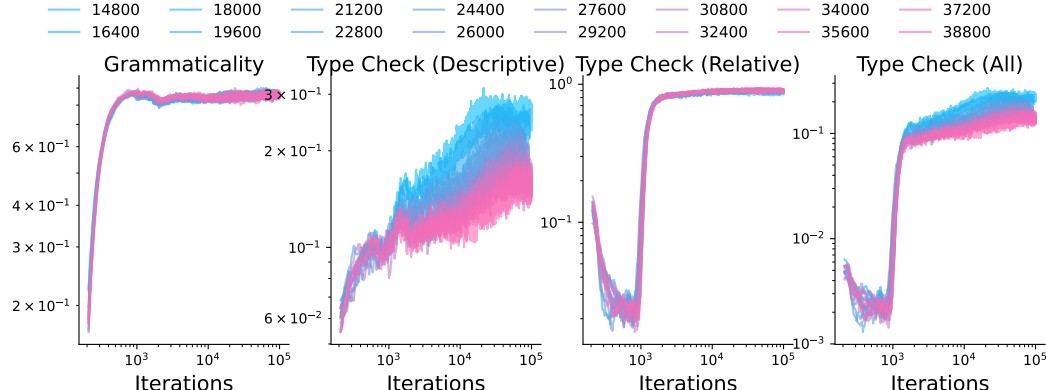

Figure 20: **Grammaticality and Type Checks with varying number of properties (1800 entities).** To demonstrate robustness of results, we increase number of entities in this experiment. We again find that as number of properties are varied, the grammar is learned around broadly the same time, i.e., grammar learning is invariant to number of properties (as also seen in learning curves). For type check, we see as soon as grammaticality reaches its maximum, relative constraints quickly improve, leading to a boost in accuracy of all constraints evaluation (rightmost panel). Descriptive constraints see a transition at this point as well, but then after a period of saturation (akin to a saddle point), start to improve at an approximately linear rate (on log-log scale) until saturating again. These results match the base setting shown in paper.

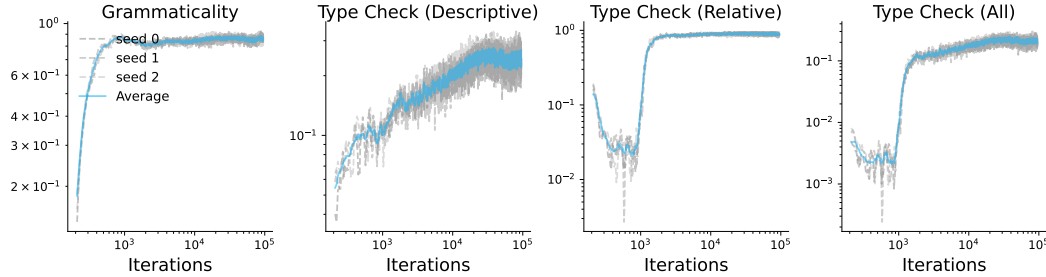

Figure 21: **Grammaticality and Type Checks with 18000 properties (1800 entities).** We report this plot to zoom into a specific configuration. As can be seen, there is some minimal variance across runs, but mostly points of transition are in a similar range.

### F.2.3 CHANGING TO 2 CLASSES AND VARYING NUMBER OF PROPERTIES

We change the number of classes to divide entities and properties over to 2 (compared to base setting of 10) and report results under varying number of properties. See Figs. 22, 23.

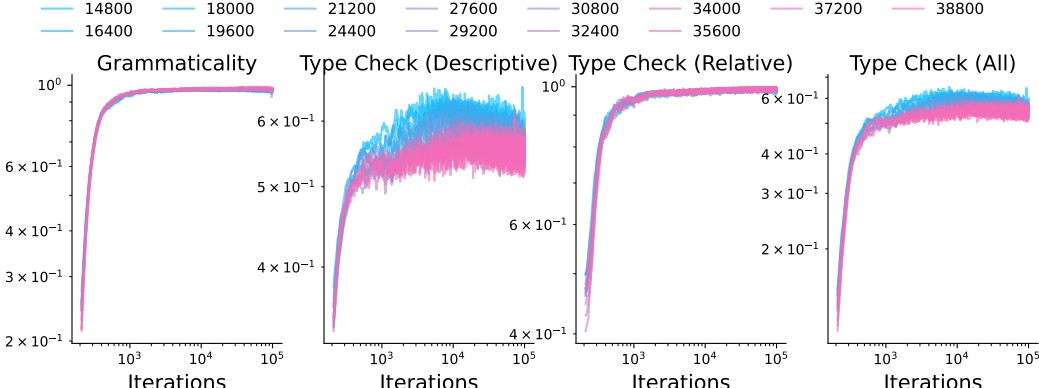

Figure 22: **Grammaticality and Type Checks with varying number of properties (2 classes).** To demonstrate robustness of results, we increase fewer classes in this experiment. We again find that as number of properties are varied, the grammar is learned around broadly the same time, i.e., grammar learning is invariant to number of properties (as also seen in learning curves). For type check, we see as soon as grammaticality reaches its maximum, relative constraints quickly improve, leading to a boost in accuracy of all constraints evaluation (rightmost panel). Descriptive constraints see a transition at this point as well, but then after a period of saturation (akin to a saddle point), start to improve at an approximately linear rate (on log-log scale) until saturating again. The results are less prominent for this setting, but zooming in (see figure below) shows the claims do follows the base setting shown in paper.

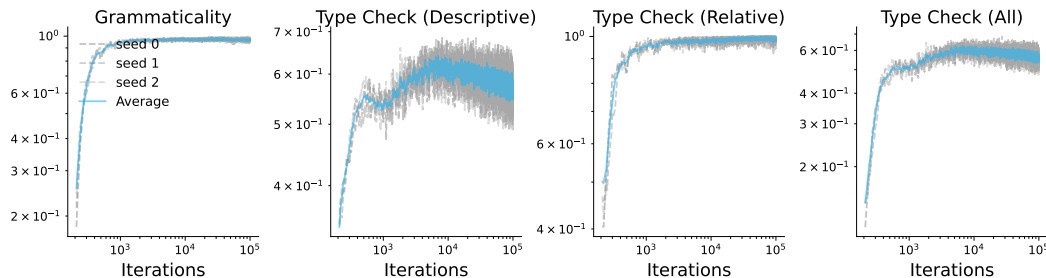

Figure 23: **Grammaticality and Type Checks with 18000 properties (2 classes).** We report this plot to zoom into a specific configuration. As can be seen, there is some minimal variance across runs, but mostly points of transition are in a similar range.

### F.3 Negative Log Likelihood of Sentences from Language and their Perturbed Versions

In this section, we report negative log-likelihoods (NLL) assigned by the model to randomly sampled sentences from the language during the course of training. To check how well the model is learning the language, and not perhaps overfitting to some specific samples (we note this is unlikely to occur in online learning, so this evaluation is just a sanity check but not crucial). To this end, we evaluate the model assigned NLLs for following settings.

- **Seen.** This is essentially the training distribution. We define valid sentences from the language by using the part of the type constraints graph that has connections between entities and properties, and evaluate the model NLLs.

- **Uniform.** Since only a fraction of valid connections are shown to the model during training, it is not necessary for it to generalize to other valid connections. To assess whether the model can make such inferences, in this evaluation, we allow any valid connection between entities and properties to be uniformly sampled. This is also the primary evaluation setting for most experiments conducted in this work.

- **Randomize values.** Arguably, the model can overly generalize and even start deeming sentences that do not satisfy the type constraints to be valid. To assess this, in this evaluation, we ensure the sentence remains grammatically correct, but intentionally use entities and properties that yield a sentence that does not follow type constraints.

- **Randomize grammar.** We simply sample a sentence and permute it to break the grammatical rules, while, technically speaking, preserving type constraints since tokens seen in the sentence are allowed to be in the same context.

Results are reported for varying number of properties, averaged over 3 seeds, and for the base setting used in the main paper where number of classes is fixed to be 18000. For the latter setting, we show the average run alongside individual runs. Broadly, our results show the following process is underway as the model undergoes training.

1. First the model learns the grammar. At this point, Seen, Uniform, and Randomize Values all see improved NLL. *This is what we expect.* Since a grammatically correct sentence does not have to satisfy type constraints, in this first phase, the model is bound to show improved NLL for sentences that respect vs. do not respect type constraints.

2. Then, there is a sudden improvement in NLL for both Seen and Uniform evaluations. *At precisely this point, the Randomize values evaluation hugely degrades.* This implies that once the model learns type constraints, it does not deem likely sentences that do not respect them.

3. For the most part, the model never deems grammatically incorrect sentences to be likely. However, there is a sudden, large degradation in NLLs for grammatically incorrect sentences later in training.

### F.3.1 BASE SETTING WITH VARYING NUMBER OF PROPERTIES

We report results under varying number of properties with the base experimental setting. See Figs. 24, 25.

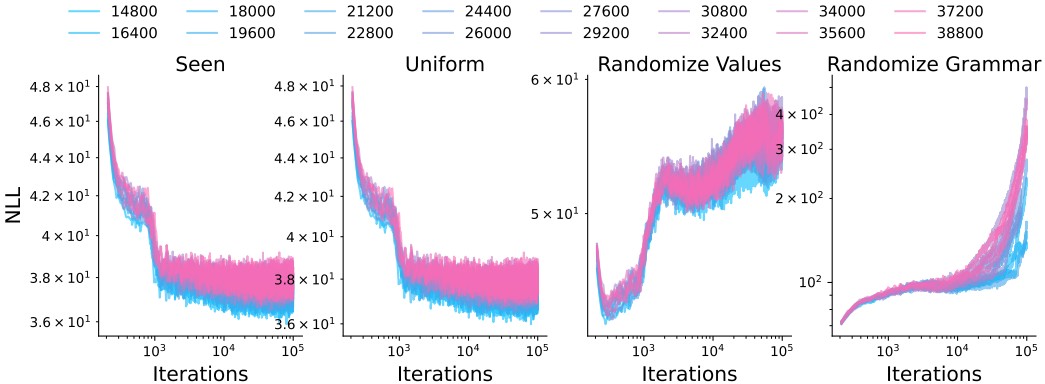

Figure 24: **NLL of sentences with different perturbations and varying number of properties (base setting).** See text in App. F.3 for a detailed discussion. Broadly, we see NLL of Seen evaluation is very slightly better than Uniform.

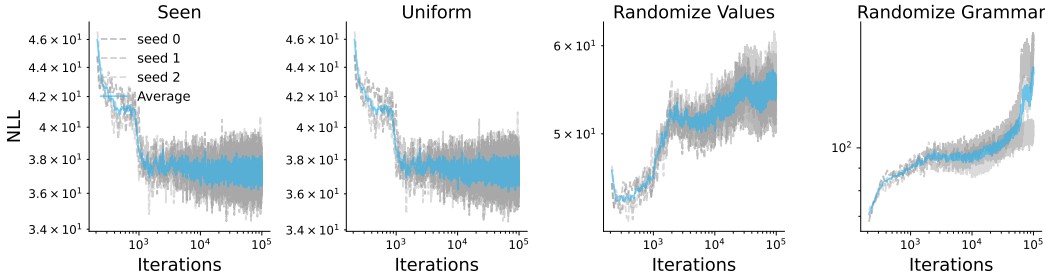

Figure 25: **NLL of sentences with different perturbations for setting with** 18000 **properties (base setting).** We report this plot to zoom into a specific configuration. As can be seen, there is some minimal variance across runs, but mostly points of transition are in a similar range.

### F.3.2 CHANGING TO 1800 ENTITIES AND VARYING NUMBER OF PROPERTIES

We change the number of entities to 1800 (compared to base setting of 900) and report results under varying number of properties. See Figs. 26, 27.

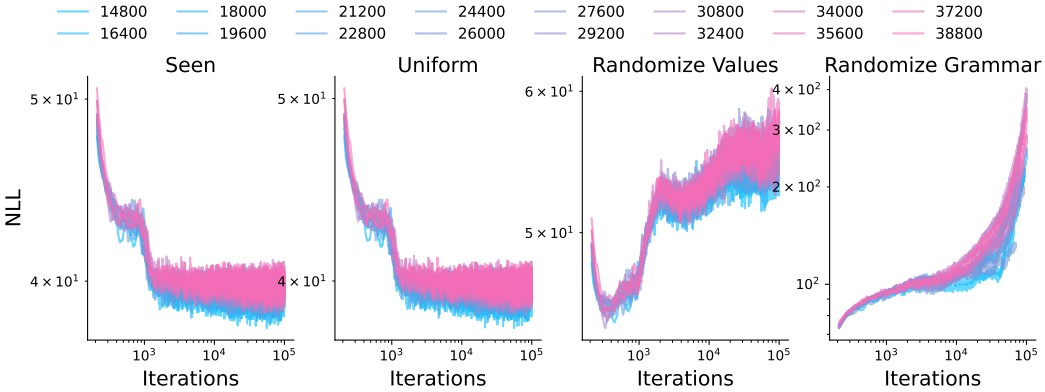

Figure 26: **NLL of sentences with different perturbations and varying number of properties (1800 entities).** See text in App. F.3 for a detailed discussion. Broadly, these plots show results with fewer classes show similar behavior as the base setting.

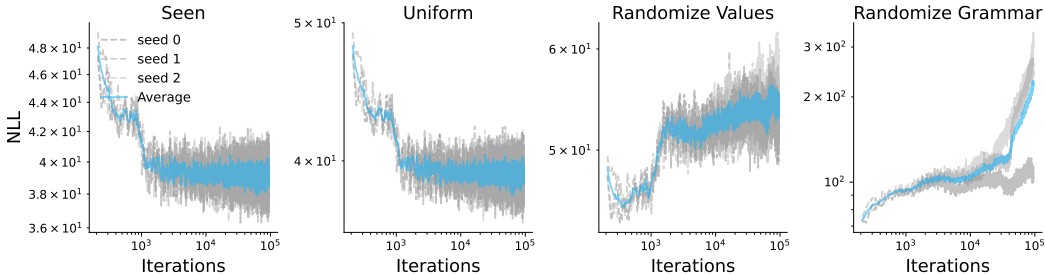

Figure 27: **NLL of sentences with different perturbations for setting with** 18000 **properties (1800 entities).** We report this plot to zoom into a specific configuration. As can be seen, there is some minimal variance across runs, but mostly points of transition are in a similar range.

### F.3.3 Changing to 2 classes and varying number of properties

We change the number of classes to divide entities and properties over to 2 (compared to base setting of 10) and report results under varying number of properties. See Figs. 28, 29.

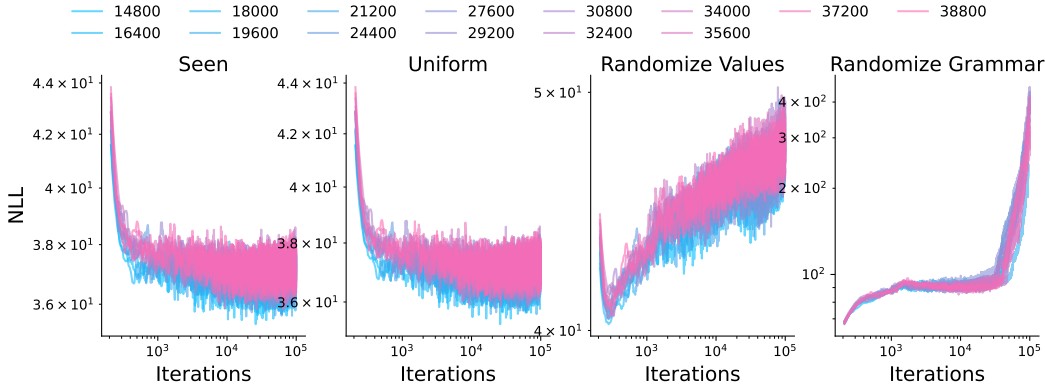

Figure 28: **NLL of sentences with different perturbations and varying number of properties (2 classes).** See text in App. F.3 for a detailed discussion. Broadly, these plots show results with fewer classes show similar behavior as the base setting.

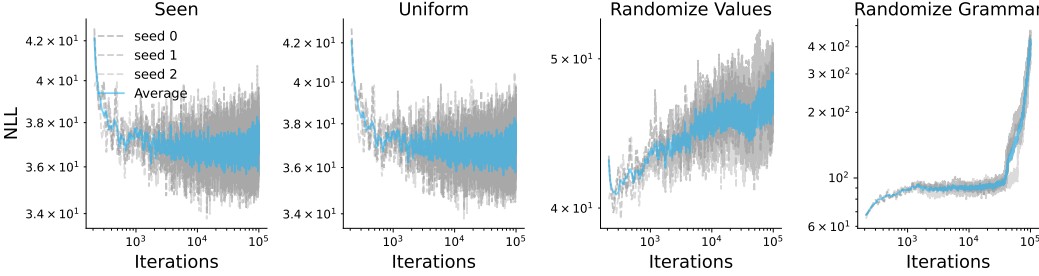

Figure 29: **NLL of sentences with different perturbations for setting with** 18000 **properties (2 classes).** We report this plot to zoom into a specific configuration. As can be seen, there is some minimal variance across runs, but mostly points of transition are in a similar range.

## F.4 HOW WELL DOES THE MODEL FOLLOW THE GRAMMAR: DISTRIBUTION OF NLLS, DEPTHS, AND LENGTHS

In this section, we analyze how well the model learns our language. Specifically, we let model produce a sentence and then use the data-generating process (PCFG and our type constraints graph) to analyze the max, min, and mean values of metrics listed below. Note that in the following, we restrict evaluations to grammatically valid sentences only, i.e., only model generations that are grammatically valid are used for this evaluation (else the NLL will be infinity). Since the model produces grammatically sentences ≈90–95% of the time (see App. F.2), this conditioning leads to filtering of *only a very minimal* number of generations.

- **NLL.** We analyze how likely the *sentences generated by the model* are under the *data-generating process*.
    - Note that if sentences from the grammar were used itself for this evaluation, min, max, and mean values over a batch of 1000 sentences turn out to be 1.0, 7.63, and 68.2 for the base setting; evaluations of model's generations are within these ranges as well.
- **Parse Tree Depth.** We use the NLTK parser to compute the parse tree underlying our model's generated sentences and the tree's depth.
    - Note that if sentences from the grammar were used itself for this evaluation, min, max, and mean values over a batch of 1000 sentences turns out to be 3, 4.78, and 15; evaluations of model's generations are within these ranges as well.
- **Lengths.** We compute the number of tokens in model's generated sentences.
    - Note that if sentences from the grammar were used itself for this evaluation, min, max, and mean values over a batch of 1000 sentences turns out to be 4, 10, and 107; evaluations of model's generations are within these ranges as well.

### F.4.1 BASE SETTING WITH VARYING NUMBER OF PROPERTIES

We report results under varying number of properties with the base experimental setting. See Fig. 30.

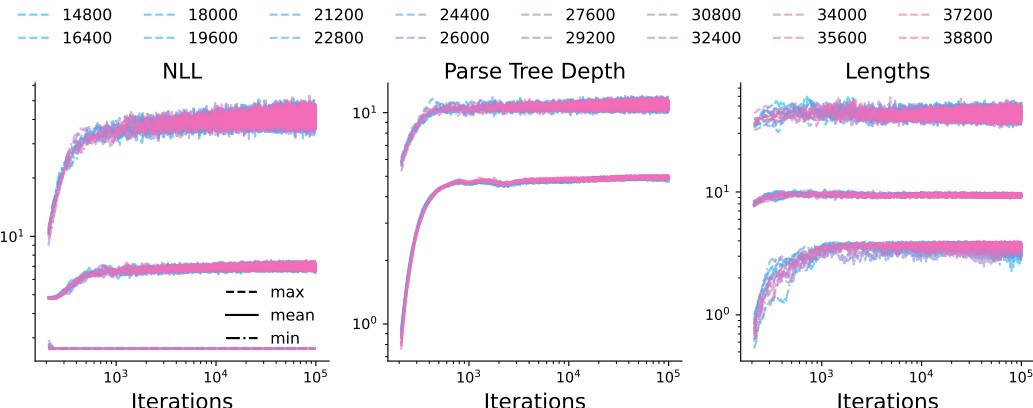

Figure 30: **Grammar characteristics of model generation as number of properties are varied (base setting).** See App. F.4 for a detailed discussion. Broadly, model generated sentences are in a similar range as our language's.

### F.4.2 CHANGING TO 1800 ENTITIES AND VARYING NUMBER OF PROPERTIES

We change the number of entities to 1800 (compared to base setting of 900) and report results under varying number of properties. See Fig. 31.

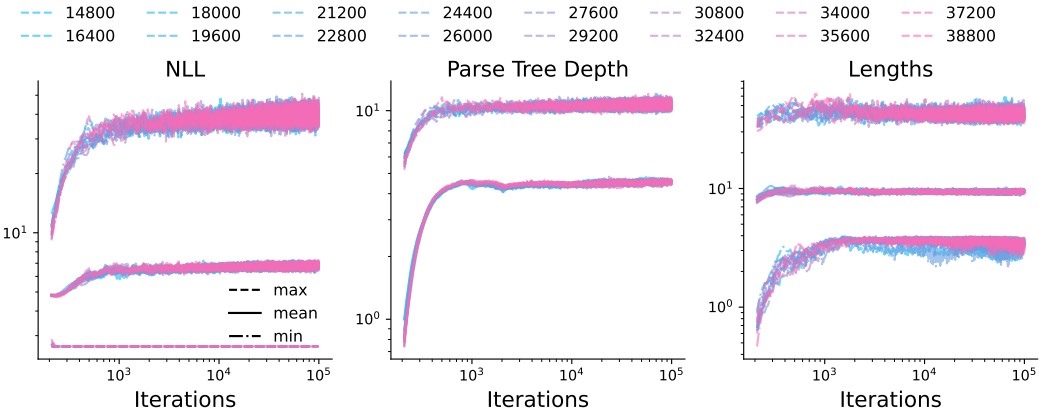

Figure 31: **Grammar characteristics of model generation as number of properties are varied (1800 entities).** See App. F.4 for a detailed discussion. Broadly, model generated sentences are in a similar range as our language's.

### F.4.3 CHANGING TO 2 CLASSES AND VARYING NUMBER OF PROPERTIES

We change the number of classes to divide entities and properties over to 2 (compared to base setting of 10) and report results under varying number of properties. See Fig. 32.

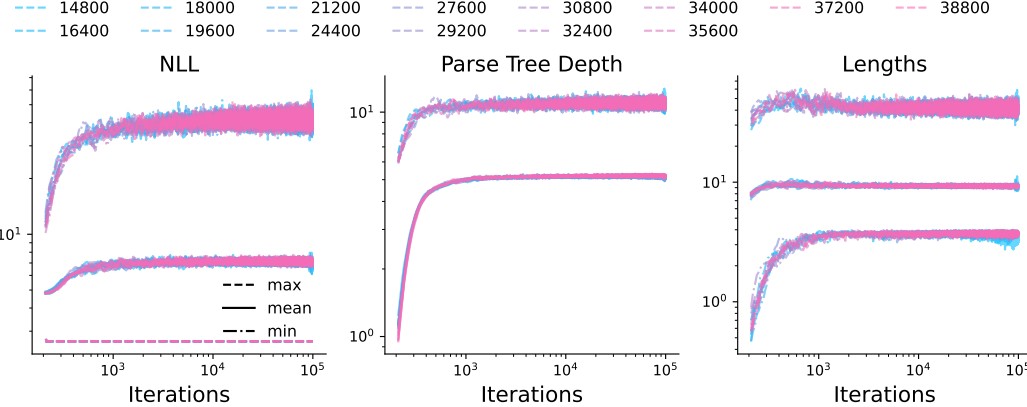

Figure 32: **Grammar characteristics of model generation as number of properties are varied (2 classes).** See App. F.4 for a detailed discussion. Broadly, model generated sentences are in a similar range as our language's.

### F.5 EVOLUTION OF ATTENTION MAPS

As the model acquires the grammar, we expect it to attend to specific parts of the context to predict the next token. Given the simplicity of our model, we can expect the attention maps to be semantically meaningful, especially after the points of emergence. To assess this, every 1000 iterations of training, we record model's attention maps in the base experimental setting on 100 sentences that are either descriptive or relative in nature. Specifically, we define a symbolic sentence that define the grammatical configuration of the sentence, and then sample values for individual tokens according to their roles a 100 times. The attention maps are then recorded and averaged.

### F.5.1 DESCRIPTIVE SENTENCES

Due to space constraints, we only report attention maps at iterations [0, 1000, 10000, 30000, 50000]. The primary motivation here is that around iteration 1000 is when the model first seems to learn the grammar. Similarly, between 1000–10000, it starts to learn type constraints. Then, between 10000–50000, it starts to learn about descriptive properties.

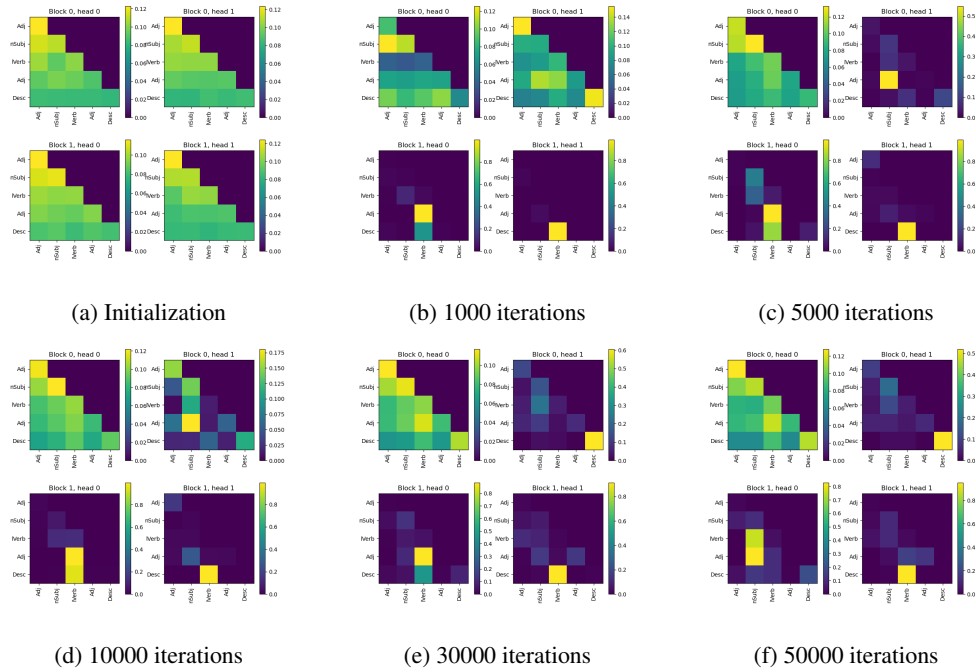

Figure 33: **Evolution of attention maps for descriptive sentences.** Since visualizing the evolution across time is difficult, we pick a few salient points according to region identified as different phase according to learning curves and visualize them. We find that indeed the first time the model sees a sparse attention structure is around 1000 iterations of training, i.e., when it becomes accurate at producing grammatically correct sentences. From 5–30000 iterations, the model learns to focus on subject when producing next token, but the link verb and descriptive property do not pay substantial attention to subject. However, there is a point after 30000 iterations post which the attention on subject substantially increases; this is the range where other evaluation show a change in performance as well.

### F.5.2 RELATIVE SENTENCES

Due to space constraints, we only report attention maps at iterations $[0, 1000, 10000, 30000, 50000]$. The primary motivation here is that around iteration 1000 is when the model first seems to learn the grammar. Similarly, between 1000–10000, it starts to learn relative type constraints. We expect after this range, the attention pattern to not change much—this is indeed what happens. We see the model is basically improving the sharpness of its attention map (pay more attention to tokens that were already being attended).

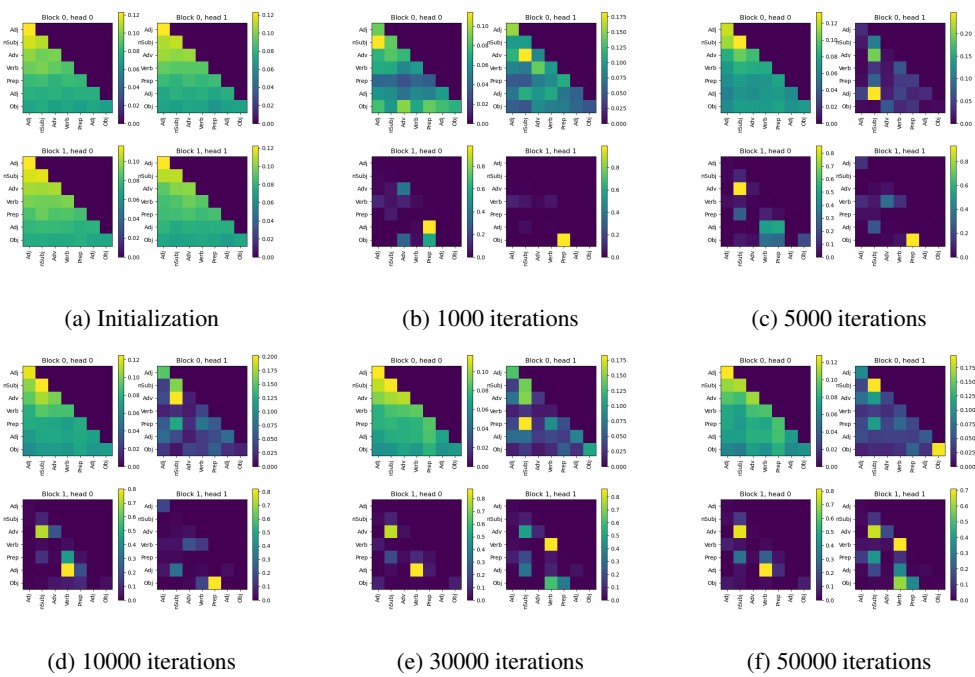

Figure 34: **Evolution of attention maps for relative sentences.** Since visualizing the evolution across time is difficult, we pick a few salient points according to region identified as different phase according to learning curves and visualize them. We find that indeed the first time the model sees a sparse attention structure is around 1000 iterations of training, i.e., when it becomes accurate at producing grammatically correct sentences. Once the model learns relative type constraints (around 5000 iterations generally), we see the attention patterns essentially stabilizes and does not change much; albeit, it does get sharper (more attention to the tokens that were already being attended). Interestingly, we see object tokens paying large attention to verbs, i.e., objects are being selected by analyzing which verbs came before them. We see some non-trivial dependence between verb and the adverb preceding it, which itself solely attends to the subject token (i.e., it copies that token's representation); this could suffice to ensure the model gets verbs right for a subject.

### F.6    MORE RESULTS ON UNSCRAMBLING

In this section, we report several more results evaluating our models' performance on unscrambling under varying number of properties for the base setting, setting with increased number of entities, and with only 2 classes to divide entities and properties over. We report results for the following metrics.

- **Exact Match.** Accuracy of the model for getting every token of the scrambled sentence into its right position in the unscrambled version.

- **Per Token Accuracy.** This metric can be thought of as a smoother version of Exact Match, i.e., it provides partial credit to the model as it learns to solve the task.

- **Accuracy on Descriptive Sentences.** In this evaluation, we compute the exact match accuracy for sentences that are descriptive in nature, i.e., sentences wherein claims are made about an entity possessing a property.
  - We note that the precise way this evaluation is done is by restricting the sentence length to the shortest sentences ($\leq 6$ tokens). This range is primarily constituted of sentences that are descriptive in nature ($\approx$94%).

- **Accuracy on Relative Sentences.** In this evaluation, we compute the exact match accuracy for sentences that are primarily relative in nature, i.e., sentences wherein claims are made about a subject relating to an object via a verb.
  - We note that the precise way this evaluation is done is by restricting the sentence length to the range of (7—9 tokens). This range is primarily constituted of sentences that are relative in nature ($\approx$85%).

- **Grammaticality.** Given the output generated by the model when it is fed in a scrambled input, we evaluate whether the output is grammatically correct or not.

- **Type Check.** Given the output generated by the model when it is fed in a scrambled input, we evaluate whether the output follows type constraints or not. We measure accuracy over all constraints, i.e., we do not decompose this evaluation over descriptive / relative properties.

### F.6.1 BASE SETTING WITH VARYING NUMBER OF PROPERTIES

We report results under varying number of properties with the base experimental setting. See Figs. 35, 36.

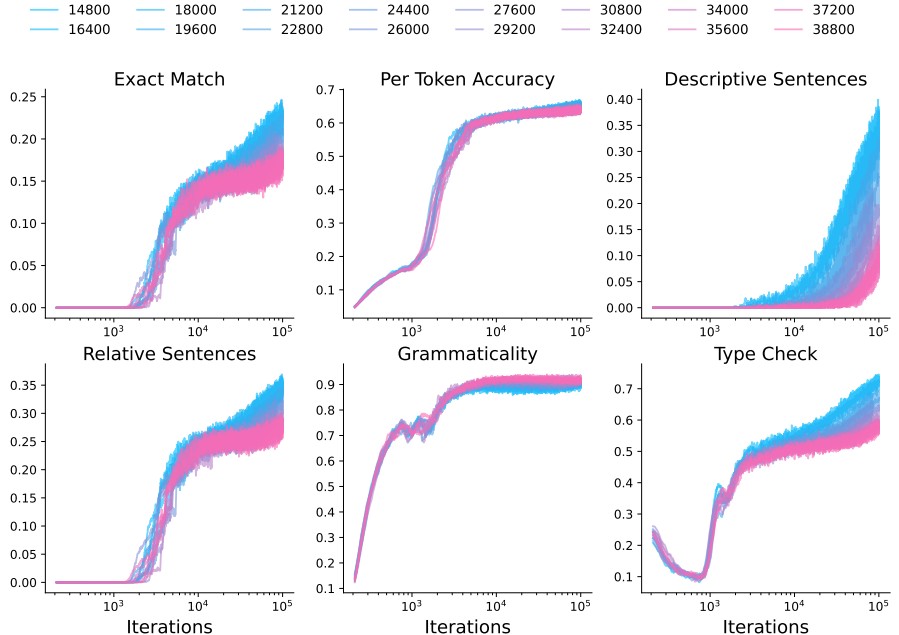

Figure 35: **Results on unscrambling task as number of properties are varied (Base setting).** See App. F.4 for a detailed discussion on metrics used. Broadly, we see the results presented in the main paper are consistent across varying number of properties: the model first witnesses improvement in exact match because relative sentences start to improve, i.e., the grammar is learned, performance then saturates, and then it finally starts improving again once descriptive sentences' accuracy starts to improve. We emphasize metrics assigning partial credit also show sudden changes, i.e., our results are not sensitive to metrics used.

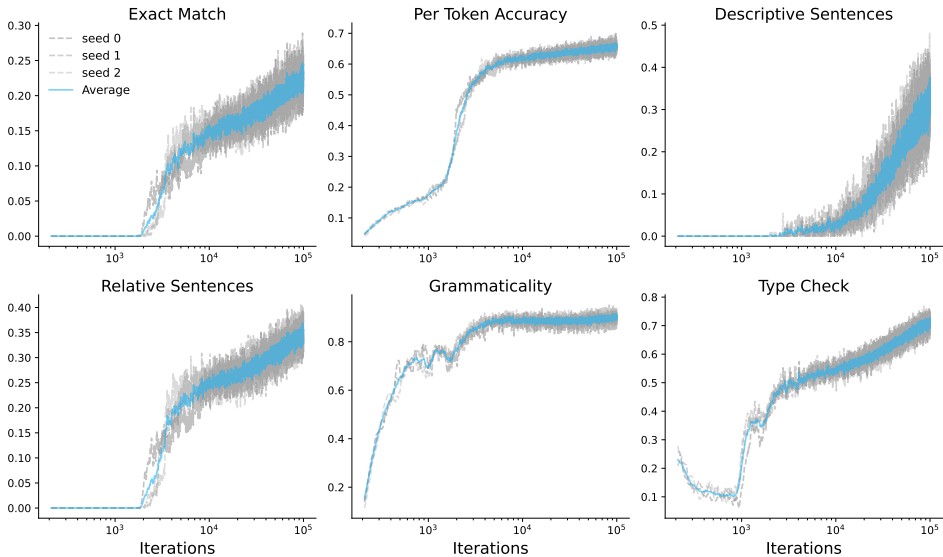

Figure 36: **Results on the unscrambling task for setting with** 18000 **properties (base setting).** We report this plot to zoom into a specific configuration. As can be seen, there is some minimal variance across runs, but mostly points of transition are in a similar range.

We change the number of entities to 1800 (compared to base setting of 900) and report results under varying number of properties. See Figs. 37, 38.

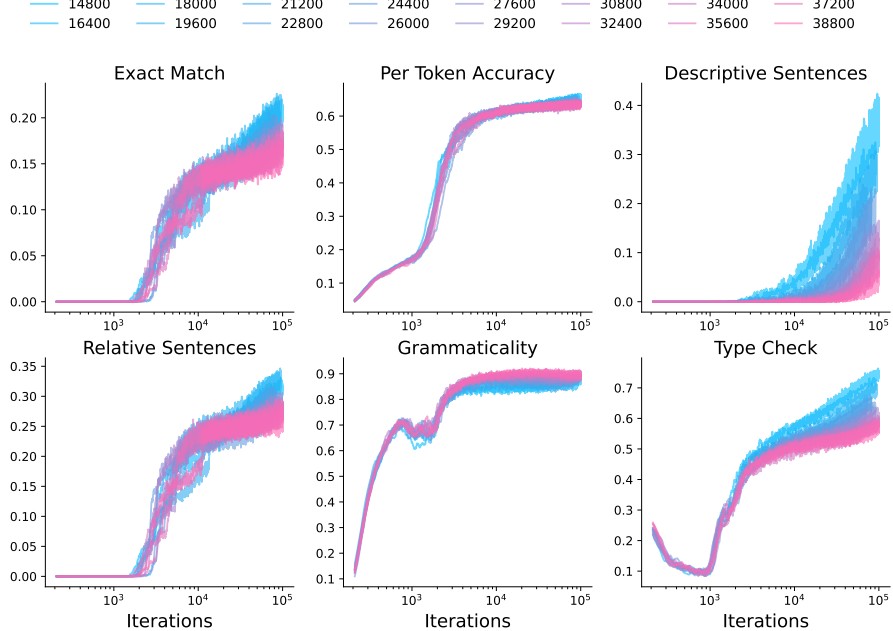

Figure 37: **Results on unscrambling task as number of properties are varied (1800 entities).** See App. F.4 for a detailed discussion on metrics used. Broadly, we see the results presented in the main paper are consistent across varying number of properties and a larger number of entities compared to the base setting: the model first witnesses improvement in exact match because relative sentences start to improve, i.e., the grammar is learned, performance then saturates, and then it finally starts improving again once descriptive sentences' accuracy starts to improve. We emphasize metrics assigning partial credit also show sudden changes, i.e., our results are not sensitive to metrics used.

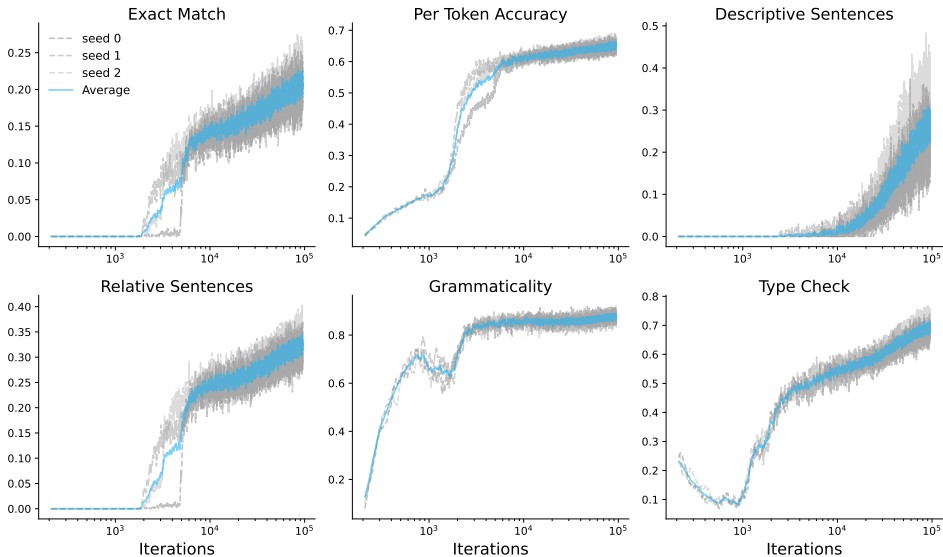

Figure 38: **Results on the unscrambling task for setting with** 18000 **properties (1800 entities).** We report this plot to zoom into a specific configuration. As can be seen, there is some minimal variance across runs, but mostly points of transition are in a similar range.

### F.6.3 CHANGING TO 2 CLASSES AND VARYING NUMBER OF PROPERTIES

We report results under varying number of properties with the base experimental setting. See Figs. 39, 40.

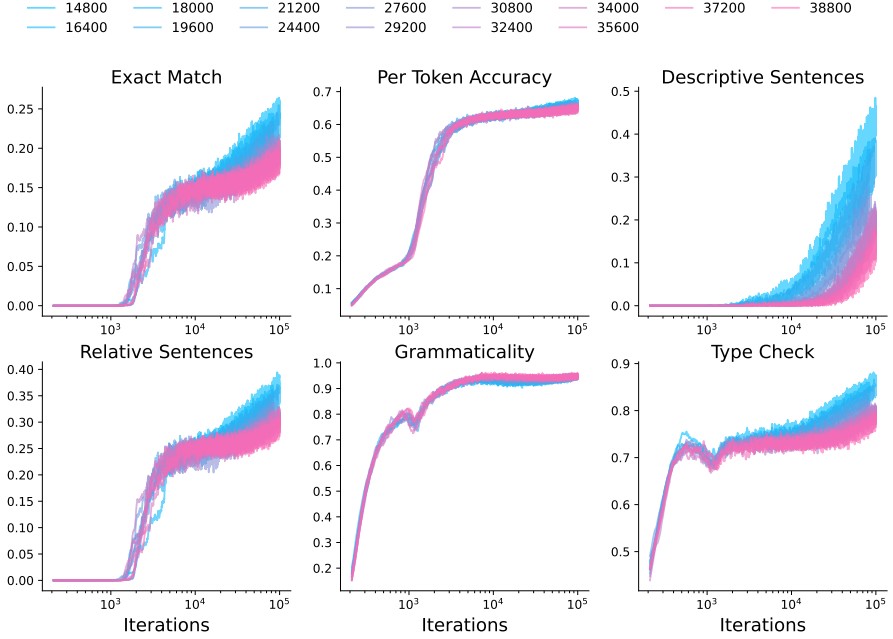

Figure 39: **Results on unscrambling task as number of properties are varied (Base setting).** See App. F.4 for a detailed discussion on metrics used. Broadly, we see the results presented in the main paper are consistent across varying number of properties and fewer classes than the base setting: the model first witnesses improvement in exact match because relative sentences start to improve, i.e., the grammar is learned, performance then saturates, and then it finally starts improving again once descriptive sentences' accuracy starts to improve. We emphasize metrics assigning partial credit also show sudden changes, i.e., our results are not sensitive to metrics used.

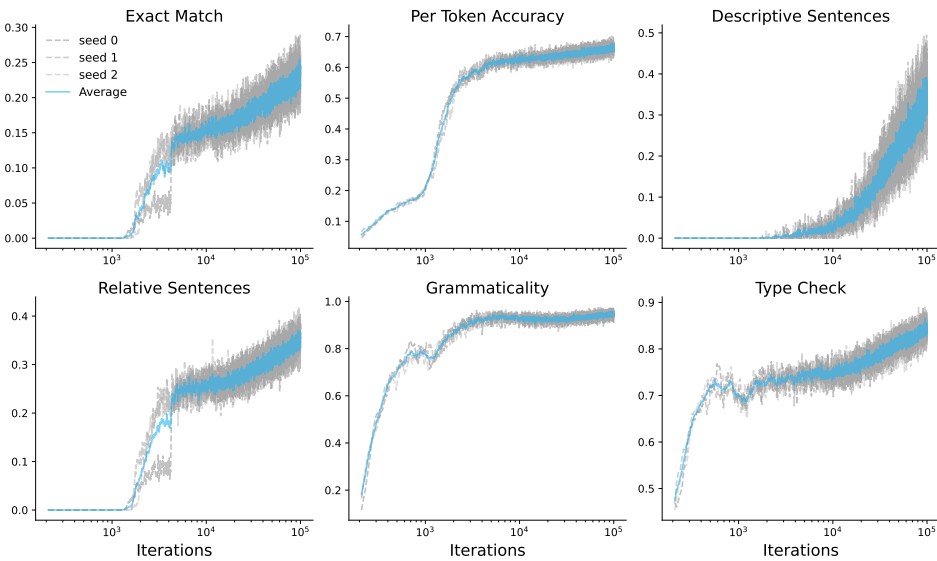

Figure 40: **Results on the unscrambling task for setting with** 18000 **properties (2 classes).** We report this plot to zoom into a specific configuration. As can be seen, there is some minimal variance across runs, but mostly points of transition are in a similar range.

### F.7 ANOTHER SET OF RESULTS WITH CONDITIONAL GENERATION

As mentioned before, Conditional generation evaluations turn out to be extremely time-expensive, with a single run taking approximately 4 days to finish when conditional generation is evaluated (compared to 12 hours without). This is likely a result of model generating extremely long sentences to compose conditioning tokens that can involve multiple subjects, objects, and properties.

While we focus solely on free generation and unscrambling in the results reported in the sections above, to demonstrate that our findings from the main paper (i.e., in the 18000 properties setting) generalize to another setting, we provide results for similar to Fig. 5 in another setting with 27600 properties. Shown in Fig. 41, we can see our findings perfectly align with results from the main paper and other results shown in the appendix: the model first learns the grammar, then type constraints, and witnesses improvements on unscrambling and conditional generation tasks as these relevant data-regularities are learned.

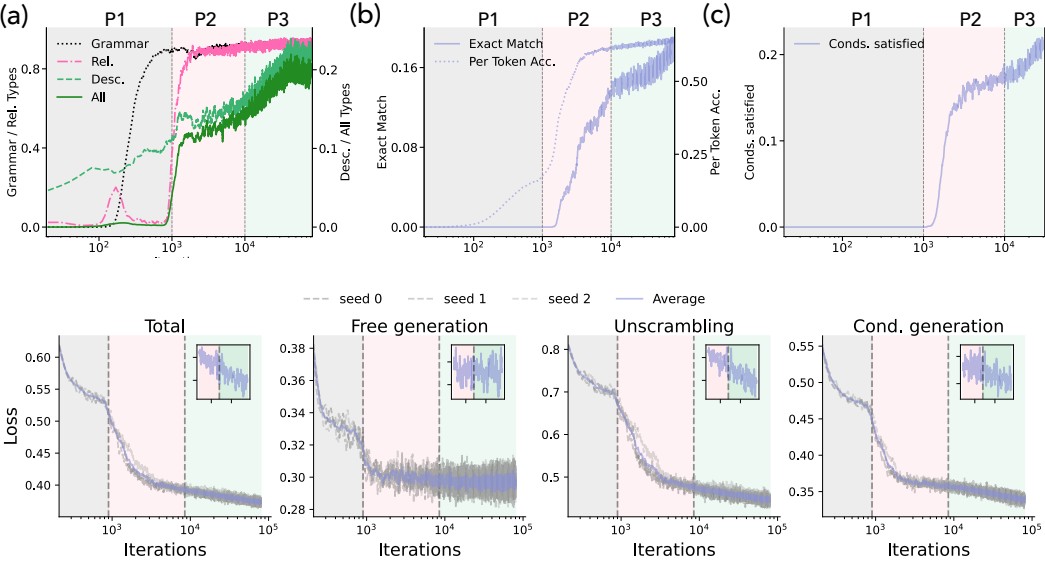

Figure 41: **Demonstration (27600 properties).** We report this plot to zoom into a specific configuration. As can be seen, there is some minimal variance across runs, but mostly points of transition are in a similar range.

# G SCALING OF POINT OF EMERGENCE (AKA TRANSITION POINT)

In this section, we repeat experiments from the main paper for different settings to analyze how the point of emergence (interchangeably called transition point or phase transition here) scales with increase in number of properties in the language. We again report results for both unscrambling and free generation tasks and also add several more metrics not reported in the main paper.

Before proceeding however, we further discuss our evaluation protocol wherein we rescale x-axis by some power of number of properties in the language by connecting it back to the notion of phase transitions and emergence in physics. We also clarify why we might at times need to rescale the y-axis.

## G.1 COLLAPSED CURVES HELP DEMONSTRATE SCALING OF THE TRANSITION POINT

Assume $\mathcal{S}$ denotes a control variable (e.g., edge density in our bipartite graph). Assume change in $\mathcal{S}$ induces a phase transition in our system (e.g., the bipartite graph), as measured by sudden change in the value of some order parameter $\mathcal{M}$ (e.g., ratio of largest cluster size to graph size). Further, say the transition point $\mathcal{S}_c$ depends on some other property of the system $\nu$ (e.g., number of nodes in the bipartite graph) via a power law relationship. That is, we have

$$\mathcal{S}_c \propto \nu^{\alpha}.$$

Accordingly, if we tracked $\mathcal{M}$ as $\mathcal{S}$ is changed, we would find its value rapidly starts to change as $\mathcal{S}/\nu^{\alpha} \to 1$. As we change the value of $\nu$, assume the value of $\mathcal{M}$ at the point of transition is some constant value. Thus, if we plot $\mathcal{M}$ as a function of $\mathcal{S}/\nu^{\alpha}$, we would find the results "collapse" onto each other at the point of transition. This is the intuition behind our experiments in the main paper: as we expect a square-root dependence on the number of properties, if we divide the control variable (training iterations) by $\sqrt{|\mathcal{K}|}$, we should see the curves corresponding to languages with different number of properties collapse onto each other. This argument however assumes we are tracking the perfect order parameter with respect to which theory is defined. This need not be the case, as discussed in the next section.

## G.2 WHAT METRICS MAKE SENSE?

Note that the theory of percolation on a bipartite graph and its corresponding phase transition focuses on the ratio of size of largest cluster in the graph to the overall graph size. That is, the theory of gauging which nodes are member of the largest cluster and how do this metric increase with scaling of edge density. By itself, however, standard metrics one would evaluate in tasks defined in this work, e.g., accuracy, need not linearly correlate with the cluster size. This can affect the collapse visualization discussed in App. G.1. To elaborate further, we build on the toy setup from above.

Say, the order parameter $\mathcal{M}$ is difficult to experimentally gauge—this is in fact the case for our work, where describing and evaluating a notion of membership within largest cluster is difficult. Accordingly, we must define alternative metrics that we expect to correlate with $\mathcal{M}$. Denote this alternative parameter as $\mathcal{M}'$ and say $\mathcal{M}' := \mathcal{M} \times \nu^{\beta}$, i.e., another dependence on $\nu$ gets involved in our experiments as we go from $\mathcal{M}$ to $\mathcal{M}'$. Accordingly, if we track $\mathcal{M}'$ as the rescaled control variable $\mathcal{S}/\nu^{\alpha}$ is varied, we will find that instead of collapsing onto a constant value, systems with different values of $\nu$ have a different value for $\mathcal{M}'$. However, importantly, we will see that the value of $\mathcal{M}'$ at this point itself follows a power law relationship $\mathcal{M}' \propto \nu^{\beta}$. Accordingly, if we rescaled the y-axis by dividing it by $\nu^{\beta}$, we would see the curves collapse onto each other again; that is, we will see that

$$\text{as} \quad \frac{\mathcal{S}}{\nu^{\alpha}} \to 1, \quad \text{we have} \quad \frac{\mathcal{M}'}{\nu^{\beta}} \to \mathcal{O}(1).$$

Thus, when using alternative metrics that are meant to correlate with the gold-standard metric (e.g., $\mathcal{M}'$ instead of $\mathcal{M}$ in the discussion above), a rescaling of the y-axis according to some property of the system may be needed to help induce a collapse of different experimental curves. As discussed in the next section, this subtlety turns out to be extremely crucial for our work.

### G.3 Evaluation Metrics for Evaluating Emergence in Our Work

We find artifacts of the toy problem discussed in sections above in our experiments. Specifically, the theory of percolation on bipartite graph focuses on largest cluster size as an order parameter. However, it can be difficult to define a cheaply calculable metric that captures a notion of 'largest cluster' and evaluates membership of properties and entities to the cluster in the context of a neural network being trained on some data distribution. To circumvent this, we define several alternative metrics that approximate the notion of largest cluster to an extent, but are not necessarily expected to show perfect collapse of experimental curves when the x-axis is rescaled by some power of the number of properties. However, if a mere rescaling of the y-axis by an independent variable (e.g., the control in this experiment, i.e., number of properties) induces a collapse of experimental curves, then we can be confident the transition point follows our expected scaling.

**Evaluation Metrics.** Having discussed the subtleties above, we now discuss the set of evaluation metrics used in this paper to evaluate how the point of emergence (i.e., transition point) scales with increase in number of properties. We analyze the following two tasks in this section: unscrambling and free generation. We use some metrics which are specific to a given task and another batch that is common to both, as discussed next.

- **Unscrambling.** Following metrics are reported *solely* for unscrambling and gauge model's accuracy on the task. As the model learns which properties belong to which entities, we can expect it to exploit that knowledge to reduce the hypothesis space for next-token predictions and get more accurate on unscrambling. Hence, we expect accuracy to suddenly start increasing or at least for its rate of increase to change once the model undergoes a percolation transition.

  - **Exact Match:** Evaluate whether the model's unscrambled sentence perfectly matches the ground-truth.

  - **Per-Token Accuracy:** Evaluate how many of the tokens from model's unscrambled sentence match the ground-truth.

  - **Descriptive Sentences Accuracy:** Exact match accuracy for solely sentences that are descriptive in nature. Similar to prior experiments, we simply filter sentences for length and use ones with $\leq 6$ tokens for this evaluation, since $94\%$ such sentences are descriptive in nature and this allows for easier batching and fast evaluation.

- **Free Generation.** Following metrics are reported *solely* for free generation. Similar to unscrambling, these metrics evaluate a model's performance on the task of free generation, wherein the goal is to produce a sentence that is grammatically valid and respect type constraints. We specifically focus on type constraints in this section. Specifically, as the model learns which properties belong to which entities, we can expect Type Check corresponding to descriptive sentences (see below) will start to improve substantially. In contrast, for Type Checks of relative properties (i.e., validity of verbs), we do not expect to see any effect of how many descriptive properties are there in the language.

  - **Type Check (Descriptive):** A type check evaluation, as discussed in main paper, that checks whether descriptive properties associated with an entity by the model are in fact valid. We expect the percolation phase change to affect this evaluation, yielding *close to* 0.5 scaling with number of properties.

  - **Type Check (Relative):** A type check evaluation, as discussed in main paper, that checks whether relative properties associated with an entity by the model are in fact valid. We expect the percolation phase change to *not* affect this evaluation, since it relies solely on the grammar, and hence there should be no clear effect of scaling number of properties on this metric.

  - **Type Check (All):** A type check evaluation, as discussed in main paper, that checks whether all properties and corresponding entities in a given sentence are allowed to be seen in each other's context. We expect the percolation phase change to affect this evaluation, since, unless the model gets descriptive constraints right, this metric will be zero. However, there will be a non-trivial proportion of sentences that do not have any descriptor tokens within them; we expect improvement on these sentences to increase the overall metric, leading to a saturation phase until the percolation transition kicks in and the model starts inferring which properties are associated with which entity.

- **Common metrics.** Following metrics are reported for both the unscrambling and free generation tasks. These metrics assess whether the model deems a given property and an entity belong to each other, regardless of whether it has seen them together as part of the same context. In this sense, these metrics test a minimal notion of cluster membership, where the cluster is defined by classes dividing the bipartite graph.

  - **Average Probability of Valid Tokens.** Used for evaluating descriptive type constraints in free generation and unscrambling. Specifically, we sample a sentence from $\mathcal{L}$ that remarks on an entity possessing a property, and then evaluate probability of this property being the next token when the sentence is inputted to the model. For example, let $x = \texttt{The fire was large}$, we evaluate $\Pr\left(\mathbb{1}(f(x)_1, \texttt{large})|x_{-1}\right)$, where $x_{-1}$ denotes the sentence up to the last token and $f(x)_1$ denotes the first token predicted by the model. The result is averaged over 1000 sentences.

  - **Negative Log-Likelihood of Valid Sentences.** For free generation, we sample a descriptive sentence from the language and evaluate how likely the model deems this sentence, reporting it as negative log-likelihood (NLL). Similarly, for unscrambling, we sample a random descriptive sentence, scramble it, and then evaluate how likely the model deems the ground-truth unscrambled version.

  - **Normalized Rank of Valid Tokens.** This evaluation is similar to the average probability evaluation above. However, we now compute the rank of randomly sampled descriptor token instead of the probability associated by the model to this token. If the model knows which properties go with an entity, the rank of tokens associated to said entity's properties will be low, indicating they are highly likely to be sampled. This metric scales as a function of vocabulary size; hence, we divide it by the number of properties $|\mathbb{K}|$ and called that the normalized rank.

  - **Percent Top-K.** Similar to the rank metric above, this metric merely checks whether the rank is less than some threshold; if so, it returns True, indicating the model understands that the property being evaluated is valid for the given entity. We set the threshold to be equal to number of properties associated with a class, i.e., $|\mathbb{K}|/C$.

### G.4 EXPERIMENTAL SETTINGS

We analyze the following settings, sweeping the number of properties in the range 14800—38800, in increments of 1600.

- Base setting: This is the setting used throughout the paper, i.e., with 10 classes and 900 entities.

- Different class setting: we change number of classes to 2 and repeat all evaluations in this setting.

- Different entities setting: we change number of entities to 1800 and repeat all evaluations in this setting.

## G.5 SCALING IN THE UNSCRAMBLING TASK

### G.5.1 BASE SETTING WITH VARYING NUMBER OF PROPERTIES

We report results under varying number of properties with the base experimental setting first. See Fig. 42 for metrics specific to unscrambling, Fig. 43 for the common metrics that more closely capture a notion of cluster membership, and Fig. 44 for different x-axis rescalings for the average probability curves that demonstrate validity of claimed scaling of the transition point.

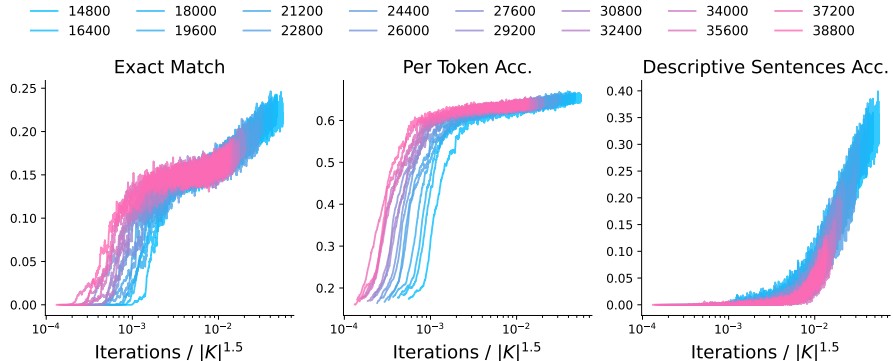

Figure 42: **Results on metrics specific to unscrambling (Base setting).** We see a collapse of all metrics under a 1.5 scaling exponent.

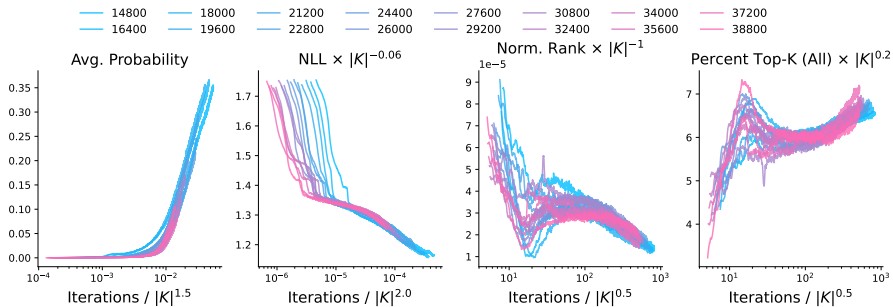

Figure 43: **Results on metrics designed to approximate cluster membership (Base setting).** We see an approximate collapse of inflection points in the normalized rank and percent top-K metrics under a 0.5 scaling exponent. Average probability is expected to follow accuracy curves, yielding a 1.5 scaling exponent. Interestingly, NLL has a scaling exponent of 2.0.

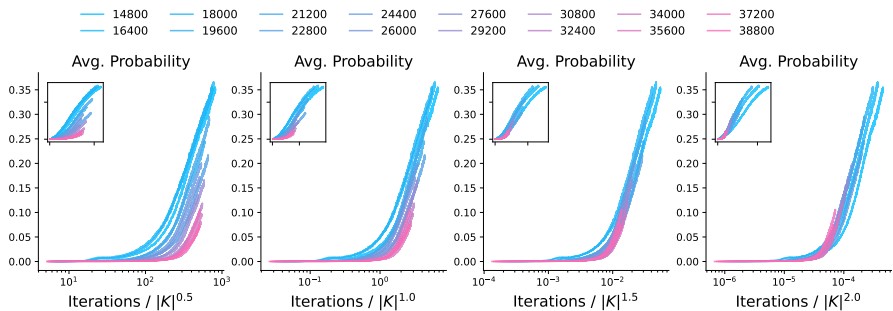

Figure 44: **Average probability curves for different x-axis rescalings (Base setting).** An exponent of 1.5 induces the best collapse of experimental curves. Inset plots zoom in near transition point.

### G.5.2 Changing to 1800 entities and varying number of properties

We change the number of entities to 1800 (compared to base setting of 900) and report results under varying number of properties. See Fig. 45 for metrics specific to unscrambling, Fig. 46 for the common metrics that more closely capture a notion of cluster membership, and Fig. 47 for different x-axis rescalings for the average probability curves that demonstrate validity of claimed scaling of the transition point.

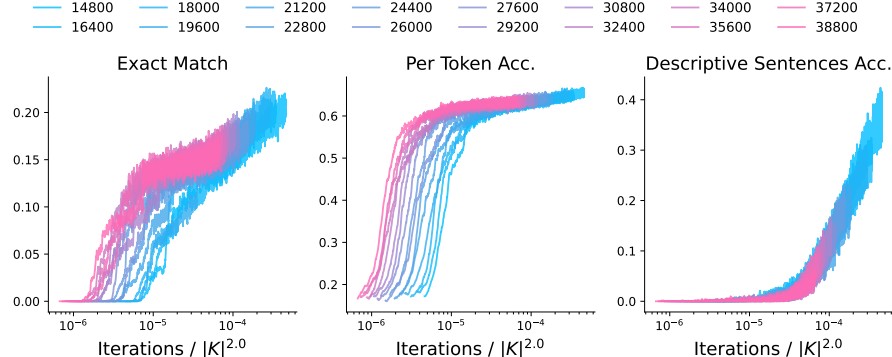

Figure 45: **Results on unscrambling task as number of properties are varied (1800 entities).** We see a collapse of all metrics under a 2.0 scaling exponent, indicating an effect of number of entities possibly on the transition point. Arguably, this is expected since unscrambling is affected by both number of entities and properties involved in the language.

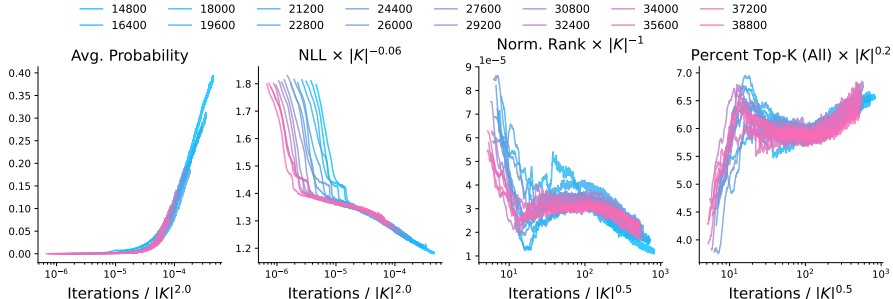

Figure 46: **Results on unscrambling task as number of properties are varied (1800 entities).** We see an approximate collapse of inflection points in the normalized rank and percent top-K metrics with a 0.5 scaling exponent. Average probability is expected to follow accuracy curves, yielding a 2.0 scaling exponent. NLL again shows a scaling exponent of 2.0, similar to base setting.

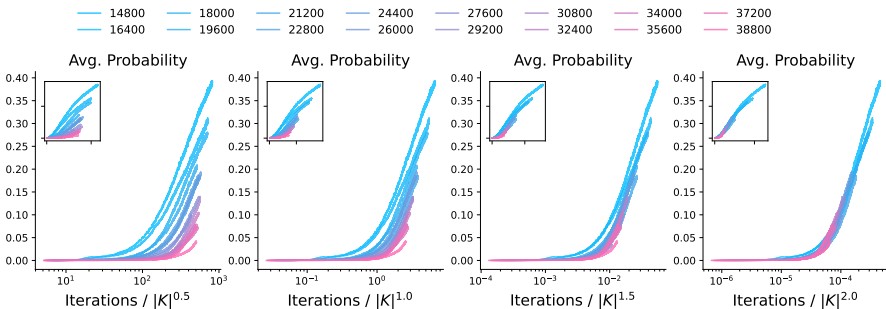

Figure 47: **Average probability curves for different x-axis rescalings (1800 entities).** An exponent between 1.5–2.0 induces the best collapse of experimental curves, similar to base setting. Inset plots zoom in near transition point.

### G.5.3 CHANGING TO 2 CLASSES AND VARYING NUMBER OF PROPERTIES

We change the number of classes to 2 (compared to base setting of 10) and report results under varying number of properties. See Fig. 45 for metrics specific to unscrambling, Fig. 46 for the common metrics that more closely capture a notion of cluster membership, and Fig. 47 for different x-axis rescalings for the average probability curves that demonstrate validity of claimed scaling of the transition point.

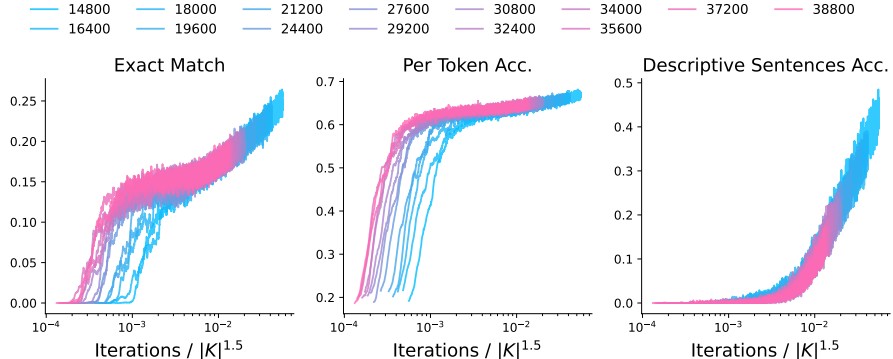

Figure 48: **Results on unscrambling task as number of properties are varied (2 classes).** We see a collapse of all metrics under a 1.5 scaling exponent.

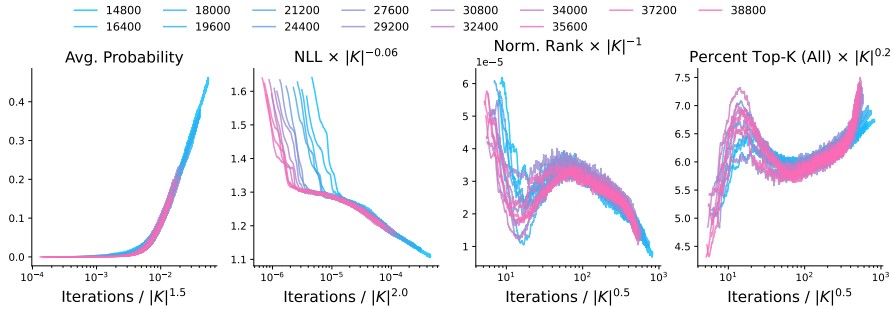

Figure 49: **Results on unscrambling task as number of properties are varied (2 classes).** We see an approximate collapse of inflection points in the normalized rank and percent top-K metrics with a 0.5 scaling exponent. Average probability is expected to follow accuracy curves, yielding a 1.5 scaling exponent. NLL again shows a scaling exponent of 2.0, similar to other settings.

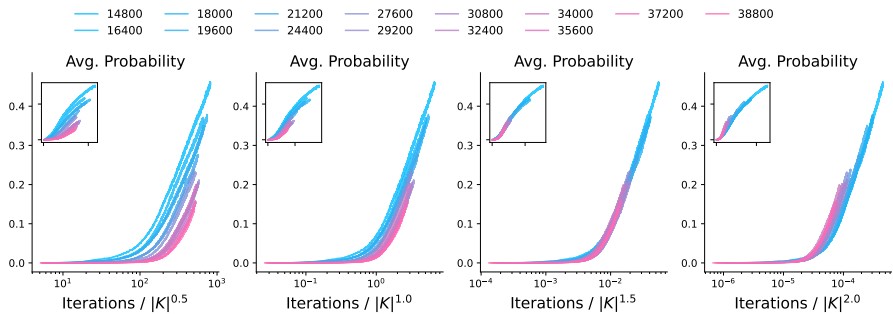

Figure 50: **Average probability curves for different x-axis rescalings (2 classes).** An exponent of 1.5 induces a perfect collapse of experimental curves, similar to base setting. Inset plots zoom in near transition point.

## G.6  SCALING IN THE FREE GENERATION TASK

### G.6.1  BASE SETTING WITH VARYING NUMBER OF PROPERTIES

We report results under varying number of properties with the base experimental setting first. See Fig. 51 for metrics specific to free generation, Fig. 52 for the common metrics that more closely capture a notion of cluster membership, Fig. 53 for different x-axis rescalings for the average probability curves that demonstrate validity of claimed scaling of the transition point and its corresponding variant in Fig. 54 where the y-axis is not rescaled to demonstrate the transition points align better with our claimed scaling exponent.

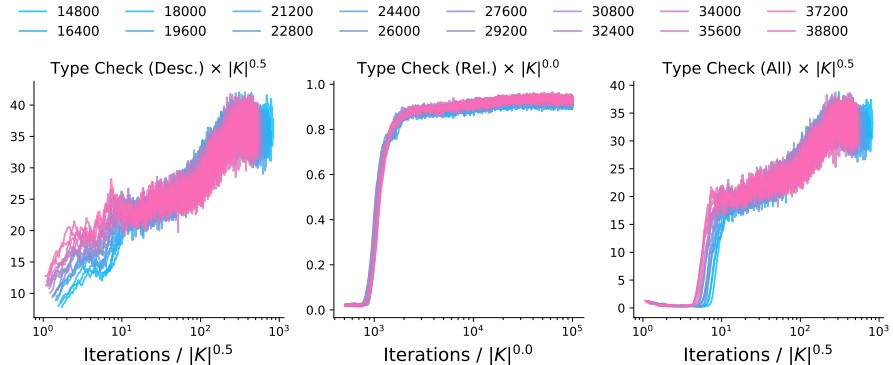

Figure 51: **Results on metrics specific to unscrambling (Base setting).** We see a collapse of descriptive and all constraints metrics under a 0.5 scaling exponent; relative constraints are clearly invariant to number of properties.

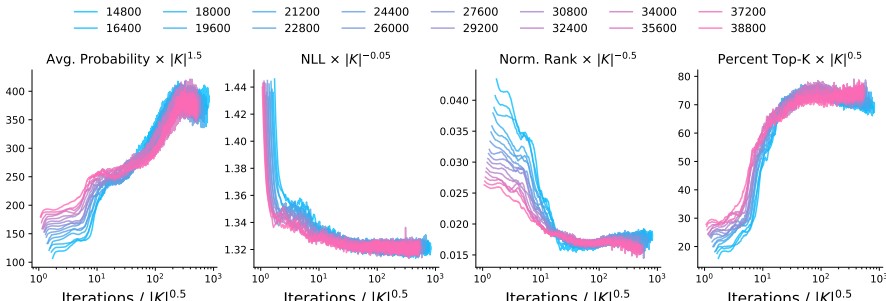

Figure 52: **Results on metrics designed to approximate cluster membership (Base setting).** We see all metrics show an approximate collapse under a 0.5 scaling exponent. Collapse is in the inflection points for the normalized rank and percent top-K metrics.

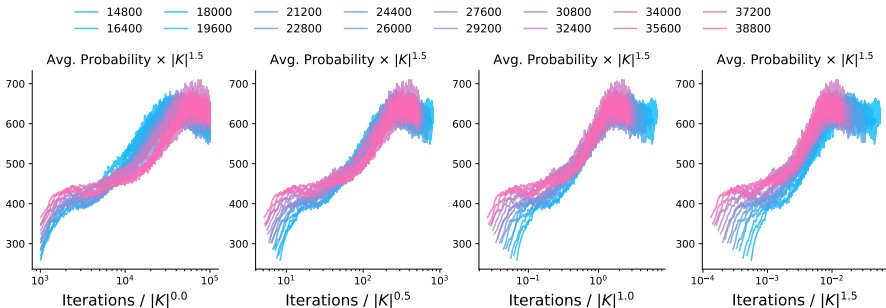

Figure 53: **Average probability curves for different x-axis rescalings (Base setting).** An exponent between 0.5–1.0 can be expected to induce the best collapse for the metric of average probability; results in Fig. 54 show the exponent is closer to 0.5.

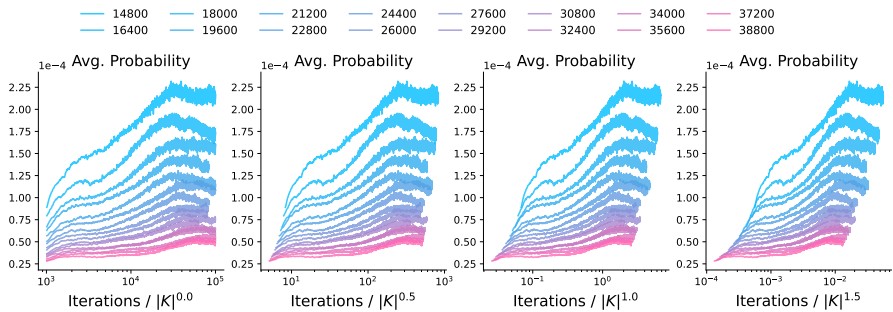

Figure 54: **Average probability curves for different x-axis rescalings (Base setting).** An exponent of 0.5 better aligns the transition points.

### G.6.2 CHANGING TO 1800 ENTITIES AND VARYING NUMBER OF PROPERTIES

We change the number of entities to $1800$ (compared to base setting of $900$) and report results under varying number of properties. See Fig. 55 for metrics specific to free generation, Fig. 56 for the common metrics that more closely capture a notion of cluster membership, Fig. 57 for different x-axis rescalings for the average probability curves that demonstrate validity of claimed scaling of the transition point and its corresponding variant in Fig. 58 where the y-axis is not rescaled to demonstrate the transition points align better with our claimed scaling exponent.

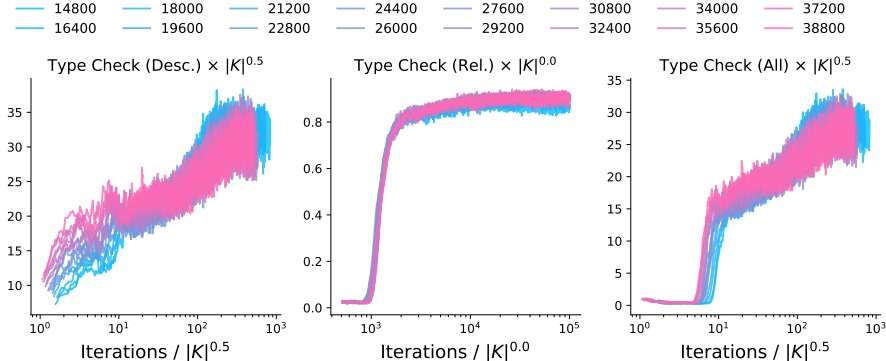

Figure 55: **Results on metrics specific to unscrambling (1800 entities).** We see a collapse of descriptive and all constraints metrics under a 0.5 scaling exponent; relative constraints are clearly invariant to number of properties.

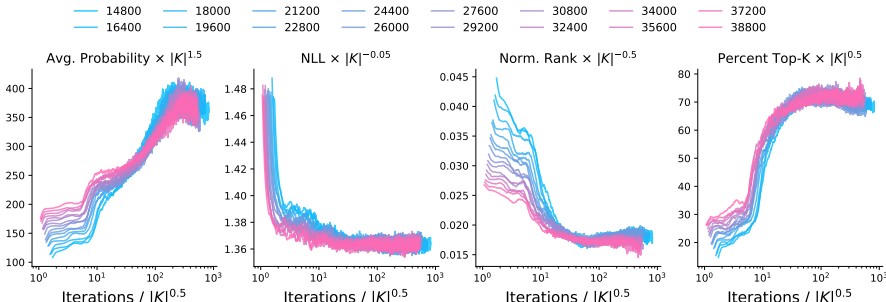

Figure 56: **Results on metrics designed to approximate cluster membership (1800 entities).** We see all metrics show an approximate collapse under a 0.5 scaling exponent. Collapse is in the inflection points for the normalized rank and percent top-K metrics.

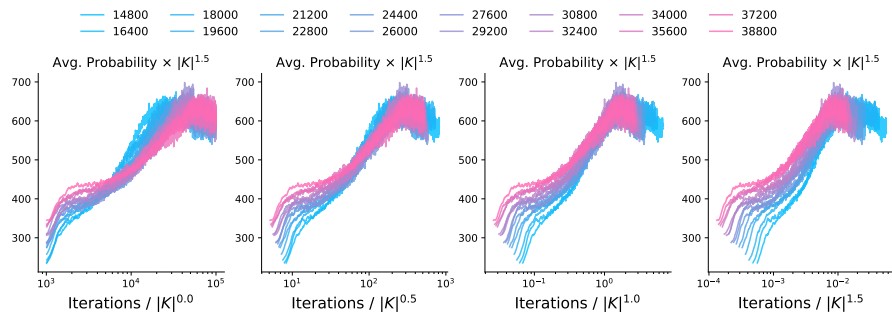

Figure 57: **Average probability curves for different x-axis rescalings (1800 entities).** An exponent between 0.5–1.0 can be expected to induce the best collapse for the metric of average probability; results in Fig. 58 show the exponent is closer to 0.5.

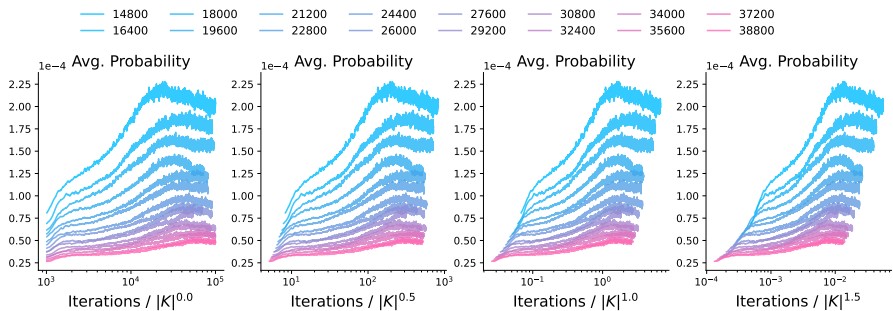

Figure 58: **Average probability curves for different x-axis rescalings (1800 entities).** An exponent of 0.5 better aligns the transition points.

### G.6.3 CHANGING TO 1800 ENTITIES AND VARYING NUMBER OF PROPERTIES

We change the number of classes to 2 (compared to base setting of 10) and report results under varying number of properties. See Fig. 59 for metrics specific to free generation, Fig. 60 for the common metrics that more closely capture a notion of cluster membership, Fig. 61 for different x-axis rescalings for the average probability curves that demonstrate validity of claimed scaling of the transition point and its corresponding variant in Fig. 62 where the y-axis is not rescaled to demonstrate the transition points align better with our claimed scaling exponent.

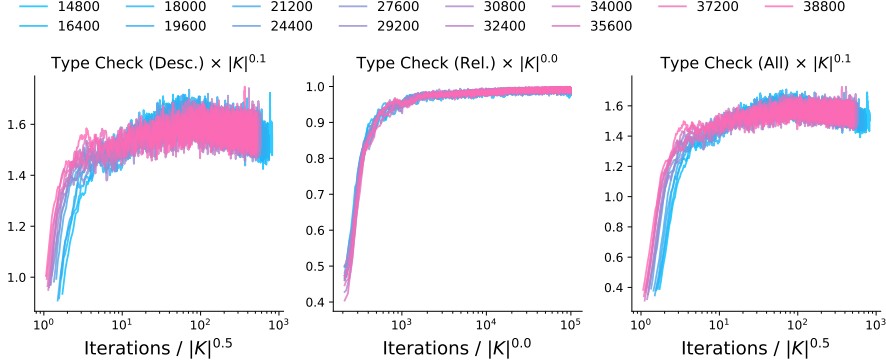

Figure 59: **Results on metrics specific to unscrambling (2 classes).** We see a collapse of descriptive and all constraints metrics under a 0.5 scaling exponent; relative constraints are clearly invariant to number of properties.

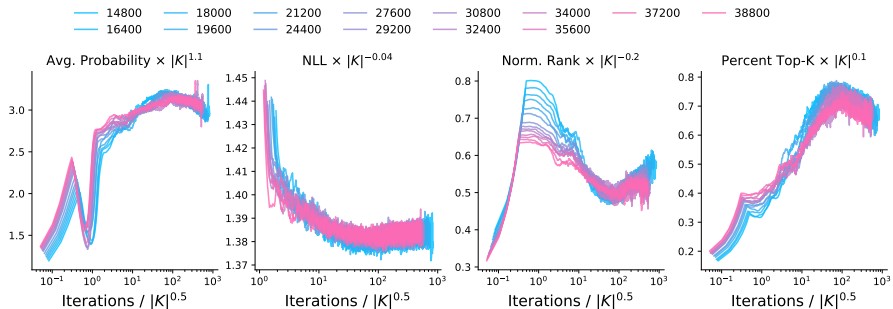

Figure 60: **Results on metrics designed to approximate cluster membership (2 classes).** We see all metrics show an approximate collapse under a 0.5 scaling exponent. Collapse is in the inflection points for the normalized rank and percent top-K metrics.

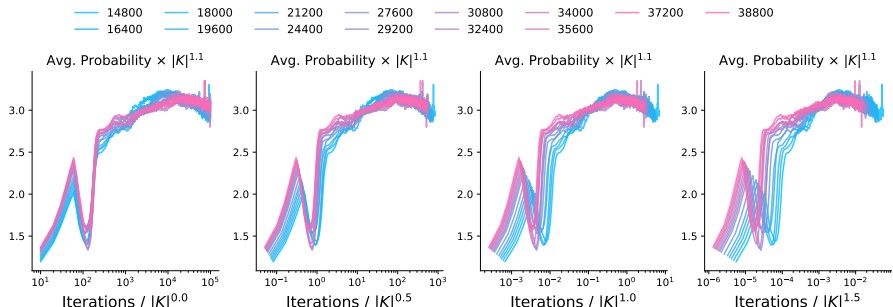

Figure 61: **Average probability curves for different x-axis rescalings (2 classes).** An exponent between 0.5–1.0 can be expected to induce the best collapse for the metric of average probability; results in Fig. 62 show the exponent is closer to 0.5.

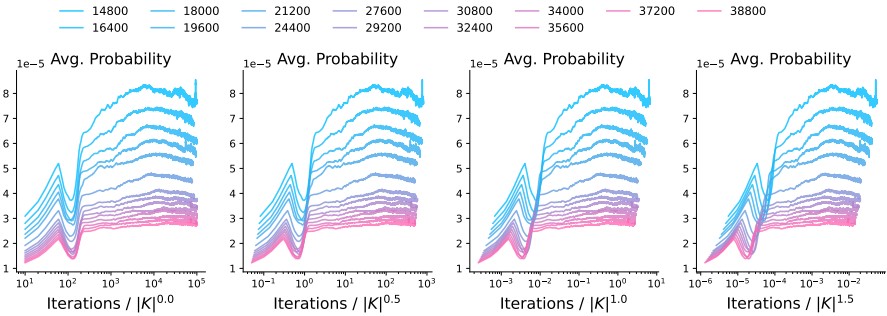

Figure 62: **Average probability curves for different x-axis rescalings (2 classes).** An exponent of 0.5 better aligns the transition points.

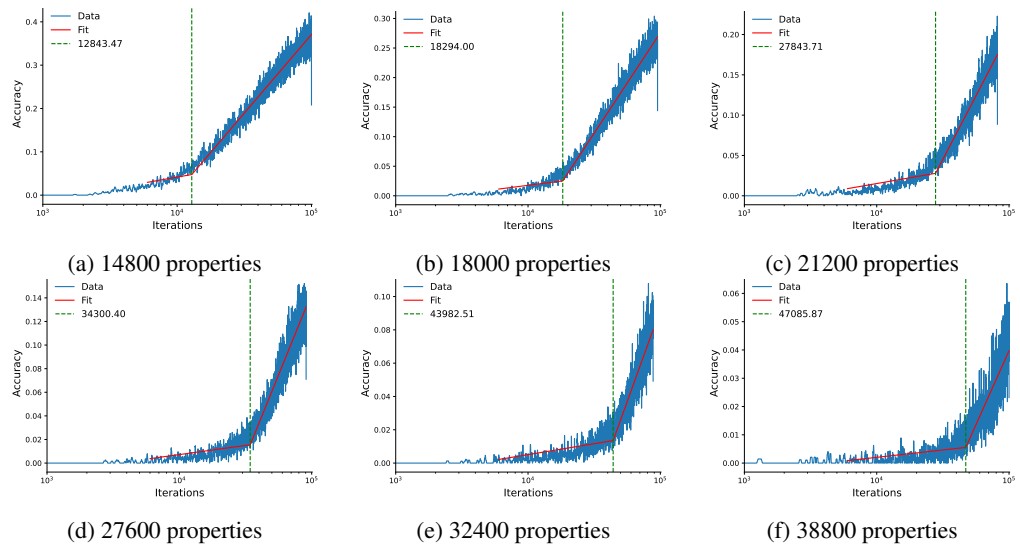

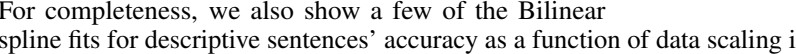

| | | |
|---|---|---|
| (a) 14800 properties | (b) 18000 properties | (c) 21200 properties |
| (d) 27600 properties | (e) 32400 properties | (f) 38800 properties |

Figure 63: **Bilinear Spline Fits.** Working with the hypothesis that the model's accuracy on unscrambling descriptive sentences undergoes first a saturation regime at low performance, before suddenly changing slope and rapidly improving in performance, we fit bilinear splines to maximally explain these results. The breakpoint identified using these fits is used to define the transition point scaling curve in Fig. 64. One can easily see in these curves that the breakpoint is moving rightwards as the number of properties are increased.

## H    ALTERNATIVE ANALYSIS OF SCALING OF TRANSITION POINT

A more conventional analysis of how the transition points scales involves simply identifying the transition point for different experiments, collating the results, and fitting a curve to the identified transition points. We chose the collapse of experimental curves protocol over this methodology since, except for unscrambling, defining an algorithmic objective for curve fitting is difficult. However, at least for unscrambling, we can follow the more usual pipeline and get the curve fits to see if they align with our alternative protocol of collapse of experimental curves.

**Setup.** One can easily see that when the x-axis is log-scaled, both average probability and descriptive sentences accuracy show a scaling curve wherein there is first a saturation at low performance, and then sudden (approximately) linear growth. Exploiting this pattern, we can simply fit a bilinear spline to minimize the mean square error from the data and use the breakpoint of this spline as an approximation to the transition point.

**Results.** We find a power law with an exponent of 1.4 explains the data fairly well (see Fig. 64). That is, the transition point, according to this method, scales as a power law in number of properties with an exponent of 1.4. This exponent is fairly close to the one identified using the collapse protocol, i.e., 1.5.

For completeness, we also show a few of the Bilinear spline fits for descriptive sentences' accuracy as a function of data scaling in Fig. 63.

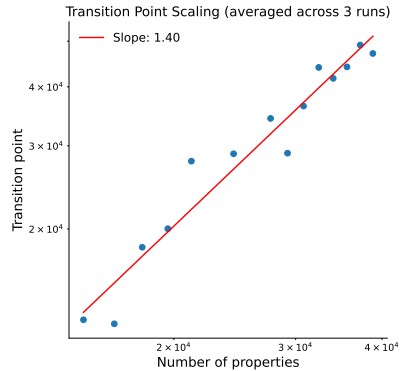

Figure 64: **Transition point scaling.** Working with the hypothesis that the model's accuracy on unscrambling descriptive sentences undergoes first a saturation regime at low performance, before suddenly changing slope and rapidly improving in performance, we fit bilinear splines to maximally explain these results. The breakpoint identified using these fits is used to define the transition point scaling curve.

