# OpenReview forum: "A Percolation Model of Emergence: Analyzing Transformers Trained on a Formal Language"
_ICLR.cc/2025/Conference — ICLR 2025 Poster_

### Official Review · Reviewer_Wp3H · 2024-10-30

**Soundness:** 3
**Presentation:** 4
**Contribution:** 3
**Rating:** 8
**Confidence:** 3

**Summary:**

This paper studies emergence of structure and capabilities of a small transformer throughout training on a formal language dataset.
They identify certain phase transitions correlate with the emergence of capabilities to do specific tasks.
They then propose a formulation to predict phase transitions where emergent capabilities, and find that it aligns well with the formal language toy setting.

**Strengths:**

The paper is well written and a pleasure to read. The paper seems to be also placed well in the context of previous and current related work on emergence. Emergence is an interesting topic for the community, and this paper provides a nice background and definition for studying it in terms of training data. And, while the setting studied is simple, the findings are well supported by their experiments, and the appendix has well-detailed additional evidence.

**Weaknesses:**

There are other aspects of emergence that are not investigated here that need further study. This paper studies emergence over training data scaling, but they mention other axes (e.g. compute or parameter size) that I feel are also important to make more general claims regarding emergence. While the results in this paper are reasonable for the chosen setting, it is unclear whether they will hold in other settings and data choices.

I also wanted to point out a few (recent) papers there are missing from related work, but seemed relevant. The first is Singh, et al.'s [1] work that studies phase transitions of learning subcircuits for in-context learning tasks. The second is Tigges, et al. [2]'s work, which studies how known circuits evolve in the Pythia suite of models over the course of training.
___
[1] Singh, et al. What needs to go right for an induction head? A mechanistic study of in-context learning circuits and their formation. 2024. (https://proceedings.mlr.press/v235/singh24c.html).

[2] Tigges, et al. LLM Circuit Analyses Are Consistent Across Training and Scale. 2024. (https://openreview.net/forum?id=1WeLXvaNJP)

**Questions:**

There was a discussion about order parameters early on in the introduction, but this was then ignored until the last paragraph of the conclusion. Can you clarify how your definition of order parameters are different than/related to "progress measures" that others have proposed to study phase transitions (e.g. [3,4])?

___
[3] Barak, et al. Hidden Progress in Deep Learning: SGD Learns Parities Near the Computational Limit. 2022. (https://openreview.net/forum?id=8XWP2ewX-im)

[4] Nanda, et al. Progress measures for grokking via mechanistic interpretability. 2023. (https://openreview.net/forum?id=9XFSbDPmdW)

---

> ### Author Response · Authors · 2024-11-19
> **Rebuttals (1/2)**
>
> We thank the reviewer for their feedback! We are glad they found the paper to be “a pleasure to read”, well contextualized with respect to past and current work on the topic, and that they liked our discussion of emergence in other fields, which directly motivates our formalization of the concept. We also appreciated their high scores on the contribution, presentation, and soundness of our paper! Below, we respond to specific comments raised by the reviewer.
>
>
> ---
> > **There are other aspects of emergence that are not investigated here that need further study. This paper studies emergence over training data scaling, but they mention other axes (e.g. compute or parameter size) that I feel are also important to make more general claims regarding emergence. While the results in this paper are reasonable for the chosen setting, it is unclear whether they will hold in other settings and data choices.**
>
> We completely agree that other scaling axes are extremely crucial to study! In fact, we are actively working on projects that explore the effects of parameter-size scaling and context-size scaling. Below, we give a high-level overview of our current directions on parameter-size scaling. As we discuss, the precise theoretical models to explain emergence with respect to other axes can be different from the one proposed in this work. In particular, we expect part of the emergent abilities seen with respect to parameter scaling can be explained by our proposed model, but there are also fundamentally new capabilities that parameter scaling unlocks and that data scaling, by itself, cannot. For such capabilities, we expect no theoretical model for data scaling (including ours) will generalize.
>
> - **Effects of scaling parameters: differences from data-scaling.** First, we note that it is easy to see how emergent abilities with respect to parameter size scaling are already predicted by neural scaling laws. For example, if we take the Chinchilla scaling law and assume infinite data therein, we will get the minimum amount of loss achievable under infinite compute and a given model size (this operation assumes Chinchilla scaling perfectly explains model-size limited scaling regimes). We will thus find that the asymptotic compute loss for a follows a power law with the number of parameters in the model---larger models achieve a smaller asymptotic loss! This implies that to the extent loss is a good proxy for capabilities (see [1] for discussion on this), by merely scaling the model size, we can unlock new capabilities. Any theoretical model of emergence that does not account for the effects of number of parameters in a network, including the percolation model we propose in this paper, is unlikely to explain the learning dynamics of such capabilities.
>
> - **Effects of scaling parameters: possible equivalences with data-scaling.** Consider the difference in asymptotic loss of a smaller model and the compute-optimal loss (following Chinchilla scaling) of a larger model. This loss difference is in effect the “speed benefit” of parameter scaling: larger models train faster, yielding more reduction in loss than smaller models. Now, assume a smaller model trained using infinite compute (i.e., using infinite data) achieves a similar loss as the larger model’s compute-optimal loss, hence demonstrating similar capabilities. In such a scenario, we can expect any novel capabilities seen in a larger model and not in a smaller model are a consequence of training under resource constraints. For such capabilities, we can expect our percolation model (and generally any model of emergence proposed for data scaling) to hold well, since scaling resources for a fixed model size primarily involves scaling data.
>
> If the reviewer deems it useful, we are happy to summarize the discussion above in the final version of the paper.
>
> [1] Understanding Emergent Abilities of Language Models from the Loss Perspective, Du et al. (NeurIPS 2024)
>
> ---
>
> > **I also wanted to point out a few (recent) papers there are missing from related work, but seemed relevant. The first is Singh, et al.'s [1] work that studies phase transitions of learning subcircuits for in-context learning tasks. The second is Tigges, et al. [2]'s work, which studies how known circuits evolve in the Pythia suite of models over the course of training.**
>
> Thank you for highlighting these papers! We have now added references to them.
>
> ---
> ---
> **[Continued below...]**

---

> > ### Author Response · Authors · 2024-11-19
> > **Rebuttals (2/2)**
> >
> > **Further questions**
> >
> > > **There was a discussion about order parameters early on in the introduction, but this was then ignored until the last paragraph of the conclusion. Can you clarify how your definition of order parameters are different than/related to "progress measures" that others have proposed to study phase transitions (e.g. [3,4])?**
> >
> > Progress measures are a related, but slightly different, concept. Specifically, progress measures are still meant to be task-centric: they are metrics, that may rely on model internals, to gauge progress towards learning of a specific task. In contrast, an order parameter measures whether a system is organized (or ordered) according to a specific structure. This regularity then enables novel properties in the system (e.g., parallel arrangement of magnetic spins in a material yields magnetic abilities). In our work, the relevant notion of order parameter is how well the model has learned the structure of the data, i.e., grammar and type constraints. Learning these structures directly yields the learning of more narrow capabilities, e.g., unscrambling, free generation, and conditional generation. Overall, an order parameter measures alignment with general structures that can affect several system properties; meanwhile, progress measures are task-centric and are better thought of as alternative metrics for a task.
> >
> > ---
> > ---
> > **Summary:** We again thank the reviewer for their feedback! We hope the reviewer will continue to champion our paper during the discussion period. Please let us know if you have any further questions and we will do our best to answer them in a timely manner!

---

> > > ### Comment · Reviewer_Wp3H · 2024-11-21
> > >
> > > Thank you to the authors for the detailed response and clarification on order parameters vs. progress measures, it is helpful. After reading the other reviews and authors' rebuttals, I think the paper would be a nice addition to the conference and have decided to keep my score.

---

> > > > ### Author Response · Authors · 2024-11-21
> > > > **Thank you!**
> > > >
> > > > We thank the reviewer for their response and are glad to see our clarifications were helpful. Please let us know if you have any further questions!

---

### Official Review · Reviewer_SNGk · 2024-11-03

**Soundness:** 3
**Presentation:** 3
**Contribution:** 3
**Rating:** 8
**Confidence:** 4

**Summary:**

This paper  investigates the phenomenon of "emergence" in neural networks, where a model suddenly acquires certain capabilities after reaching a critical data or computational threshold. The authors propose a new definition of emergence in neural networks, linking it to the acquisition of general structures within the data that drive sudden performance improvements in specific tasks. The authors experiment with Transformers trained on a context-sensitive formal language and observe that once the underlying grammar and context-sensitivity structures are learned, performance on various tasks improves dramatically. This phase transition in model learning is likened to percolation on a bipartite graph, where learning dynamics mirror phase changes. Their results suggest that emergence can be theoretically predicted by understanding the structure of the data-generating process, offering insights for regulating and anticipating the behavior of AI models.

**Strengths:**

Generally, this paper builds a bridge between the LLM and the Physics complex system. The paper uses the phase transition from complex system theory to analyze the emergence of LLMs. This paper has the following strengths:

1. This paper provides a clear definition of emergence, which is slightly differently from previous paper, but it is more formal and general. Also, this definition helps further research the measurement of emergence.

2. This paper trained the LLM from formal languages, which generated from a strict grammar with type check. It aligns with current research.

3. he paper’s findings on emergence and phase transitions are potentially generalizable to other neural network models, not just Transformers trained on formal languages.

**Weaknesses:**

1. Previous paper[1] has already claimed that the emergence abilities is mirage. The paper does not clearly address contradictions with previous work: why does the phenomenon of emergence still occur in this study?

2. The selection of formal language, though it is very popular in recent researches, but the situation is that the models not trained on formal languages still shows good performance. The observation is not convincing for such situations.

3. In graph theory, diminishing marginal effects are quite common; however, there is no clear evidence linking this to the percolation model proposed in this paper. Many graph-theoretic functions exhibit properties such as submodularity, which is one of the reasons behind these phenomena. The final emergence modeling presented in this paper is not entirely intuitive.

[1]Schaeffer R, Miranda B, Koyejo S. Are emergent abilities of large language models a mirage?[J]. Advances in Neural Information Processing Systems, 2024, 36.

**Questions:**

Please refer to the weakness part:

1. Please justify the relationship with previous paper, and the reason why we can still believe the current LLMs have emergence. If the definition is different, please justify the reason why the new definition is equal or a proper approximation of previous ones.

2. Please justify the use of formal languages, and what will happen if we do not train on well-typed formal languages.

3. Please provide more physics intuation of current emergence model. For example, during the freezing of water, the Gibbs free energy of water molecules changes, thereby affecting the intermolecular distance and, on a macroscopic level, the volume. We consider this process to be a phase transition.

---

> ### Author Response · Authors · 2024-11-19
> **Rebuttals (1/4)**
>
> We thank the reviewer for their feedback! We are glad to see they enjoyed our contribution bridging the study of emergence in neural networks and complex systems, found our formal definition can help instigate further work on the topic, and liked our formal languages setup and findings therein, expecting them to generalize to real systems. We also appreciated their high scores on the contribution, presentation, and soundness of our paper! Below, we respond to specific comments.
>
> ---
> > **Previous paper[1] has already claimed that the emergence abilities is mirage. The paper does not clearly address contradictions with previous work: why does the phenomenon of emergence still occur in this study?**
>
> First, we would like to clarify that (i) Schaeffer et al.’s work argued that a very specific type of emergence curve, under a strong set of assumptions, could be transformed into a smooth curve by redesigning the evaluation metric, and (ii) in the past year and a half since Schaeffer et al.’s work, numerous papers have been published that have refined the definition of emergence and reproduced emergent phenomena across diverse setups, further validating its robustness.
>
> Building on this, we note that we discuss and contrast our results with the Schaeffer et al. work [1] at several points in our paper, including references in the intro, Section 4, and a detailed discussion in related work (Appendix B). For example, we mention in Section 4 (L262) that our motivation for reporting multiple metrics is to ensure our results are not confounded by Schaeffer et al.’s argument, i.e., that emergence is driven by use of poorly defined, discontinuous metrics. While we refer the reviewer to Appendix B.1 for a more detailed discussion, *broadly, our argument is that Schaffer et al.’s claim is narrower in scope than what is claimed in their paper:* we provide three precise arguments to demonstrate this in Appendix B.1, and, for brevity, summarize two of them below. *These arguments clearly demonstrate that we see emergence in our setting because our setup is outside the scope of Schaeffer et al.’s claims.*
>
> - **Several studies have shown emergent abilities with continuous metrics.** There is significant evidence for emergent abilities in neural networks that does not rely on use of discrete metrics. For example, in a recent work, Gopalani et al. [2] show a sudden performance increase in a regression setting with an *entirely continuous metric*---specifically, mean square error. Similarly, phase transitions have been formally demonstrated in recent works, e.g., by Lu et al. [3] in regards to in-context linear regression and by Cui et al. [4] in regards to positional code learning in a histogram computation task. Meanwhile, Chen et al. [5] have shown a sudden loss drop in BERT training, which, again, is not a continuous metric. *Overall, given this substantial evidence, we believe the argument by Schaeffer et al. is narrower in scope than what is claimed in their paper*---there are legitimate cases of emergence in neural network training that occur without the use of discrete metrics.
>
>     - **Relating to our work:** In our work, we show performance improvements co-occur with loss drops, i.e., a continuous metric. We can thus comfortably say our results are not confounded by Schaeffer et al.’s argument. In fact, as we mention in Section 4, we also report several other (both continuous and discrete) metrics that show sudden improvements, indicating our results are not confounded by the use of poorly defined metrics.
>
> **[Continued below...]**

---

> ### Author Response · Authors · 2024-11-19
> **Rebuttals (2/4)**
>
> - **Schaeffer et al. make a very strong assumption.** To develop their toy theoretical model that underlies the claim “emergence is a mirage”, Schaeffer et al. make the assumption that an individual token’s loss follows power-law scaling. *This assumption is too strong and arguably incorrect*: the loss, *when averaged across a large number of tokens*, follows a power law (as popularly shown in literature on scaling laws); however, individual token dynamics can in fact be wildly different, and in general unlikely to follow a power law scaling. *In fact, empirical evidence of this point was demonstrated most recently by Schaeffer et al. [8] themselves!* In their recent paper, the authors show that learning dynamics of individual answer tokens' loss can show discontinuous progress. For further evidence in this vein, see the paper by Michaud et al. [9] that shows sudden loss drops for individual task dynamics, but recreates scaling laws via an averaging of loss dynamics across several tasks. Du et al. [10] also demonstrate similar results for pretrained LLM checkpoints.
>
>     - **Relating to our work:** In our work, we can concretely show Schaeffer et al.’s assumption does not hold: e.g., in Figure 4, we show loss curves for individual tasks, finding that learning dynamics of tokens corresponding to just these individual tasks (a subset of the overall tokens in a sentence) does not follow a power law! Thus, we can again comfortably conclude that Schaeffer et al.’s assumption, which was too strong to begin with, does not hold in our setting and hence their claims cannot explain our results.
>
> **Summary:** Overall, we believe that the claim made by Schaeffer et al. is narrower in scope than what is claimed in the paper, and the underlying rationale behind their argument involves a rather strong assumption that is unlikely to hold for language model training (as shown by authors’ own follow-up work), and definitively does not hold in our setting (as shown by our results).
>
> [1] Are emergent abilities of large language models a mirage?, Schaeffer et al. (NeurIPS 2023)
>
> [2] Abrupt Learning in Transformers: A Case Study on Matrix Completion, Gopalani et al. (NeurIPS 2024)
>
> [3] Asymptotic theory of in-context learning by linear attention, Lu et al. (arXiv 2024)
>
> [4] A phase transition between positional and semantic learning in a solvable model of dot-product attention, Cui et al. (NeurIPS 2024)
>
> [5] Sudden drops in the loss: Syntax acquisition, phase transitions, and simplicity bias in MLMs, Chen et al. (ICLR 2024)
>
> [6] Compositional abilities emerge multiplicatively: Exploring diffusion models on a synthetic task, Okawa et al. (NeurIPS 2023)
>
> [7] Emergence of hidden capabilities: Exploring learning dynamics in concept space, Park et al. (NeurIPS 2024)
>
> [8] Why has predicting downstream capabilities of frontier ai models with scale remained elusive?, Schaeffer et al. (NeurIPS 2024)
>
> [9] The quantization model of neural scaling, Michaud et al. (NeurIPS 2023)
>
> [10] Understanding Emergent Abilities of Language Models from the Loss Perspective, Du et al. (NeurIPS 2024)
>
> ---
>
> > **The selection of formal language, though it is very popular in recent researches, but the situation is that the models not trained on formal languages still shows good performance. The observation is not convincing for such situations.**
>
> Thank you for this question. Indeed, modern language models are primarily trained on “natural language” data, largely sourced from internet corpora. However, to achieve *scientific* progress, we have designed synthetic formal language data and reproduced the phenomenon of “emergent abilities in LLMs.” This controlled setup allows us to precisely characterize and understand the mechanisms behind emergence, which would be challenging to achieve with the complexities of natural language data. This approach is akin to using “model systems” in other scientific disciplines, such as medical research, where organisms like mice, marmosets, and macaques are studied to uncover biological mechanisms before translating findings to humans. Similarly, our use of formal languages as a model system has enabled us to uncover novel mechanisms underlying emergent behaviors in LLMs, providing a foundation for future investigations. While testing these claims directly on natural language pretraining dynamics is beyond the scope of the current paper, as the reviewer noted in their comments, our setup yields a sufficiently decent abstraction of natural language settings such that we can expect the results to generalize. We thus hope our insights will motivate further study of emergence from this lens, including an empirical study of in-the-wild LLMs.
>
> ---
>
> **[Continued below...]**

---

> ### Author Response · Authors · 2024-11-19
> **Rebuttals (3/4)**
>
> ---
> > **In graph theory, diminishing marginal effects are quite common; however, there is no clear evidence linking this to the percolation model proposed in this paper. Many graph-theoretic functions exhibit properties such as submodularity, which is one of the reasons behind these phenomena. The final emergence modeling presented in this paper is not entirely intuitive.**
>
> Thank you for your comment. The “diminishing marginal effects”, where the impact of adding edges decreases as the graph becomes denser, is indeed common in **saturated regimes**. However, our focus in this paper is **emergence** and its explanation via percolation phase transition **before** the saturated regime. In this regime, adding a small number of edges can have a disproportionate impact, such as the sudden formation of gigantic connected clusters of nodes. This distinction of regimes is central to our use of percolation theory to model emergence. Please let us know if further clarification would be helpful, and we will incorporate it into the paper.
>
> ---
> ---
> **Further questions**
>
> > **Please justify the relationship with previous paper, and the reason why we can still believe the current LLMs have emergence. If the definition is different, please justify the reason why the new definition is equal or a proper approximation of previous ones.**
>
> Please see our (now further expanded) discussion on the Schaeffer et al. paper [1] in Appendix B.1, and our response above where we summarize why emergence should indeed be expected in LLMs’ training. We believe the claims by Schaeffer et al., though credible in some scenarios, are likely overly broad and hence undermine legitimate cases of emergent abilities. We also believe the assumption underlying their work, i.e., that individual tokens’ loss is governed by a power law scaling, is too strong. Finally, we note that comparing our definition of emergence with Schaeffer et al.’s paper is not possible, since the authors do not offer a formal definition in their work.
>
> [1] Are emergent abilities of large language models a mirage?, Schaeffer et al. (NeurIPS 2024)
>
> ---
> > **Please justify the use of formal languages, and what will happen if we do not train on well-typed formal languages.**
>
> **On use of formal languages.** As noted in the paper (Section 3), our goal was to define a controlled system where we can run a rigorous set of experiments and demonstrate the sudden learning of capabilities by a neural network. To this end, we want to ensure the controlled system is sufficiently realistic such that its analysis can be expected to generalize to more naturalistic scenarios: as the reviewer noted in their comments, our use of a formal language setup is quite likely to enable this! More broadly, we note that this methodology is quite common in other scientific communities; e.g., in neuroscience, one often studies fruit flies for understanding human navigation abilities.
>
> **Consequences of using a not well-typed language.** We have in fact run experiments on languages which lack type constraints! Our results show that the second transition, i.e., the one explainable via the problem of graph percolation, is removed in such a setting; however, the first transition corresponding to syntax acquisition continues to exist. This is expected, since the second transition is about the learning of type constraints---hence, lack thereof will remove the transition. We also refer the reviewer to a recent, very detailed investigation of untyped formal languages’ learning dynamics by Pandey [1].
>
> [1] gzip Predicts Data-dependent Scaling Laws, Pandey (arXiv 2024).
>
> ---

---

> > ### Author Response · Authors · 2024-11-19
> > **Rebuttals (4/4)**
> >
> > ---
> > > **Please provide more physics intuition of the current emergence model. For example, during the freezing of water, the Gibbs free energy of water molecules changes, thereby affecting the intermolecular distance and, on a macroscopic level, the volume. We consider this process to be a phase transition.**
> >
> > We thank the reviewer for requesting a more detailed physics intuition based on phase transition. The Ising model of magnetization, for example, is a simplified model of interacting spins, that exhibits a phase transition and has been used to understand phenomena beyond magnetization such as the liquid-gas transition in water and flocking behavior in birds. This suggests that even simplified models capturing the essence of cooperative interactions can demonstrate phase transition behaviors. Similarly, the percolation model, which we use as the primary analogy for our work, is another example of this broad applicability.
> >
> > Percolation, while not tied to a specific physical system like water, provides a framework for understanding how connectivity can influence phase transitions. In a porous medium, beyond a critical connectivity threshold, liquid flows through the medium. Analogously, in our model, when the density of conceptual connections (e.g., subject-verb pairs) exceeds a critical point, these connections "percolate," linking previously isolated concepts and leading to emergent conceptual structures. This change in connectivity, like the flow in the porous medium, represents a qualitative shift in the system's behavior. This percolation perspective, while rooted in physical intuition, offers a general statistical framework for understanding the emergence we observe in our model as it is exposed to increasingly diverse data.
> >
> > ---
> > ---
> > **Summary:** We again thank the reviewer for their detailed feedback that has helped us better contextualize our paper with respect to past work. To this end, we have substantially expanded our discussion of Schaeffer et al.’s paper (see Appendix B): we provide three arguments showing why that work’s claims are narrower in scope (compared to what is presented in their paper) and reliant on a strong assumption. These arguments result in our work being out of the scope of Schaeffer et al.’s claims. In case our answers and these changes help address the reviewer’s concerns, we hope they will consider increasing their score to support the acceptance of our paper.

---

> > > ### Comment · Reviewer_SNGk · 2024-11-19
> > >
> > > Thank the authors for their in-detail rebuttals. In general I agree with the author's claim: Emergence still exists in their measurements. Also the new intuation from connectivity threshold makes more sense to me, helping me understand how the authors develop their imitation. I will raise my score accordingly.
> > >
> > > Here I have another smaller question based on the author's experiment on not well-typed language: Can these changes be linked to the concepts of first-order and second-order phase transitions? As the authors mentioned the Ising model, perhaps some connections will be behind.

---

> ### Author Response · Authors · 2024-11-19
> **Response to follow-up**
>
> We thank the reviewer for their quick response and increased score!
>
> > **Here I have another smaller question based on the author’s experiment on not well-typed language: Can these changes be linked to the concepts of first-order and second-order phase transitions? As the authors mentioned the Ising model, perhaps some connections will be behind.**
>
> This is an excellent point, and we do believe such a link is plausible! For example, given the sharpness of the transition corresponding to syntax acquisition, we believe it is possible there is a first-order transition at play here; meanwhile, the learning of type constraints, which is a more gradual change, seems likely to be a second-order transition. The latter is in fact inline with what we would expect from our theory of percolation phase transition: the percolation transition is a second-order transition, and hence we expect the progress curve to be continuous and smooth at the first-order derivative level, but there should be an undefined second-order derivative at the point of transition, which is clearly visible in our results (e.g., see Figure 4). This is also demonstrated very clearly in Figure 63, where we show that using a model of two linear fits with a discontinuity best explains the data, again indicating we are observing a second-order transition.
>
> Building on the above, we also take this opportunity to highlight that our work is among the first to theoretically propose and empirically validate a "phase transition" model of emergence, creating a novel interdisciplinary bridge between the study of emergence in physics and AI---exactly the kind of conversation we are engaging in right now!

---

### Official Review · Reviewer_NVjv · 2024-11-04

**Soundness:** 3
**Presentation:** 2
**Contribution:** 3
**Rating:** 6
**Confidence:** 3

**Summary:**

The paper studies the emergence of abilities over the course of training in a transformer language model trained on a formal language. The authors identify distinct phases where different abilities emerge. They also study the point at which one of the abilities transitions from memorization to generalization, and show that this point empirically follows a scaling law that matches the theoretical scaling of bond percolation in a bipartite graph.

**Strengths:**

The phenomenon of emergence of model abilities with scale, and how suddenly this can occur, is of both scientific and societal importance, together with related questions about the transition from memorization to generalization. The paper studies these using a toy setup that is both similar enough to realistic setups to be interesting, but simple enough to be able to isolate and study both of these phenomena. The theoretical explanation using bond percolation is insightful and deserving of follow-up work.

**Weaknesses:**

The paper makes claims about "structures learned by the model" (in Definition 1 and Section 5.1 Phase 1), but I do not think that these are really justified by the evidence in the main body of the paper, which only look at performance metrics. There is some analysis of attention maps in Appendix F.6. However, the main evidence given there seems to be that there is increased sparsity at partially-trained checkpoints compared to initialization, and other qualitative claims that are hard to read off from the plots. It would be easier to tell if these were quantified, but my impression is that this evidence is rather weak. I also think that if this evidence were stronger, it should be in the main body of the paper, since it would be necessary to justify this prominent claim.

That being said, I think there is enough interesting material in the paper without looking at model internals, so my suggestion would be to remove or significantly de-emphasize these claims/this aspect of the paper.

More broadly, I found some of the opening discussion and the definition given in Section 2 a little unnecessary, and took up space that would have been better devoted to explaining the experimental setup and results more clearly, and perhaps covering more results that only made it into appendices. In my opinion it would have been enough to give the high-level motivation, instead of couching it in terms of a new definition that doesn't really add much (especially if the claim about structure in the model is removed).

I also found that at times the presentation got too bogged down in formal details (e.g. Definition 2), and would have preferred to have seen a more accessible, plain-language explanation of things and simple examples, with formal details relegated to appendices for reference if necessary. At other times I found the exposition too rambling (e.g. Section 5.1 Phase 3), and it would have been easier to follow if the main points had been separated out and made concisely (e.g. using bullet points / short headings).

More minor points:
- In definition 1 (if you are keeping it), "nonlinear" could be confusing (e.g. quadratics are non-linear but still change gradually). Maybe you mean "discontinuous"? Or I would perhaps argue that the relevant thing is how gradual the change is (steepness of slope, even if it is locally linear).
- In definition 2, I would have found it a bit clearer to say that S is a non-terminal symbol, and just say you start from S, instead of treating it as a special case and saying you first map S to other non-terminal symbols – like the definition in Appendix C. (Also, the definition in Appendix C looks messed up, you seem to be swapping between N and NT / Sigma and T, unless I am misunderstanding something.)
- I found definition 3 hard to follow. E.g. "Entities have unique identifiers associated with them to help define subjects and objects in a sentence" - do you mean e.g. "John" will have certain attributes like "tall", "brown-eyed" etc.? Consider using plainer language and an example.
- Line 227 "Humans" vs line 228 "humans" - inconsistent capitalization could cause confusion (I assume these are the same thing).
- Line 260: For the indicator variable, maybe consider \mathbbm{1} (from package bbm) instead of \delta (though this is maybe just personal preference)

**Questions:**

Is the specific task (free generation/unscrambling/conditional generation) specified to the model somehow, e.g. with a special token?

For the unscrambling task, is the solution necessarily unique? If not, what's the justification for using exact match/average probability of valid tokens?

---

> ### Author Response · Authors · 2024-11-19
> **Rebuttals (1/2)**
>
> We thank the reviewer for their detailed feedback! We are glad to see they found our contribution relating the problem of bond percolation to emergence insightful and deserving of follow-up work, while appreciating our proposed setup that enabled this connection. We also appreciated their high scores on the contribution and soundness of our paper! Below, we respond to specific comments.
>
> ---
> > **The paper makes claims about "structures learned by the model" (in Definition 1 and Section 5.1 Phase 1), but I do not think that these are really justified by the evidence in the main body of the paper, which only look at performance metrics…**
>
> Thank you for this comment! We believe there is a misunderstanding because of poor phrasing on our end. Specifically, when we write *“structures learned by the model”*, we do not intend to suggest that the model has undergone some fundamental reorganization of, e.g., its neurons or representations. This can certainly happen, but it is not the central focus of our work. Instead, by *“structure”*, we mean an entirely data-centric concept: we mean to suggest there are regularities present in the data, learning of which will aid learning of downstream tasks. For example, syntax is a structure (i.e., a regularity) present in our formal language; when our model learns that structure, we claim and show there is a sudden improvement in performance of the narrower task of sentence unscrambling. This can certainly manifest a mechanistic reorganization in the model, but we do not aim to investigate it within the scope of this project (though we indeed are interested in it!).
>
> To clarify that the notion of "structure" in this paper is intended to be data-centric, we initially used the term *“data structure”* in earlier parts of the paper (e.g., contribution 2 was titled “Learning of general data structures underlies simultaneous jumps in specific metrics”). However, given the reviewer’s comment, we understand that merely saying “data structure” is likely still ambiguous. To address this issue, we have now replaced the term “structure” with variants of *“regularities underlying the data”* throughout the paper. We hope this change helps address the reviewer's concerns!
>
> ---
>
> > **More broadly, I found some of the opening discussion and the definition given in Section 2 a little unnecessary, and took up space that would have been better devoted to explaining the experimental setup and results...**
>
> Thank you for your thoughtful feedback. We agree that achieving a balance between introductory discussion and detailed results is important. However, we would like to highlight that our submission is a rather special case as it introduces one of the first “phase transition” models of emergence in LLMs, and is aimed at building a conceptual bridge to physics---a field that has been studying emergent phenomena for decades. We believe this interdisciplinary approach can inspire researchers from physics to engage with the study of emergence in neural networks while simultaneously bringing valuable concepts from physics into ML research.
>
> To this end, we intentionally adopted a somewhat pedagogical writing style, even if it increases the length, as we believe it enhances accessibility and broadens the paper’s impact. Based on personal interactions and feedback, this approach has been serving its intended purpose effectively. That said, if there are specific parts of the introductory sections that the reviewer feels could be streamlined or adjusted, we are more than happy to consider revisions to address these concerns.
>
> ---
> > **I also found that at times the presentation got too bogged down in formal details (e.g. Definition 2), and would have preferred to have seen a more accessible… At other times I found the exposition too rambling (e.g. Section 5.1 Phase 3), and it would have been easier to follow if the main points had been separated out…**
>
> We thank the reviewer for this feedback and agree Definition 2 can be made more legible. We have attempted this in the updated draft: the definition, while still being formal, is now focused on solely emphasizing the relevant concepts that constitute a PCFG. The following discussion, i.e., the two paras after the definition, now discusses how these core concepts are instantiated in our work. For example, how terminal symbols in our work are usual parts-of-speech from English.
>
> For Section 5, we note that while we do already decompose discussion according to phases, a more fine-grained itemization of observations in these individual phases is difficult to achieve because of space constraints. However, following reviewer’s suggestion, we have separated out the central takeaways of each phase now by boldfacing them. We hope this helps improve the legibility of that section!
>
> ---
> ---
> **[Continued below...]**

---

> ### Author Response · Authors · 2024-11-19
> **Rebuttals (2/2)**
>
> **Minor points**
>
> > **(Paraphrased) In definition 1, "nonlinear" could be confusing. Maybe you mean "discontinuous"? ...**
>
> The reviewer is correct. We have replaced the term “nonlinear” with “discontinuous” now.
>
>
> > **In definition 2, I would have found it a bit clearer to say that S is a non-terminal symbol, and just say you start from S...**
>
> Thanks for pointing this out! We have fixed the typos in appendix C, which had messed up the definition, and have also accommodated S as a non-terminal symbol in Definition 2 now.
>
>
> > **I found definition 3 hard to follow....**
>
> Thanks for raising this point! The quoted sentence that caused confusion is indeed a bit pedantic. The motivation behind having it was to create a delineation between a node and a token used to refer to that node (i.e., an identifier). This is useful for completeness, but arguably unnecessary for understanding either of the empirics or the theory. We have thus removed it from the definition.
>
> > **Line 227 "Humans" vs line 228 "humans" - inconsistent capitalization...**
>
> Thanks for catching this! We have fixed the typo.
>
>
> > **Line 260: For the indicator variable, maybe consider \mathbbm{1}...**
>
> We have switched to using \mathbbm{1}. Thanks for the recommendation!
>
> ---
> ---
> **Further questions**
>
> > **(Paraphrased) Is the specific task specified to the model somehow, e.g. with a special token?**
>
> Yes! There is a special token at the beginning of a sentence which conditions the model to perform one of the tasks.
>
> > **For the unscrambling task, is the solution necessarily unique?**
>
> There are indeed sentences for which there is no unique solution, e.g., if there are multiple subjects present in a sentence and an adjective is valid for all of them, then its unscrambled version could have the adjective used for any of the subjects. Handling this ambiguity was noncritical since the percentage of sentences where this is a concern is relatively low (~10%).
>
> ---
> ---
> **Summary:** We thank the reviewer for their detailed feedback that has helped us improve the legibility of the paper. Specifically, we have now updated Section 3 to focus more on the intuitive interpretation of definitions. Please let us know if there are any further concerns and we would be happy to address them!

---

> > ### Comment · Reviewer_NVjv · 2024-11-20
> > **Response to rebuttal**
> >
> > Thank you for the thorough rebuttal, and I am glad to see that some of my suggestions were helpful.
> >
> > The new phrasing of definition 1 is indeed clearer. However, it seems to me that the way you check the third bullet point (that the model "learns regularities underlying the data") is by checking the performance on downstream tasks that require learning the regularity, making this requirement somewhat redundant with the second bullet point. Would not a more parsimonious definition that better reflects your usage simply be "a discontinuous improvement in the performance of tasks benefiting from C"? But then the definition is almost tautological, which gets back to why I didn't find this discussion section especially productive.
> >
> > That said, I don't find this section objectionable, only somewhat redundant, so if you want to keep this section because it appeals to another audience (such as physicists), then I don't hold it against you. My score reflects the merits in the remainder of the paper.

---

> > > ### Author Response · Authors · 2024-11-21
> > > **Thank you!**
> > >
> > > We thank the reviewer for their response and are glad to see they appreciated the revisions. Please let us know if you have any further questions!

---

### Official Review · Reviewer_dr5M · 2024-11-04

**Soundness:** 2
**Presentation:** 4
**Contribution:** 2
**Rating:** 6
**Confidence:** 3

**Summary:**

The paper studies emergent capabilities in transformers via two case studies. In the first study, they look at learning of formal languages (in particular, a language generated via a PCFG). For this setting, they train GPT-2 sized models from scratch for:
-  left-to-right auto-regressive language modeling
- an unscrambling task that requires the model to take a set of words and convert it into a valid string
- a conditional generation task that requires the model to generate a sentence that has certain words in it.

As the model trains they track grammaticality (as measured by whether model generates strings that the PCFG accepts), and if the generated strings follow type constraints. They break down learning into 3 phases, and find that these phases correspond to jumps in the downstream performance (either exact match acc for unscrambling, or loss for language modeling).

In the second study, they study concept acquisition where entities are associated with types. In particular, they model a concept matrix where row i corresponds to the ith entity, and column j corresponds to the jth type, and the ij entry in the matrix is the probability with which these are seen together. They then define a concept propagation matrix, and use connectedness properties of this propagation matrix to define phase changes. They find that analytic values of these connectedness properties correlate with whether the transformer learns specific concepts.

**Strengths:**

- The paper is extremely well written, and focuses on clearly understanding the phenomenon of emergence (albeit in the limited setting of language modeling of formal languages).
- Explores a new setting of learning entity type relationship, as percolation on a bipartite graph. I believe such a setting has not been explored before (though i'm not sure how it connects to emergence of skills / behaviors in transformers)

**Weaknesses:**

Phases of learning: I’m not convinced with the learning dynamics story here. Just because the model can generate accurate sentences does not mean that it has acquired grammar. Understanding whether the model has acquired grammar has been studied previously in NLP: a better method to do this would be to create minimal pairs with one grammatical and one ungrammatical sentence, and check if the model assigns a higher prob to the grammatical sentence. Ofcourse, the design of the minimal pair needs to be well thought-of, to eliminate shortcuts. Here is an example of a minimal pair that checks if a model can produce the correct number for a verb:

S1: The man who likes apples is here

S2: The man who likes apples are here

Not clear what is the point of the percolation model: This seems less about emergence of structure in the model, and more about how at a specific data setting, generalization can happen. I’m not sure what the analogy is between learning type constraints (which is a function of training time), and graph percolation (which is a function of the data properties |E| and |K|). But if the authors can clarify this, i'm happy to increase my score.



Not clear what are new findings in this paper:
- Many of the conclusions from this paper are also in Murty et al. 2024, who also train transformer language models on formal languages, and find emergence of the correct learning rule, with extended training. They also find that such emergence happens alongside the emergence of tree-structures in transformers.

- Similarly, Chen et al. also have a very similar setting but with masked language models, and show that grammar is abruptly acquired, and such grammar acquisition has a causal relationship with downstream performance.
- There’s also other work by Allen-Zhu et al, who train transformers on formal languages, and find evidence of learnability under some constraints.

**Questions:**

- Do you see similar phase transitions for language learning with smaller models or bigger models? In general, do architecture tweaks change the dynamics in non-trivial ways?

---

> ### Author Response · Authors · 2024-11-19
> **Rebuttals (1/3)**
>
> We thank the reviewer for their feedback. We are glad they found our paper “extremely well written”, “focused on clearly understanding the phenomenon of emergence”, and the connection between learning of type constraints and percolation a novel result! Below, we respond to specific comments.
>
> ----------------------------
> > **Phases of learning: I’m not convinced with the learning dynamics story here. Just because the model can generate accurate sentences does not mean that it has acquired grammar … create minimal pairs with one grammatical and one ungrammatical sentence, and check if the model assigns a higher prob to the grammatical sentence…**
>
> Those are great points, and we agree that mere generation of valid sentences is insufficient to claim the model has learned grammar. *We’d like to highlight that this is precisely the reason why we reported in Appendices F.3–F.5 several experiments that stress-test or evaluate how accurately the model captures our formal language!* This includes experiments that are inline with the one suggested by the reviewer, i.e., assessing whether valid sentences are assigned higher probabilities when compared to invalid ones (Appendix F.3), and experiments where we assess whether broader statistics of sentences (e.g., distribution of parse tree depths and sentence lengths) generated by our language versus our trained model are similar (Appendix F.4). We also report attention maps in Appendix F.5 at different points in training, finding they (partially) encode parse trees. Below, we briefly summarize these results. We also note that to reflect your valuable feedback, we have now edited Section 4 to better emphasize that experiments evaluating and stress-testing how accurately the model captures our formal language are available in the appendix.
>
> 1. **Evaluating likelihoods of grammatically (in)valid sentences.** In Appendix F.3, we generate valid sentences from our language and compute their negative log-likelihood (NLL) under the model. We then perturb these sentences (see list below), creating their invalid counterparts and comparing their NLL to the valid ones. We find that the precise point where the NLL of valid sentences suddenly improves, the NLL of invalid sentences suddenly degrades! This first occurs at the point where we claim the model acquires syntax, then the point where it learns which subject-object pairs can be seen in the same context, and finally at the point where it starts to accurately learn which adjectives can be assigned to an entity. This one-to-one match gives credence to the idea that, at these points, the model is learning the rules underlying our language, since these points are precisely where it starts to associate a much worse NLL to sentences that are invalid under our language!
>
>     - **Perturbation 1: Randomize grammar.** Herein, we randomly permute tokens from a valid sentence, hence breaking its grammatical structure. While this is an extremely strong intervention, we note the tokens constituting the sentence themselves are allowed to be seen in the same context, i.e., they follow type constraints. As we show in Figure 23d and 24d, the NLL of such sentences is much higher than the valid sentences (the latter is reported in Figure 23a and 24a). Thus, the model deems the perturbed sentences to be severely less probable than the valid ones.
>
>     - **Perturbation 2: Randomize values.** This perturbation is similar to the one suggested by the reviewer. Specifically, we alter sentences to randomize specific parts of speech, e.g., we use either incorrect adjectives or incorrect subject-objects pairs to form a sentence. As an example, consider a valid sentence such as “Tall man walked on road”. Our perturbation may replace the object herein to create an invalid sentence such as “Tall man walked on water”, or replace the adjective to create a sentence such as “Plastic man walked on water”. We perform one such perturbation per sentence. Results again show that the model assigns higher NLL to such perturbed sentences (Figure 23c and 24c) than it does to valid ones (Figure 23a and 24a)---that is, it deems perturbed sentences to be much less probable than the valid ones.
>
> **[Continued below...]**

---

> ### Author Response · Authors · 2024-11-19
> **Rebuttals (2/3)**
>
> 2. **Evaluating broader language statistics.** In Appendix F.4, we report results on how accurately the model captures the statistics of our language. Specifically, we randomly sample 1000 sentences from the language at different checkpoints and report the min / mean / max value of (i) sentence NLL under the data-generating process (i.e., the language itself), (ii) parse tree depths, and (iii) sentence lengths. The latter two metrics, while crude, give an idea of whether random sampling from the model yields sentences that are relatively unlikely under the language itself (e.g., sentences with large tree depths or large lengths). As results show, the model’s distribution, as captured by min / max / mean values of these metrics, matches that of the ground truth language’s! For example, parse tree depths vary from 3–15 in both the ground truth language and the model generations.
>
> 3. **Attention maps.** We also report the attention maps at different points in training in Appendix F.5, finding the parse tree of a sentence is broadly represented in the attention maps themselves. This is especially clear for longer sentences, e.g., ones with relative type constraints.
>
> ----------------------------
> > **Not clear what is the point of the percolation model: This seems less about emergence of structure in the model… I’m not sure what the analogy is between learning type constraints ... and graph percolation ...**
>
> Thank you for raising this question! We have now added a new figure and section in the appendix (see App. D.1) to better clarify the percolation analogy, while slightly rewording the exposition in the main paper and adding a reference to this new section for details. We provide a summary of the analogy below, but first clarify what structure means in our paper.
>
> **On the term “structure”:** We note that our claims and results are not about the emergence of a “structure in the model”, i.e., we do not claim model neurons suddenly organize in some structured fashion at the point of emergence (as happens, say, in grokking). Instead, our claim is about the model learning the *structure underlying the data*. For example, we deem *type constraints as a structure present in our language*; these constraints define a set of rules that control which entities and properties can be seen together in a sentence, hence yielding context sensitivity in our language. By the phrase “model learns a structure”, we mean that the model identifies and learns a *regularity in data*. The analogy to the problem of percolation on a bipartite graph is meant to capture the learning of this data-regularity, giving us a predictive model of how the neural network learns to suddenly predict an unseen combination of entity and property that can be used to form a valid sentence.
>
> **Analogy between percolation and learning of type constraints.** Broadly, we analogize a model’s learning of type constraints over time as the addition of edges to a bipartite graph that represents our neural network’s knowledge: this imagined graph represents which entity–property combination the network can possibly use to form a valid sentence. More specifically, consider a time-varying bipartite graph whose nodes denote entities (left side of the graph) and properties (right side of the graph) (see the newly added Figure 11a). As training proceeds, the model sees sentences that show a subset of these entities and properties can be combined to form a valid sentence; we thus put an edge between an entity and property nodes if the model has seen a sentence defined using them (see Figure 11b). Then, every training iteration, a new set of edges is added to the bipartite graph (see Figure 11c).
>
> After enough time has passed, we will have added enough edges to the graph above such that for any randomly picked entity and property nodes, there exists a path on the graph that connects them (see Figure 11d). That is, from an entirely statistical perspective, the model will have been shown enough data to know that even an unseen combination of entity and property can be used to form a valid sentence. *The critical edge density that leads to the existence of such paths, and hence assures information necessary to learn about which entities and properties can be combined to form a valid sentence has been seen, is exactly the problem of bond percolation on a bipartite graph:* in that problem, we aim to identify the critical number of edges needed for the formation of a large connected component in the graph, i.e., for formation of paths between any random set of nodes from the graph. If this analogy is correct, we can expect the transition point of learning descriptive type constraints to follow the theory of bond percolation on a bipartite graph. Our results show this prediction checks out: we find the transition point scales in the same way as the percolation theory predicts.
>
> -----
>
> **[Continued below...]**

---

> ### Author Response · Authors · 2024-11-19
> **Rebuttals (3/3)**
>
> > **Not clear what are new findings in this paper: Many of the conclusions from this paper are also in Murty et al. 2024, who also train transformer language models on formal languages ... Similarly, Chen et al. also have a very similar setting but with masked language models, and show that grammar is abruptly acquired ... There’s also other work by Allen-Zhu et al, who train transformers on formal languages ...**
>
> Thank you for highlighting the need to clarify the novelty of our contributions in relation to recent works. While the papers referenced by the reviewer are highly relevant to our work, we emphasize there are crucial differences that distinguish our contributions, as we summarize below.
>
> - **A percolation model of emergence.** To our knowledge, we are the first to establish a connection between the theory of percolation on a bipartite graph and emergent abilities in neural networks, yielding a predictive estimate of the point of emergence. We note none of the papers referenced by the reviewers offer this contribution; in fact, the papers do not offer any predictive models for their studied setups. The closest result is perhaps the tree-structuredness metric by Murty et al. (2024); however, that metric merely correlates with their task of interest, i.e., it is not a predictive measure that can preemptively describe at how much amount of training, the model will possess their capability of interest.
>
> - **A novel formal language setup with context-sensitivity for studying neural networks.** Prior works studying neural networks via formal languages primarily focus on *context-free* languages, which are fundamentally constrained in their richness: such languages only possess syntactic properties, thus offering a setup for studying merely syntax acquisition in neural networks. Further, learning such languages is relatively easy, which is likely the reason why model scaling or data scaling rarely has any effect on the transition point where syntax is learned. In contrast, we study *context-sensitive* languages in our work, which are a step towards capturing the richness of natural language, but still offer a controllable setup for performing a theoretical analysis. To our knowledge, we are in fact the first to define a context-sensitive language setup for studying modern neural networks’ learning dynamics. We expect this setup will be of interest to the community at large.
>
> - **A study of emergence.** We also emphasize our work’s primary focus is the study of emergence---to this end, we offer a novel definition of the concept and make a connection to other disciplines (specifically, physics) that can yield insights for further study. However, except for Chen et al., the listed papers by the reviewer are **not** focused on studying emergent abilities. For example, the paper by Allen-Zhu et al. is merely focused on how neural networks trained on a PCFG learn to implement it. Similarly, the paper by Murty et al. on structural grokking, despite its title, does not exhibit sudden / emergent learning---in fact, the learning dynamics of their studied tasks are very smooth. While Chen et al. does share a similar motivation to ours, i.e., to analyze sudden drops in MLMs’ training loss, their focus is solely on syntax acquisition---an entirely context-free property. Meanwhile, we focus on an emergent ability that corresponds to a context-sensitive property (modeling of type constraints).
>
> Overall, given the differences above, we believe the contributions of our work are entirely novel and not captured by prior works.
>
> ---
> ---
> **Further Questions**
>
> > **Do you see similar phase transitions for language learning with smaller models or bigger models? In general, do architecture tweaks change the dynamics in non-trivial ways?**
>
> We did try to elicit effects of model size scaling by increasing both depth and width to twice their current values, but did not see any meaningful changes that could not be captured by our theory---primarily, the transition points seemed to occur earlier by a constant offset, which is expected since larger models are known to train faster. We believe this direction is worth exploring further, e.g., by defining a mixture of tasks similar to Michaud et al. [1], but since our current paper was focused on emergence via data-scaling, we did not push further on it.
>
> [1] The quantization model of neural scaling, Michaud et al. (NeurIPS 2023)
>
> ---
> ---
>
> **Summary.** We again thank the reviewer for their detailed feedback! Our updated paper now better emphasizes experiments stress-testing our models’ understanding of the formal language it is trained on (Section 4 & Appendix F), and has a longer discussion on the analogy between percolation and learning of type constraints (Appendix D.1). In case our answers and these changes help address the reviewer’s concerns, we hope they will consider increasing their score to support the acceptance of our paper.

---

> > ### Author Response · Authors · 2024-11-23
> > **Gentle reminder**
> >
> > Dear Reviewer dr5M,
> >
> > We thank you again for your detailed feedback on our work. Given the discussion period ends soon, we wanted to check in if our provided responses address your concerns, and see if there are any further questions that we can help address.
> >
> > Thanks!

---

> > > ### Comment · Reviewer_dr5M · 2024-11-26
> > > **Thanks**
> > >
> > > Thanks for the comprehensive response!
> > >
> > > I'm still not super convinced about percolation being a _predictive model_ of emergence because it's just a data-driven property (instead of a model-driven property). It's also a property that I cannot see generalizing outside of your current experimental setup to real language modeling.
> > >
> > > But I can see the context of it in this paper, and given the novelty, and experimental thoroughness, I'm willing to increase my score.

---

> > > > ### Author Response · Authors · 2024-11-27
> > > > **Thank you!**
> > > >
> > > > We thank the reviewer for their response and are glad to see our clarifications were helpful. Please let us know if you have any further questions!

---

### Meta-Review · Area_Chair_N8pm · 2024-12-08

**Metareview:**

This paper examines emergence in neural networks through the lens of percolation theory, using transformers trained on formal languages. The work provides a formal definition of emergence, demonstrates phase transitions in learning, and establishes theoretical connections to percolation on bipartite graphs. The reviewers were positive overall, with scores from 6 to 8, praising the paper's take on theory and clear presentation. While concerns were raised about the scope being limited to formal languages and its relationship to prior work on emergence, the authors provided responses that addressed these points. Though the generalizability of the theoretical framework to natural language itself is not yet made clear, this is a worthwhile toy model of a topical phenomenon that will be of interest at ICLR.

**Additional Comments On Reviewer Discussion:**

During the rebuttal period, the authors addressed three key concerns: the relationship to prior work claiming emergence is illusory, the generalizability of results from formal to natural languages, and the physical intuition behind the percolation model. The authors' responses led to increased reviewer scores. Given the paper's strong theoretical foundation, clear presentation, and satisfactory handling of reviewer concerns, the work merits acceptance.

---

### Decision · Program_Chairs · 2025-01-22

Accept (Poster)